# Single cell spatial transcriptomics integration deciphers the morphological heterogeneity of atherosclerotic carotid arteries

The process of arterial atherosclerosis is characterised by accumulation of lipids and fibrous material with accompanying inflammation. As plaques progress, they restrict blood flow and cause rupture, which results in life threatening organ ischemia and dysfunction. Although extensively studied, a clear understanding of plaque heterogeneity and mechanisms that trigger their destabilization remains elusive. Our study reveals the molecular micro-architecture of human carotid artery plaques, using bulk and single-cell RNA sequencing combined with single-cell spatial transcriptomics, for which we present optimized cell segmentation algorithms. We identified distinct plaque morphologies linked to different cell type compositions, impacting early and advanced lesion formation, as well as destabilization. Spatial transcriptomics enabled us to further determine an inflammatory smooth muscle cell subtype, localize regions of neovascularization, and assign hotspots for macrophage activity within distinct cellular neighbourhoods across lesions. For different macrophage substates, we propose gradual and locally contained transdifferentiation of subluminal inflammatory *HMOX1*+ macrophages into a lipid-handling *TREM2*+ phenotype within border zones of the fibrous cap and necrotic core. Our findings provide insight into the complex heterogeneity of human atherosclerosis by unravelling location and proximity of different mural and immune cell substates involved in plaque progression and vulnerability.

Atherosclerosis is a leading cause of cardiovascular disease and contributes to myocardial infarction and stroke[1]. Plaques are complex structures that consist of endothelial cells (ECs), vascular smooth muscle cells (VSMCs), fibroblasts, and immune cells, including macrophages and B and T lymphocytes, with each cell type playing a distinct role during lesion formation and progression[2].

Atherosclerotic plaque development involves complex cellular and molecular interactions initiated by the focal subendothelial retention of apolipoprotein B-containing lipoproteins[3]. Focal EC dysfunction increases permeability and inflammation, and blood-born monocytes enter the nascent lesions, differentiate into macrophages, and, along with adaptive immune cells, secrete inflammatory cytokines and promote an inflammatory environment[4]. Plaque growth is further stimulated by VSMCs that migrate from the media while adapting their phenotype. Both intimal macrophages and VSMCs can become lipid laden (foam cells), presumably due to ingestion of retained lipoproteins in the intima[5] and other yet to be determined mechanisms. ECM

✉e-mail: michael.menden@unimelb.edu.au; lars.maegdefessel@tum.de

components, including collagen and elastin, provide structural support, with ECM (dys)balance influencing plaque stability[6]

Atherosclerotic plaques from patients undergoing carotid endarterectomy (CEA), the surgical removal of plaque buildup in carotid arteries, consist of three major subregions with distinct cellular and molecular features. The necrotic core (NC), largely acellular, contains dead cell debris and cholesterol crystals. A major mechanism of NC development is secondary macrophage necrosis due to failure of proper clearance, or efferocytosis, of apoptotic macrophages by neighboring lesional macrophages[7,8]. The fibrous cap (FC), mainly made of VSMCs and collagen, separates the NC from the arterial lumen. The intima-media layer, rich in VSMCs and fibroblasts, represents the pre-atherosclerotic arterial wall. The adventitia is not excised during the surgical procedure and is retained in the patient. All three substructures with their different cellular and molecular components interact in a dynamic and complex manner. However, this interaction continues to be incompletely understood, which is why research on human lesions remains highly relevant[9].

Sequencing technologies, including bulkRNA-Seq, single cell (sc) RNA-Seq, and spatial transcriptomics, have helped tremendously to better determine molecular alterations and cellular composition in human diseases[10–15]. scRNA-Seq biases toward immune cells, therefore offering limited interpretability of fibroblasts, VSMCs, and apoptotic cells, which limits our understanding of plaque progression[16,17]. Integrating all three methodologies enables high-resolution gene expression analysis, revealing cellular dysfunction and interactions. Spatial transcriptomics preserves spatial context, thus providing key insights into disease mechanisms not captured by other methods.

This study aims to characterise early and late-stage human atherosclerotic carotid plaques, using an integrated multi-transcriptomic approach, combining bulk, single cell, and spatial transcriptomics. Analyzing paired early and advanced plaques from the same patient undergoing CEA, we assessed lesion progression, destabilization, and molecular alterations in plaque development, clarifying dynamic interactions and identifying drivers of plaque progression.

## Results

Human carotid plaques from CEA patients were selected based on conclusive morphology, including remnant media and macroscopically assessed and dissected into advanced atherosclerotic (plaque) and adjacent early lesion (control) tissue from the same patient (matched paired diseased-control samples). Plaques were characterized for stability based on the Oxford Plaque Studies[18–20]. Both pieces of this unique setup (statistically powerful paired analysis possible) underwent bulk- and scRNA-Seq. In addition, formalin-fixed paraffin (FFPE)-embedded samples of neighbouring pieces, maintaining the same paired plaque vs. control configuration, were processed for histomorphological and spatial transcriptomics analyses (Fig. 1).

### Transcriptomic landscape of human atherosclerotic carotid plaques
The initial bulkRNA-Seq analysis compared advanced plaque and early lesion control tissues in unpaired (n = 232, n = 65) and paired (n = 42) designs (Fig. 2A, B). Plaque-control comparisons identified 903 dysregulated genes, with 70% upregulated, increasing to 1775 in paired analyses, with 76% upregulated. Key DEGs that were increased in advanced plaque vs. early lesion controls included immune regulators like FLT4[21] or B-cell markers like CD79A[22] and, mitochondrial stress marker MZB1[23], pro-osteogenic factor STMN2[24], and extracellular matrix degradation enzymes MMP9 and MMP13[25] (Fig. 2B). The advanced plaques were further classified into stable (n = 97) and unstable (n = 126) lesions based on FC thickness above or below 200 μm[18]. Nine advanced plaque specimens (of n = 232) were rendered non-classifiable, and not included into the stability-based comparison.

While a previous study[11] identified 107 DEGs between stable and unstable coronary plaques (n = 10 in total), our bulk transcriptomes of 223 carotid plaques identified only four DEGs, including CXCL5 and PI3K (Fig. 2C). The lower DEG count in our study reflects lower susceptibility to sample-, technical-, and regional-specific biases. In addition plaque stability, when defined solely by FC thickness as a key predictor of clinical outcomes, seemed to not be primarily driven by transcriptional differences at the whole-tissue level. We could further not detect any DEGs by comparing plaques from symptomatic (n = 67) vs. asymptomatic (n = 162) patients (FDR threshold <0.05; minimum FDR in our analysis 0.13).

To resolve cell type-specific gene expression in carotid plaques, we performed scRNA-Seq alongside bulkRNA-Seq on advanced plaque and adjacent early lesion control tissues, integrating data from two studies (Paloschi et al.[26] for the first 10 patients and patients 11-17 in this study). The analysis confirmed the expected cellular heterogeneity, identifying ECs, VSMCs, fibroblasts, macrophages, dendritic cells, mast cells, plasma cells, B cells, NK cells, and T cells (Fig. 2D, E)[27,28], underscoring the complex immune and stromal landscape of carotid artery disease.

Comparative analysis of scRNA-Seq data identified cell type-specific differentially expressed genes undetected by bulkRNA-Seq in plaque vs. adjacent control tissues (Supplementary Fig. 1A-D), including ALOX5 and HIF1A in IL10high macrophages. Interestingly, stable vs. unstable lesion comparisons revealed an even greater number of DEGs (Supplementary Fig. 2A-K), such as CXCR4 and HSPA5 in proliferating T cells, both markers of T cell infiltration in atherosclerosis[29,30]. Additionally, CXCL2 was enriched in ECs, while CCL18 was altered in C1Q+ macrophages, both linked to atherosclerosis[31,32].

Microfluidics-based capture biases in combination with enzymatic and mechanical digestion is a potential issue with scRNA-Seq, leading to an overrepresentation of immune cells. Indeed, our scRNA-Seq revealed high mean macrophage (24%) and T cell fractions (29%), together exceeding VSMCs (22%) and ECs (15%; Fig. 2F). To correct for this, we applied a deconvolution algorithm[33] to estimate cell type proportions in bulkRNA-Seq data using scRNA-Seq-derived labels (Fig. 2G). Deconvolution determined VSMCs as most abundant in plaques (62%) and controls (81%), while macrophages were enriched in plaques (28%) compared to controls (12%; Fig. 2H). Other immune cells, including T and B cells, were rare, comprising less than 0.2% in both plaques and control tissues upon deconvolution. In addition to this deconvolution approach, Xenium-based spatial transcriptomics (designed with an equal amount of probes for all cell types) similar to bulkRNA-Seq confirmed that the observed immune cell dominance (Refs. 34–36) solely occurred in our scRNA-Seq study. Of note, these findings reflect the inclusion of the VSMC-rich media from our CEA specimens (Fig. 1A), a structure that has likely not been analyzed in previous studies that solely focused on the intimal (NC and FC) lesion content.

### Spatial component complements transcriptomic landscape
To localise single cell expression signatures within the plaques, we applied single cell spatial transcriptomics, using the Xenium platform on 12 advanced plaques and 4 early lesions/controls. Four patients provided paired samples, meaning that one CEA specimen from the same carotid artery was divided into early and advanced lesions (Supplementary Fig. 3), while eight provided advanced plaques only. Two panels containing probes for 548 genes were designed. Panel 1 identified cell types based on scRNA-Seq data, and panel 2 captured atherosclerotic disease-associated transcripts (Supplementary Data 1). Cell segmentation and transcript assignment were performed using Baysor[37], as recommended by Salas et al[38]. Testing multiple input and parameter combinations, we selected the optimal segmentation parameters based on Baysor's cell assignment confidence and detected lesion cell types (see "Methods" for details). Manual annotation

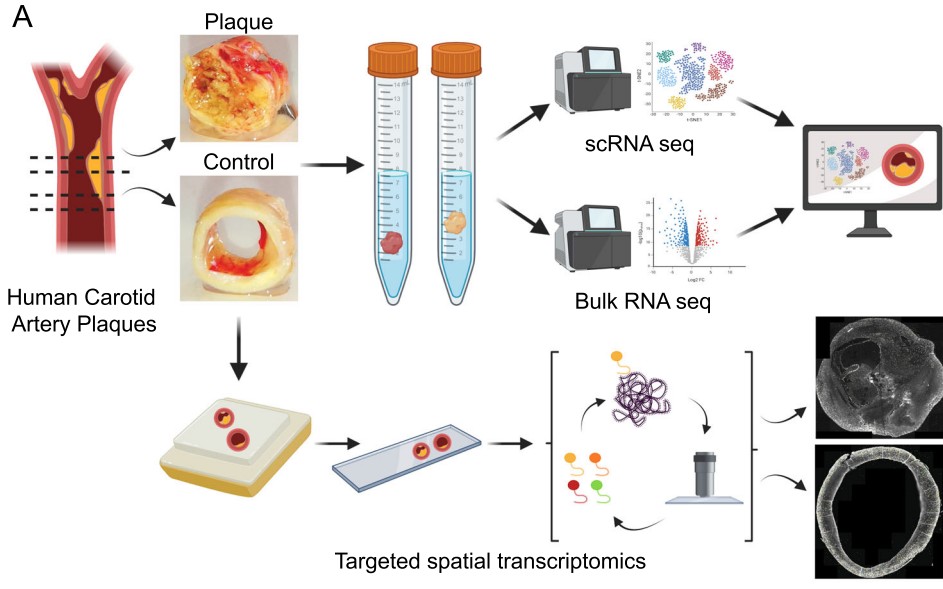

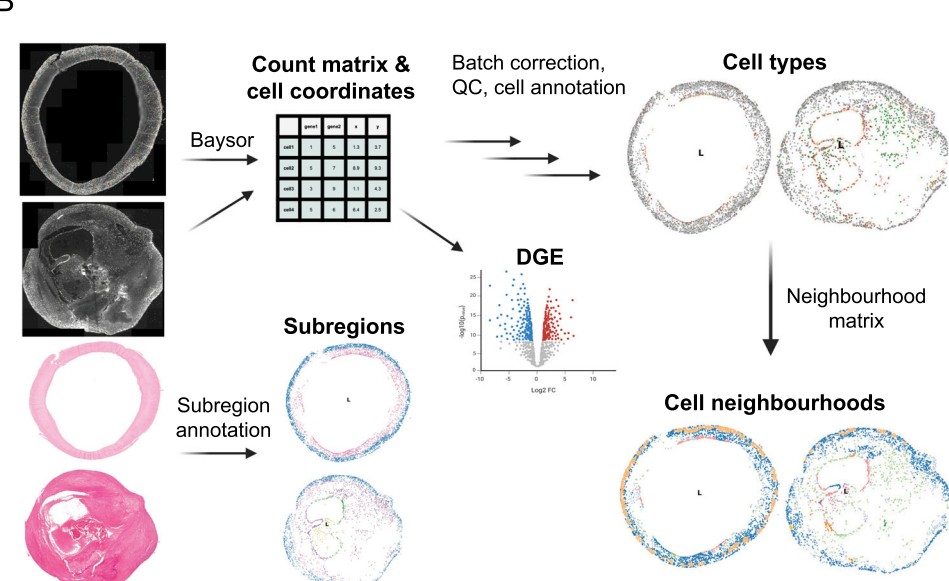

**Fig. 1 | Study design and workflow. A** Overview of study design and workflow. Atherosclerotic lesions were divided into plaque and control and either freshly used for total and scRNA sequencing or embedded (FFPE) and cut for Xenium spatial transcriptomics. Data is combined, analysed and novel cell types are discovered. **B** Xenium spatial transcriptomics workflow. Transcript count matrix and cell coordinates were obtained with Baysor from Xenium output. Using the count matrix we annotated the cells, performed differential gene expression (DGE), and, together with cell coordinates, identified cell neighbourhoods. Subregions of samples were determined based on Hematoxylin-eosin-stained sample slides. Created in BioRender. Paloschi, V. (https://BioRender.com/cjcc8vu).

identified three hierarchical levels: high- and low-level cell types representing primary identities and subtypes (Fig. 3C-D, Supplementary Fig. 4A, B), and low-level substates capturing finer transcriptional differences (Supplementary Fig. 5–8).

Spatial transcriptomics revealed robust cellular profiles across gene panels. Final datasets comprised 2000–17,500 cells per sample for panel 1 (Figs. 3A) and 1000–12,000 cells for panel 2 (Supplementary Fig. 4C), with both panels detecting similar cell types (panel 1: Fig. 3C, D; panel 2: Supplementary Fig. 4A, B), mostly VSMCs and macrophages (Figs. 3B, and Supplementary Fig. 4D). Panel 2 yielded fewer transcript reads per sample, likely reflecting its focus on disease-associated genes rather than cell-defining transcripts (Supplementary Fig. 9A vs. Supplementary Fig. 9B). These findings underscore the value of integrating spatial transcriptomics to refine unbiased cellular and disease-related insights in plaque analysis.

## Subregional Plaque Analysis Reveals Spatial Organization and Cellular Composition

To assess plaque morphology and spatial organization, we performed haematoxylin and eosin (HE) staining, identifying the media, intima, NC, and FC (Fig. 4A, B; and Supplementary Fig. 10, Supplementary Fig. 11). This approach provided a spatial framework for downstream analysis. Plaques consist of media, the outermost layer of CEA plaques, the NC as actual plaque region containing lipids and dead cell debris, as well as the FC separating the lumen from the NC. Plaques sometimes display lumen with thrombi and blood cells, while controls consist only of media, intima, and lumen. However, no tunica adventitia that could be the source of fibroblasts, neurons or lymphatic ECs[14] is excavated during CEA. A cardiovascular pathologist annotated subregions, which were then converted into boundaries for polygons for area

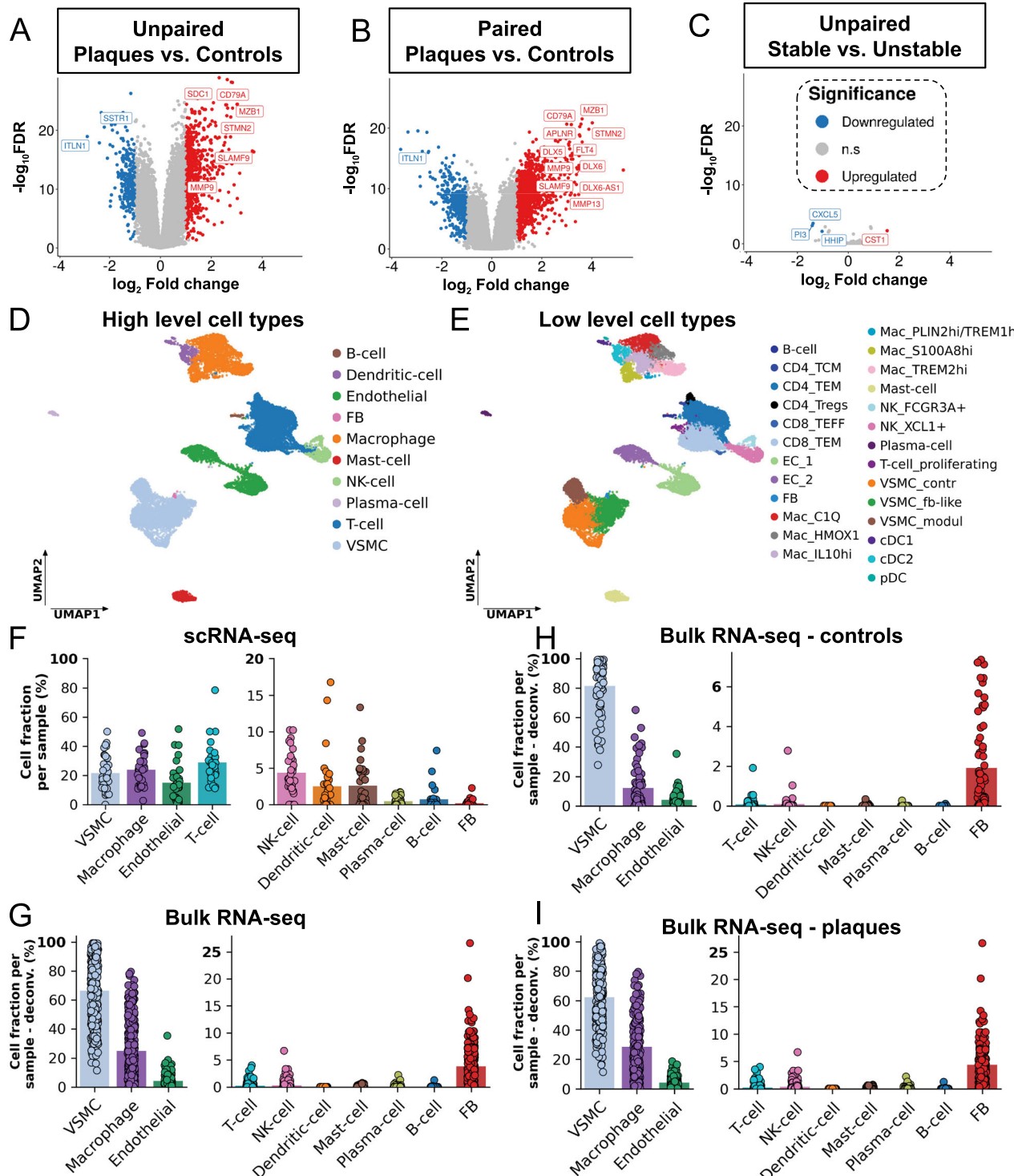

**Fig. 2 | Single-cell and bulk RNA sequencing results. A–C** Differential gene expression analysis was performed with DESeq2 on bulk RNA-seq data: (**A**) unpaired Plaque vs. Control, **B** paired Plaque vs. matched Control from the same patients, and (**C**) unpaired Stable vs. Unstable samples. The x-axis shows $\log_2$ fold change, and the y-axis shows -$\log_{10}$ Benjamini-Hochberg-adjusted p-values (two-sided Wald test; false discovery rate, FDR). Significance thresholds were set at FDR < 0.05 and |$\log_2$ fold change| > 1; significantly upregulated genes are shown in red, downregulated in blue, and genes with Xenium probes are highlighted. **D, E** UMAP low-dimensional representation of gene expression in cells of human atherosclerotic plaques, measured with scRNA-seq method. Cells are coloured by

high-level cell types (**D**), and by low-level cell types (**E**). **F, G** Cell fractions of high-level cell types per sample in scRNA-seq (**F**) $n = 31$) and bulk RNA-seq (**G**), $n = 297$). In the case of bulk RNA-seq, cell fractions are calculated by a deconvolution algorithm. Bars represent the mean of cell fractions per cell type across all samples. **H, I** Cell fractions of high-level cell types of control (**H**), $n = 65$) and plaque (**I**), $n = 232$) bulk RNA-seq samples, calculated by deconvolution. For clarity, cell types in panels (**F–I**) with lower fractions are displayed using a secondary y-axis with rescaled limits. Bars in (**F–I**) indicate the mean. UMAP: Uniform Manifold Approximation and Projection, *FB* fibroblast, *NK-cell* natural-killer cell, *VSMC* vascular smooth muscle cell.

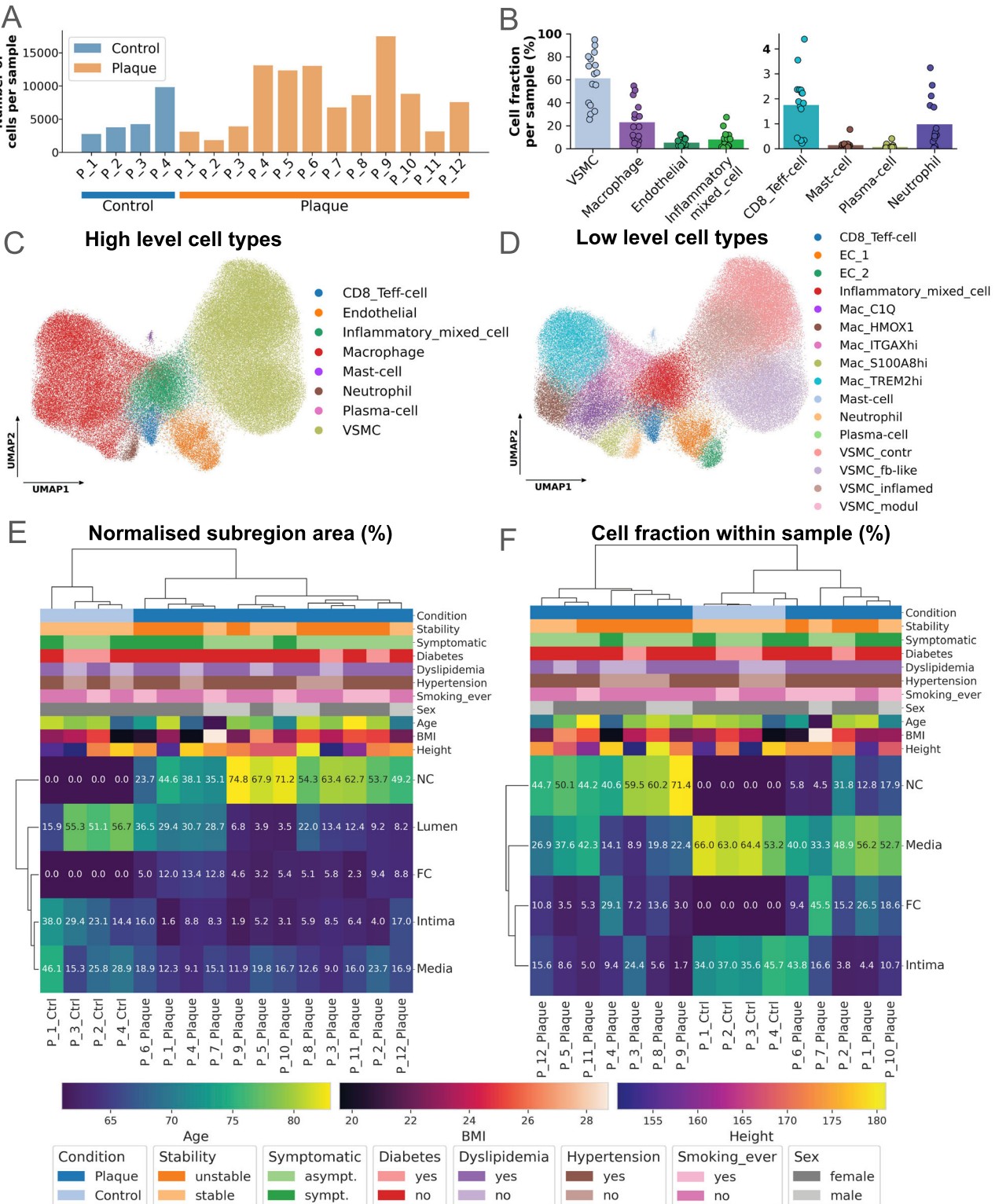

**Fig. 3 | Overview of cells coming from spatial transcriptomics with panel 1 genes and sample clusterings. A** Number of cells per sample, measured with Xenium panel 1 genes. „P" stands for „Patient" in the x-axis labels. **B** Cell fractions of high-level cell types per sample with panel 1 genes (*n* = 16; bars indicate the mean). For clarity, cell types with lower fractions are displayed using a secondary y-axis with rescaled limits. **C, D** UMAP low-dimensional representation of gene expression in cells of human atherosclerotic plaques, measured with Xenium method, panel 1 genes. Cells coloured by: high-level cell types (**C**), and by low-level cell types (**D**). **E** Hierarchical clustering of all samples based on normalised subregion areas.

Normalisation was performed to total sample area. **F** Hierarchical clustering of all samples based on relative cell fractions in the 4 main cell-rich subregions. Cells were derived by Xenium panel 1 genes, normalisation was performed to total number of cells per sample, patient metadata are depicted below clustermap. „P" stands for „Patient" in the x-axis labels of (**E**) and (**F**), patient metadata legend is depicted below clustermaps (**E**) and (**F**). UMAP: Uniform Manifold Approximation and Projection, *NC* necrotic core, *FC* fibrous cap, *sympt.* symptomatic, *asympt.* asymptomatic, *VSMC* vascular smooth muscle cell, *Ctrl* control.

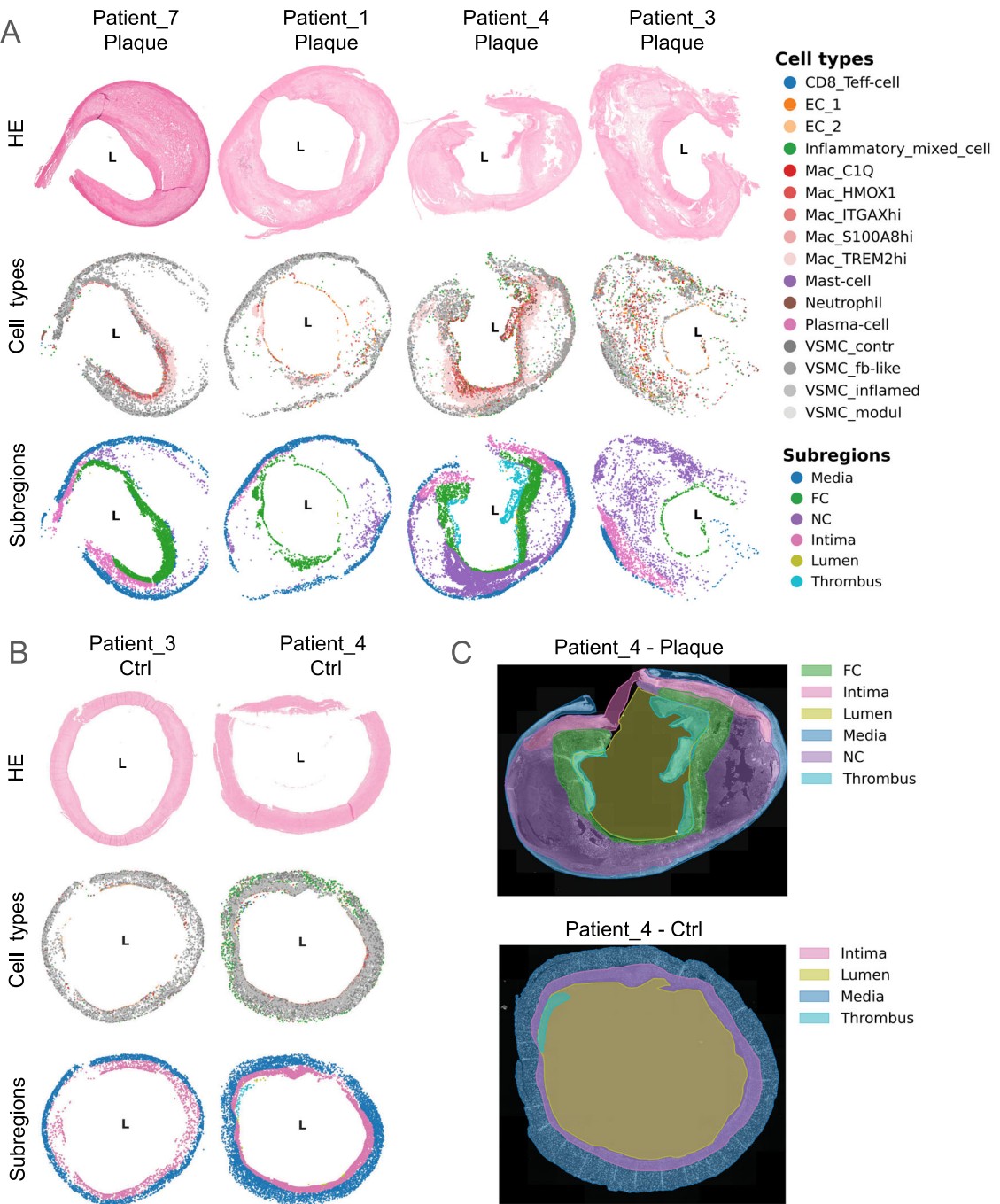

**Fig. 4 | General plaque compositions in panel 1. A** The upper row shows HE stainings of the plaques, the middle row indicates the panel 1 low-level cell types, and the lower row shows the subregions of representative plaques in panel 1. L indicates the lumen. (**B**) The upper row shows HE stainings of the controls, the middle row indicates the panel 1 low-level cell types, and the lower row shows the subregions of representative controls in panel 1. L indicates the lumen. **C** Subregion annotation example shown on Patient 4 plaque (upper panel) and control sample (lower panel). The shaded areas represent the subregion polygons created based on the manual subregion annotation. These polygons were used for subregion area calculation. *HE* hematoxylin-eosin, *NC* necrotic core, *FC* fibrous cap, *VSMC* vascular smooth muscle cell, *Ctrl* control.

calculations and cell assignments (Fig. 4C). Structural distortions in some samples required approximation of subregion borders.

Clustering of normalized and absolute subregional areas distinguished plaques from controls, which lacked FC and NC regions and showed moderate variability in intima and media areas (Fig. 3E; Supplementary Fig. 4E). Plaques formed three clusters: one with ~30% normalized lumen area and NC below 50% (patients 1, 4, 6, 7), another dominated by NC (~70%) with a narrow lumen (<7%) (patients 5, 9, 10), and an intermediate group with NC ~ 50% and lumen 10–20% (patients

2, 3, 8, 11, 12). Correlation analysis revealed a strong inverse relationship between normalised NC and lumen areas (Supplementary Fig. 12B), which could imply a constant sum, however, this was not observed (Supplementary Fig. 13). In contrast, absolute area correlations showed no association between NC and FC or intima, suggesting NC expansion reduces the relative proportions of other regions (Supplementary Fig. 12A). A weak negative trend between absolute NC and lumen areas suggests NC growth narrows the lumen without affecting intima and media size.

Associations between plaque morphology, patient metadata, and medical conditions showed distinct trends (Supplementary Fig. 14–17). Patient age inversely correlated with FC absolute area, height correlated positively with total plaque area (Supplementary Fig. 14A, 15C), and hypertensive patients presented with increased normalized media areas (Supplementary Fig. 17A). Plaque stability and symptoms (e.g., stroke or transient ischemic attacks) showed no correlation with subregion morphology.

Clustering based on relative cell distributions across the four main subregions (media, intima, FC, NC) differentiated plaques from controls (Fig. 3F). Controls formed a distinct cluster with a consistent media-to-intima cell ratio (~65% to 35%), except for patient 4 (45% intima). Plaques clustered into two groups: one with most cells around the NC (patients 3, 4, 5, 8, 9, 11, 12) and another with higher media cell fractions and sparsely populated NCs (patients 1, 2, 6, 7, 10). Comparison with normalized region clustering (Fig. 3E) revealed that some plaques with enlarged NCs retained proportional cell fractions (patients 3, 5, 8, 9), while others had disproportionately low NC cell content (patients 1, 7, 10), suggesting advanced necrosis, as confirmed by DAPI-stained spatial transcriptomics (Supplementary Fig. 18).

Control samples consisted mainly of VSMCs with few macrophages in the neointima, while plaques exhibited diverse cell types (Fig. 4A, B; and Supplementary Fig. 10 and Supplementary Fig. 11). In plaques, VSMCs localized to the media and FC, macrophages accumulated around the NC and in the FC, while ECs lined the lumen - but interestingly also appeared intraplaque near the NC where they co-localized with immune cells. In this area, intraplaque haemorrhages of leaky neovessels facilitate immune cell infiltration, promoting subsequent inflammation. Immune cells, apart from macrophages, predominantly accumulated in neovessel-rich regions of advanced plaques, highlighting the link between neovascularization and immune activation (panel 1: Supplementary PDF 1, panel 2: Supplementary PDF 2).

## Morphological plaque clustering

We performed hierarchical clustering of plaques using the relative low-level substate cell fractions per sample. This grouped our plaques into four distinct morphological clusters with a unique composition of structural and cellular features in every cluster (panel 1: Fig. 5, panel 2: Supplementary Fig. 19B). For intelligibility we decided to term the two main groups of our plaque clusters "structured" (morphological clusters 1 & 2) and "chaotic" (morphological clusters 3 & 4).

Morphological cluster 1 consists of plaques of patients 7 and 10 (Fig. 6A). Both plaques are organized in a round shaped contractile VSMC-rich media, a huge cell free NC and a stable, thick FC (with many "synthetic" fibroblast-like VSMCs), and shoulders rich in *HMOX1+* and *TREM2+* macrophages. Morphological cluster 2 consists of 4 plaques (patients 1, 2, 6 and 11) that are all VSMC-rich (contractile, inflamed, and fibroblast-like) with panel 1 (Fig. 6B), while macrophages and other immune cells appear underrepresented. These plaques have huge, rather cell free NCs and overall, a similar morphology as morphological cluster 1. Panel 2 shows similar cell type compositions across these morphological clusters. Cluster 1 is characterised by *TREM2+* macrophages and VSMC substates (contractile and fibroblast-like), while cluster 2 predominantly consists of different VSMC substates (Supplementary Fig. 19B).

Morphological cluster 3 consists of plaques from patients 4, 8 and 9 (Fig. 6C). They are less structured, showing two NCs on each side of the lumen (potentially one NC in another plane of the plaque; patient 4 and 8) or one large surrounding NC (patient 9), that are cell-rich with many macrophages and fibroblast-like VSMCs. The media is not as present as in morphological clusters 1 and 2 (Fig. 6A, B). With panel 2, these plaques are characterised by macrophages, fibroblast-like and inflamed VSMCs.

Morphological cluster 4 is even more chaotic (Fig. 6D). It is marked by numerous inflamed and fibroblast-like VSMCs in panel 1, as well as *C1Q+* macrophages and T-cells in panel 2, infiltrating the cell-rich NC. *C1Q+* macrophages are characterised by high *SELENOP* expression, an antioxidative gene, as well as *CD14*, which apart from its role in inflammation mediates apoptotic cell clearance[39], enhancing phagocytosis and efferocytosis[34]. Upregulation of *CCL8* in these cells is indicative of a "M2-like" polarization[40]. In plaques from patients 3, 5, and 12, the media is either not present at all or very thin, which could be due to minor differences in surgical removal of plaques during CEA (Supplementary Fig. 10A, and Supplementary Fig. 11B).

To explore regional differences within plaques, we performed hierarchical clustering using relative low-level substate fractions calculated per subregions in each sample. FC-clustering identified four main clusters (Supplementary Fig. 20A), a macrophage-rich (patients 1, 4, 7, 8), an EC-rich (patients 3, 11), one high in contractile VSMCs (patients 6, 10), and one high in fibroblast-like VSMCs (patients 2, 9, 12).

Intima substate fractions clustering (Supplementary Fig. 20B) formed two major clusters: a macrophage-rich with fibroblast-like VSMCs (patients 4, 5, 7, 12) and a VSMC rich (contractile and fibroblast-like; patients 1, 2, 3, 6, 8, 9, 11). The media substate fraction clustering split the plaques into two groups (Supplementary Fig. 20C), one with high fibroblast-like and low contractile VSMC fractions (patients 4, 5, 9, 12), and vice versa (patients 1, 2, 3, 6, 7, 8, 10, 11). Interestingly, samples from patients 3 and 8 had previously organized into the morphological clusters 3 and 4 (Fig. 6C, D), however their media substate fractions appeared more similar to the media of contractile VSMC substate-dominated plaques.

Clustering with NC substate fractions (Supplementary Fig. 20D) identified a group rich in various *TREM2*hi macrophage substates (patients 4, 8, 9, 10), encompassing plaques of morphological cluster 3, and another group with high fractions of fibroblast-like VSMCs and *C1Q* macrophages, containing almost all plaques of the other morphological clusters (patients 1, 2, 3, 5, 6, 7, 11, 12). Similar results with slight variations were observed with panel 2 substate clustering (Supplementary Fig. 21A–D). These results show that despite similar morphology and composition, plaques still exhibit significant regional heterogeneity.

## Smoking status impacts carotid plaque progression

We noticed a separation pattern with smoking status ("Smoking ever") in the plaque clustering (Fig. 5). Plaques from current or former smokers grouped into morphological clusters 1 and 2 (except for patient 1), whereas morphological clusters 3 and 4 included non-smoking patients only. This was driven by higher contractile VSMC cell fractions in smokers, seen in both, low-level cell types (Supplementary Fig. 22A–E, Supplementary Fig. 22K–N) and substates (Supplementary Fig. 22F–J, Supplementary Fig. 22O–U). However, we did not observe reproducible differences in VSMC cell fractions across smoking status with either bulk- or scRNA-Seq which reflects rigor in assessing reproducibility of our findings (Supplementary Fig. 23A, B).

Smoking status was associated with the most DEGs across metadata variables in our spatial transcriptomics approach (Supplementary Fig. 24, Supplementary Fig. 25, Supplementary Data 2, Supplementary Data 3). To validate these findings with more robust data, we conducted differential expression analysis with our bulkRNA-Seq dataset across plaque samples ($n = 99$ "smoking" vs. $n = 106$ "non-smoking"; Fig. 7A, and Supplementary Data 4), identifying 44 overlapping DEGs between spatial transcriptomics and bulkRNA-Seq with concordant (48%) (Fig. 7A, and Supplementary Fig. 26, Supplementary Fig. 27, Supplementary Data 5) and discordant (52%) (Supplementary Fig. 19A, Supplementary Fig. 26, Supplementary Fig. 27) $\log_2$ fold change values.

*ATM* was concordantly upregulated in smokers. This gene has previously been linked to lung cancer development by increasing the

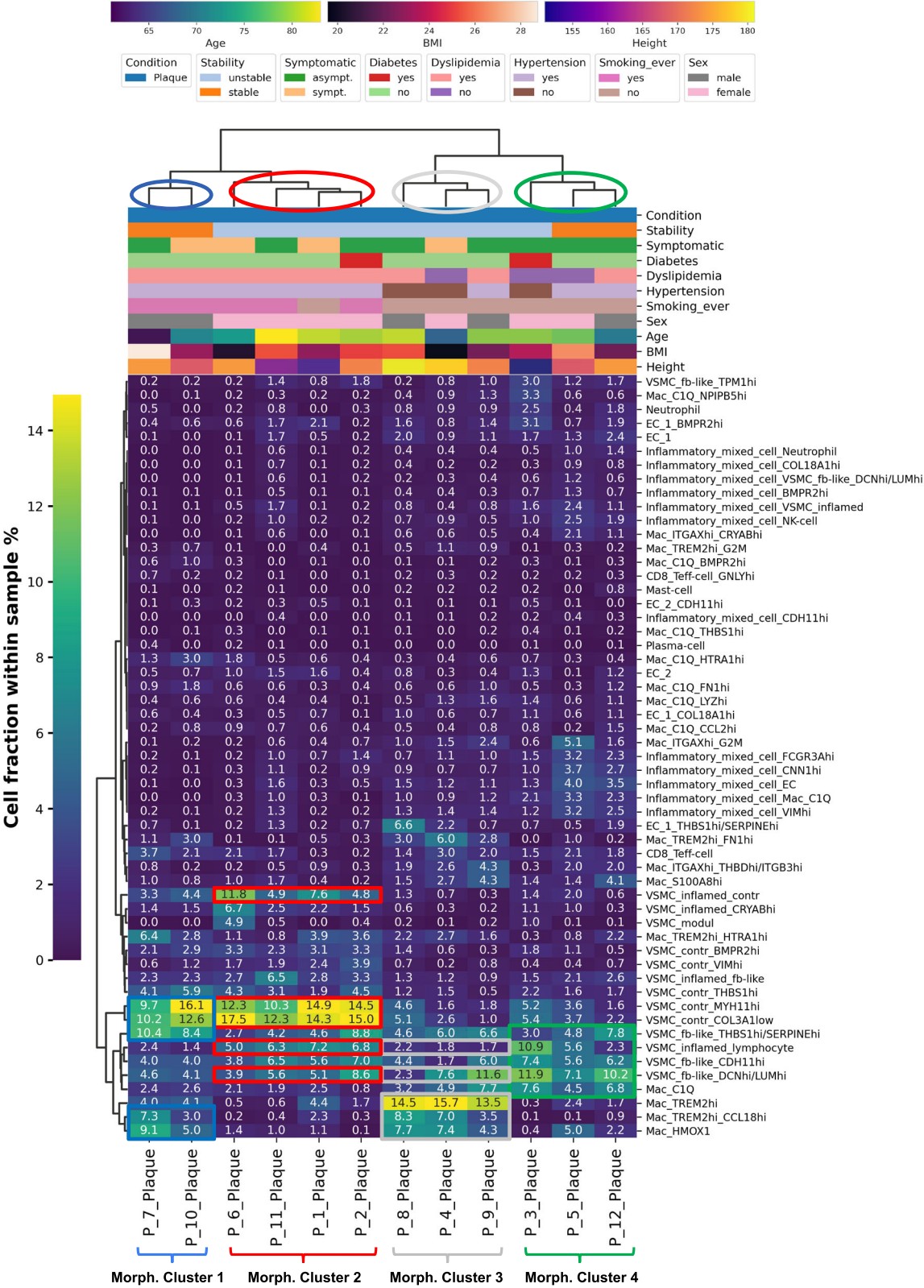

**Fig. 5 | Clustering of plaques into morphological clusters.** Clustermap showing hierarchical clustering of plaques into 4 morphological clusters based on panel 1 low-level substate cell fractions within each sample. „P" stands for „Patient" in the x-axis labels, patient metadata legend is depicted above clustermap. „sympt". symptomatic; „asympt.": asymptomatic, VSMC: vascular smooth muscle cell.

risk in non-smokers compared to smokers[41]. Tissue fibrosis related genes *BMPR2* and *TGFBR1*, associated with smoking in lung fibrosis[42,43], were also upregulated in smokers. Inflammatory genes *IL6* and *IFNG* were downregulated in plaques of smokers, while *ADAMTS7*,

downregulated in smokers, was reported to be lower in current smokers *vs.* non- and ex-smokers in human CEA specimens[44]. Fig. 7B shows the spatial distribution of *ATM* and *TGFBR1* as representative genes in two plaques (patients 7 and 10).

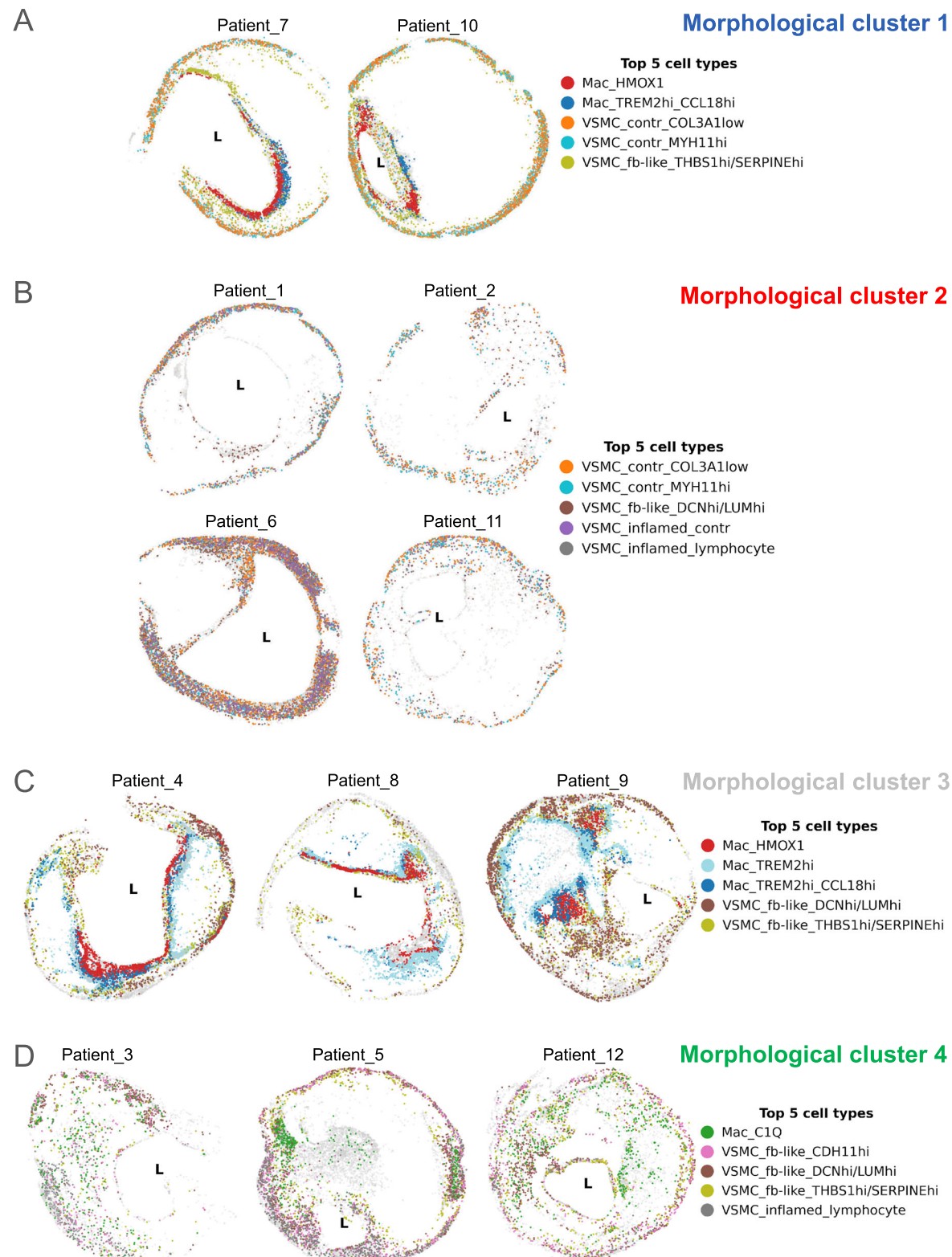

**Fig. 6 | Spatial scatterplots of morphological clusters. A–D** Spatial scatterplots of plaques grouped by morphological clusters. Dot colouring refers to the top 5 most abundant cell subtypes per cluster, „L" depicts lumen. *FC* fibrous cap; *NC* necrotic core, *VSMC* vascular smooth muscle cell.

Although plaques rendered as stable were enriched in morphological clusters 1 and 4, no obvious cell substate fractions explained this clustering pattern. In addition, no other patient metadata variables showed discernible patterns of distribution across any of the morphological clusters (Fig. 5, and Supplementary Fig. 19B).

## Cellular neighbourhoods determine the most prominent plaque features

To fully utilize the potential of our combined single cell and spatial transcriptomics data, we next investigated the composition of our samples on the cellular scale. Our goal was to identify

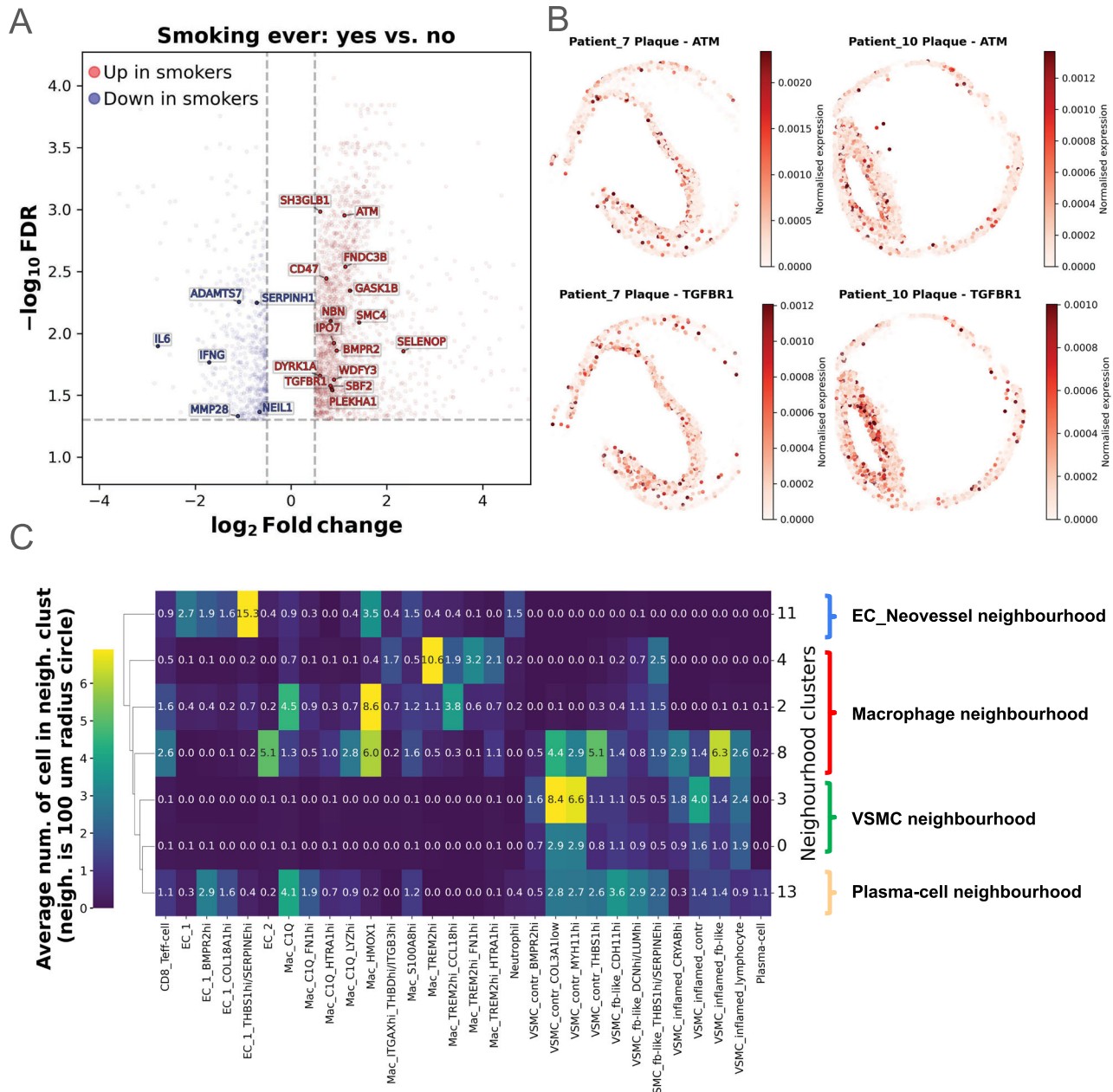

**Fig. 7 | Smoking related DGE results and cell signatures of interesting panel 1 cell neighbourhoods. A** Bulk RNA-seq differential gene expression (DGE) results between „Smoking ever" conditions of plaques. The same DGE comparison was conducted with both Xenium gene panels. Labelled genes are significantly up (red) or downregulated (blue) in both bulk and Xenium, with concordant log2 Fold change directions. FDR: false discovery rate, Benjamini-Hochberg adjusted significance level of gene expression change. Vertical dashed line: log2 Fold change threshold at +/- 0.5, horizontal dashed line: FDR threshold of −log10(0.05). **B** Example of plaques from patients with a history of smoking. Cells are coloured by normalised expression of 2 representative genes, selected from the genes concordantly upregulated in Xenium and bulk RNA-seq in patients who ever smoked. **C** Clustermap showing the average local neighbourhoods (signatures) of panel 1 neighbourhood clusters of interest. Clusters 2,4,8 were termed as Macrophage neighbourhood, clusters 0 and 3 as vascular smooth muscle cell (VSMC) neighbourhood, cluster 11 as endothelial cell (EC) neovessel neighbourhood, cluster 13 as Plasma-cell neighbourhood. For brevity, only the most abundant substates of selected clusters are shown on the x-axis.

similar local cell neighbourhoods across samples that could reveal local processes impacting plaque development and destabilization.

After defining the local neighbourhood of a cell (counting nearby low-level substates within a 100μm radius), we detected 21 neighbourhood clusters in panel 1 and 26 in panel 2 (see "Methods" section for details). We averaged the local neighbourhoods of cells within each neighbourhood cluster, creating local neighbourhood signatures (Supplementary Fig. 28A, B). We next focused our analysis on neighbourhood clusters 0, 2, 3, 4, 8 and 11 of panel 1 due to their interesting local signatures (Fig. 7C).

Neighbourhood clusters 2 and 4 were identified as "macrophage neighbourhoods", with different macrophage substates residing in close proximity to each other. Cluster 2 mainly consists of substate *C1Q+*, a known enhancer of efferocytosis[45], and the substate of *HMOX1+* involved in heme degradation[34]. Cluster 4 is dominated by *TREM2*hi macrophages[35,46], lacking overexpression of inflammatory genes[34]. Substate Mac_TREM2hi_CCL18hi, a slightly transitioned state of classical *TREM2*hi macrophages, localises to both clusters, but is more prevalent in cluster 2.

In plaques of patients 4, 8 (Figs. 8A), 7, and 10 (Supplementary Fig. 29A), most of the four macrophage substates reside in the crucial,

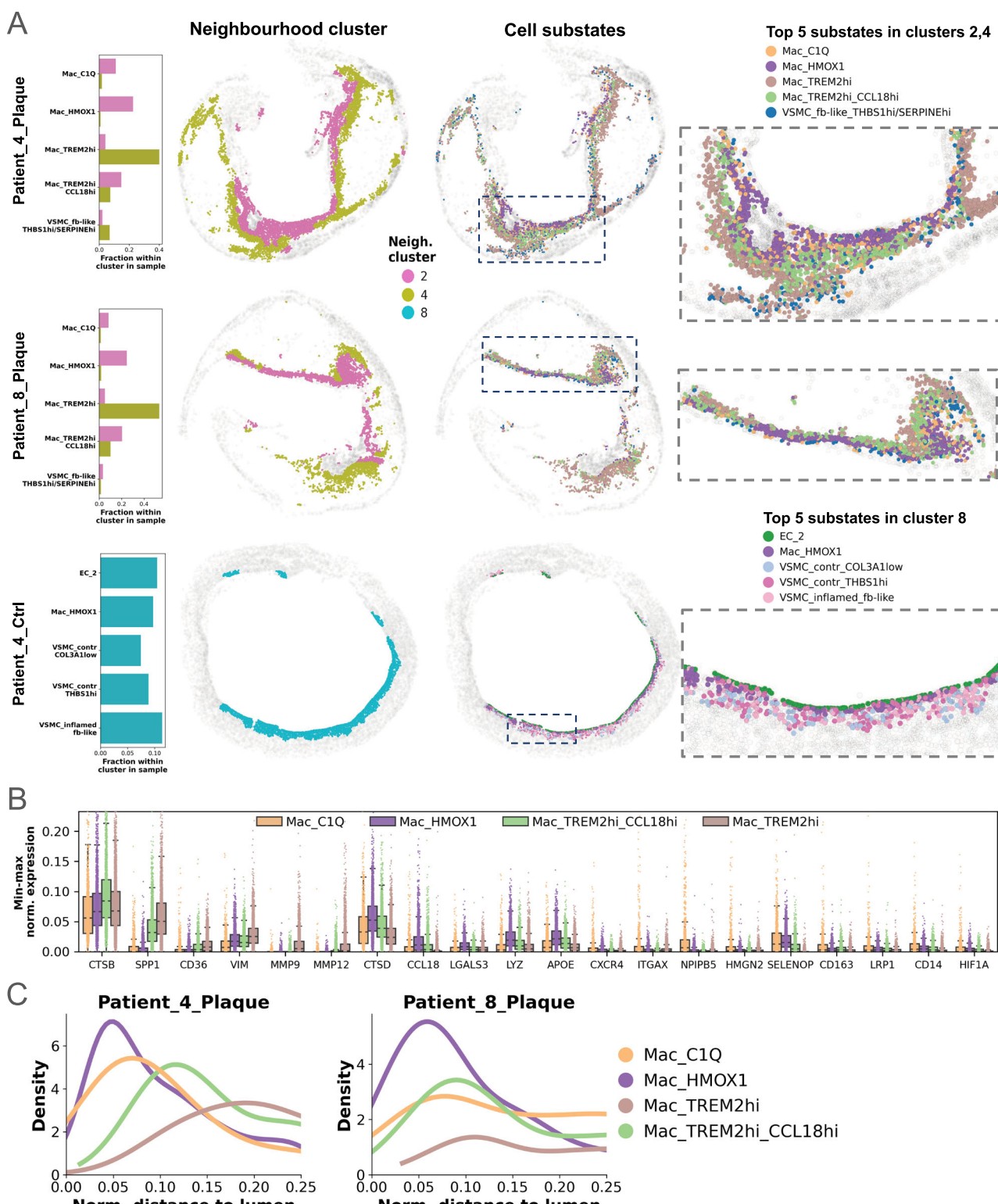

**Fig. 8 | Macrophage neighbourhoods. A** Composition of macrophage neighbourhoods shown based on the five most abundant cell substates per panel 1 neighbourhood cluster in representative plaques. The first column displays relative fractions of these substates within each neighbourhood cluster per plaque. The second column shows their spatial distribution coloured by neighbourhood cluster, while the third column presents the same distribution coloured by cell substate. The fourth column provides a zoomed-in view of regions with interesting substate spatial distribution. Dashed rectangles in column three represent the original location of interesting regions within sample. **B** Per sample min-max normalised

expressions of four macrophage substate cells (*n* = 9113) belonging neighbourhood clusters 2&4 in plaques of patients 4,7,8,and 10, coloured by cell substate. Centre lines indicate the median, boxes represent the interquartile range (IQR; 25th–75th percentile), and whiskers extend to the most extreme values within 1.5 x IQR. **C** Density plot showcasing the ordered layering of four macrophage substates in the two selected plaques. *X* axis depicts the normalised distance from the lumen, (capped at 0.25 to only consider subluminal regions), *y* axis depicts the distribution of distances. *VSMC* vascular smooth muscle cell, *Ctrl* control.

rupture-prone shoulder region within FCs, displaying an interesting common spatial pattern from lumen towards NC. *C1Q*+ and *HMOX1*+ macrophages are located subluminally, followed by Mac_TREM2hi_CCL18 and finally *TREM2*hi macrophages (Fig. 8C, and Supplementary Fig. 29B). Spatial correlation analysis underpinned this and showed significant colocalization of *C1Q*+ and *HMOX1*+ macrophages near the lumen. Neither of them showed spatial association with *TREM2*hi cells in any other lesion area, a pattern that is consistent in all four plaques (Supplementary Fig. 30, and Supplementary Fig. 31). Interestingly, the colocalization pattern of *C1Q* +, *HMOX1*+ and *TREM2*hi appears in other samples, but not necessarily in the FC (Supplementary Fig. 32–34).

The distinct spatial distributions of macrophage substates suggest differential roles in plaque development and progression. Subluminal *HMOX1*+ macrophages, beyond their role in heme degradation, highly express the antioxidant *SELENOP*, as well as the known lipid handling mediator *APOE* (Fig. 8B, and Supplementary Fig. 35E). *TREM2*hi cells, located deep in the shoulder regions of FCs, express *SPP1* at very high levels, as well as the ECM regulators *FN1*, *VIM*, *MMP9*, and *MMP12* (Supplementary Fig. 35D, E), suggesting their relevance in fibrosis[47] in addition to lipid handling. These *TREM2hi* macrophages seem likely involved in plaque shoulder remodelling within the FC[48]. A decreasing gradient of expression in *CCL18*, *CTSD*, and *LYZ* was observed from *HMOX1*+ to *TREM2*hi macrophages, while *SPP1*, *MMP9*, and *MMP12* show the opposite trend (Fig. 8B, and Supplementary Fig. 35A). Based on our spatial and DEG results, we hypothesize a gradual transdifferentiation of *HMOX1*+ into *TREM2*hi with Mac_TREM2hi_CCL18 as an intermediate, transient state. We assessed this with velocity analysis using TFvelo[49], which showed increasing mean pseudotime along subtypes Mac_C1Q → Mac_HMOX1 → Mac_TREM2hi_CCL18hi (panel 1) → Mac_TREM2hi (Supplementary Figs. 36, 37), accompanied by a lumen-to-NC gradient in patients 4, 7, 8, and 10 (Supplementary PDF 3,4). Partition-based graph abstraction (PAGA) supported the same progression (Supplementary Figs. 38C, 39C), though the link between Mac_TREM2hi_CCL18hi and Mac_TREM2hi was weak in panel 1. Despite not being fully concordant across all subtypes, complementary analyses of our scRNA-Seq data with two orthogonal methods (TFvelo, scVelo[50]) yielded consistent trajectories from *HMOX1*+ to *TREM2*hi cells and lower pseudotime values for *HMOX1*+ relative to *TREM2*hi (Supplementary Figs. 40; Supplementary Fig. 41), suggesting a possible transdifferentiation route that requires future experimental validation.

Neighbourhood cluster 8 appears specific to the control tissue of patient 4 (Fig. 8A), an early lesion with dawning infiltration of macrophages in the intima layer. We observed a subluminal accumulation of *HMOX1*+ macrophages, with neighbouring VSMCs and EC_2 cells, which display an ECM modulation phenotype by highly expressing *ITLN1, CDH11* and *OMD*[51,52]. Interestingly, the subluminal *C1Q*+ and *HMOX1*+ cells are not spatially separated from *TREM2*hi cells here (Supplementary Fig. 32), suggesting that this separation occurs later during plaque advancement.

Neighbourhood cluster 11, termed "EC neovessel" neighbourhood, was uncovered only in two plaques (patients 4, 8). It consists of *HMOX1*+ macrophages and EC_1 substates, with the *THBS1* + /*SERPINE*+ substate dominating the cluster (Fig. 9A). This appears to be an area of neovascularization, as *THBS1* and *SERPINE1* are closely linked to angiogenesis[53] and EC dysfunction[54]. Interestingly, this neighbourhood is located in shoulder regions of plaques, rather than luminally, suggesting a potential entry point for immune cells into the lesion core through leaky neovessels formed by pro-angiogenic EC_1 cells. These cells upregulate *ACKR1*, recently suggested to play a role in human plaque neovascularization and immune cell trafficking[55].

Neighbourhood clusters 0 and 3 were termed "VSMC neighbourhoods", as they mainly consist of contractile VSMCs along with "synthetic" fibroblast-like and inflamed VSMC substates (Fig. 9B),

detected in the media of plaques (patient 7, 10: Fig. 9B; patient 1, 3, 11: Supplementary Fig. 42) and control (patient 3, Fig. 9B). Only very few "classical" fibroblasts could be identified overall. As mentioned above, the surgically removed CEA specimens lack the adventitial layer where these cells typically reside in human arteries[56].

Cluster 3 presents a high abundance of "classical" contractile VSMCs (substates *MYH11*hi and *COL3A1*low) and is situated on the outer ring of the media. This cluster seems to represent remnants of the healthy arterial wall with high contractile VSMC density. Cluster 0 is positioned more inwards facing the NC and FC. It contains similar transitioning VSMC substates as cluster 3, but lower contractile VSMC counts. This cluster represents areas of media with less densely packed structures, based on the lower average substate counts in its signature (Fig. 7C). This suggests that the media not only consists of contractile VSMCs but offers a gradual shift from contractile to modulated cell states towards the lesion core.

A recent publication identified plasma and B-cell rich areas, interspersed with VSMCs and macrophages as plaque tertiary lymphoid organs (PTLOs)[12]. In our samples, spatial analysis revealed significant plasma cell enrichment in the plaques of patients 6 and 7 (Supplementary Fig. 43A,B; Supplementary Fig. 44A, B). In addition, the plasma cell aggregation domain in the NC of patient 6 was identified as cluster 13 (Supplementary Fig. 45), enriched in VSMCs, macrophages and plasma cells (Supplementary Fig. 28A). Although suggestive of PTLOs, only two (*MZB1*, *XBP1*) out of four PTLO marker genes[12] in our panels showed high expression in plasma cells (Supplementary Fig. 43C, D; Supplementary Fig. 44C, D), highlighting the need for further clarification with more specialised gene panels.

By performing neighbourhood clustering for panel 2, we detected similar neighbourhoods with similar features in the same plaques, with neighbourhood 0 and 1 being "VSMC neighbourhoods", neighbourhoods 2, 5 and 8 being "macrophage neighbourhoods", neighbourhood 10 being the "EC_Neovessel neighbourhood", and neighbourhood 13 the "plasma-cell neighbourhood" (Supplementary Fig. 28B, C).

## Discussion

Atherosclerotic plaques are heterogeneous structures of cellular, lipid-rich, fibrotic, and calcified components that dynamically influence morphology[57]. The composition varies significantly across individuals and plaque subregions, affecting plaque stability and occurrence of life-threatening ischemic events[58]. Understanding the morphology of these endarterectomy specimens is critical to improve clinical decision making for patients at risk for stroke or myocardial infarction. Traditionally human plaques have either been analysed and compared based on isolated plaque features (e.g., stable vs. vulnerable[59], calcified *vs.* noncalcified[60,61], lipid-rich *vs.* fibrous[62]), or more commonly by patient symptomatology (asymptomatic vs. symptomatic for stroke, transient ischemic attack or amaurosis fugax[63,64]).

Bulk, single-cell and spatial transcriptomics offer complementary advantages for transcriptomic profiling. scRNA-Seq provides single-cell resolution, enabling the unbiased identification of rare cell types and subpopulations[65]. However, it is limited by high cost, complex analysis, and dissociation-related biases that may result in cell-type overrepresentation[16,17]. Alternatively, this bias could be overcome by performing single nuclei sequencing, with the downside of losing cytoplasmic RNAs[66,67]. Bulk RNA-Seq offers deeper transcript coverage and is more cost-effective and analytically tractable but lacks cellular resolution and masks tissue heterogeneity[68]. Spatial transcriptomics retains morphological context, detects thousands of transcripts, and, in some platforms, achieves near single-cell resolution while being compatible with FFPE samples[69].

We used Baysor with multiple parameters and data input combinations to achieve optimal cell segmentation for each sample. Our results highlight the critical role of an appropriate scale parameter,

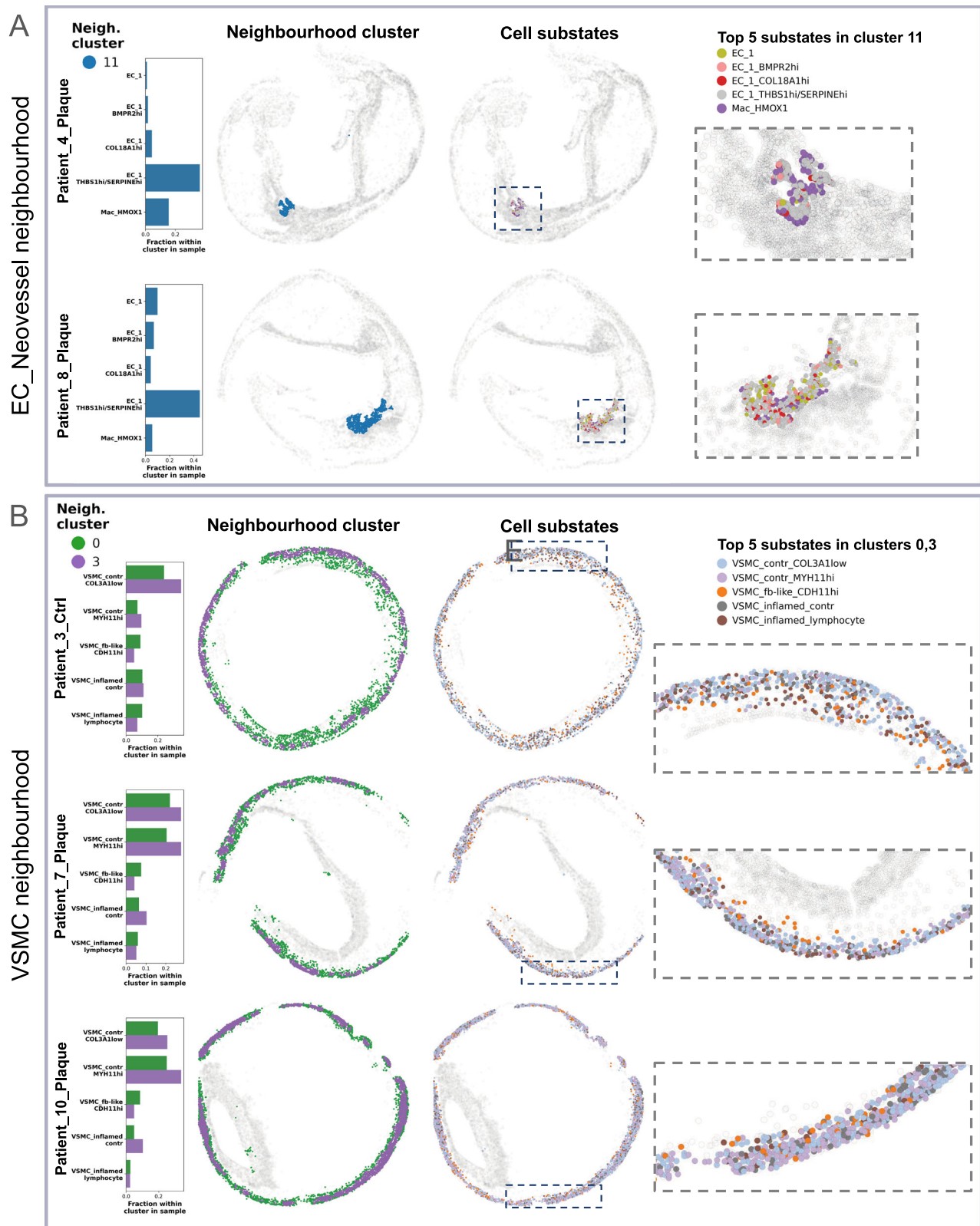

**Fig. 9 | Endothelial cell (EC) and vascular smooth muscle cell (VSMC) neighbourhoods.** Composition of EC neovessel (**A**) and VSMC (**B**) neighbourhoods shown based on the five most abundant cell substates per panel 1 neighbourhood cluster in representative samples. The first column displays relative fractions of these substates within each neighbourhood cluster per plaque. The second column shows their spatial distribution coloured by neighbourhood cluster, while the third column presents the same distribution coloured by cell substate. The fourth column provides a zoomed-in view of regions with interesting substate spatial distribution. Dashed rectangles in column three represent the original location of interesting regions within sample. *Ctrl* control.

representing the sample's approximate cell radius. We found a trade-off effect with the three applied scale parameters, i.e., 5, 10 and 15 µm. The low value led to a failure in detecting large-sized macrophage subtypes, whereas the high setting resulted in segmenting multiple cells as one (especially spindle-shaped VSMCs). A 10 µm scale led to a balanced detection of cell subtypes of varying shapes and sizes.

This study evaluated key clinical variables and risk factors associated with carotid stenosis. Despite the limited sample size for spatial transcriptomics (*n* = 12), smoking status was linked to distinct cellular and morphological plaque features, including a higher proportion of contractile VSMCs and differentially expressed genes in T cells, macrophages, and VSMCs. Bulk RNA-Seq analysis (99 smokers *vs.* 106 non-smokers) identified *ATM* and *ADAMTS7* as candidate driver genes of smoking-related plaque alterations. Due to low representation of other subgroups (*e.g.*, diabetics, *n* = 2), further studies focusing on single risk factors are warranted. No significant associations were observed between plaque morphology and patient symptoms or other clinical variables, likely due to limited statistical power.

Our transcriptomic landscape analysis highlights a key role for ECs in the advanced and vulnerable stages of atherosclerotic plaques. ECs have long been attributed to drive the angiogenic process by microvascularizing human lesions. The recruitment and accumulation of inflammatory cells, along with plaque hypoxia, stimulates the expansion of existing *vasa vasorum* and formation of permeable neovessels[70]. As these neovessels within plaques are due to inadequate pericyte coverage and insufficient ECM support often structurally defective, they deem the lesion to become more prone to intraplaque hemorrhage, plaque instability, and consequently acute ischemia[71].

VSMCs represented the most abundant cell subtype in our plaques based on sc-/bulkRNA-Seq deconvolution. Single cell profiling identified three distinct subclusters - contractile, modulated and fibroblast-like. Spatial transcriptomics further revealed a fourth substate, characterised by inflamed VSMCs co-expressing classical VSMC markers alongside immune-related transcripts. These, suspected disease triggering VSMCs, are likely missed with scRNA-Seq due to tissue digestion and cell processing protocols needed for library preparation. The cellular content of the media layer went beyond the primarily for this location described contractile VSMCs. We observed that cells - in this highly relevant structure for arterial integrity - undergo phenotypic switching before migrating towards the plaque core.

We further identified distinct spatial expression patterns of macrophage subpopulations in the FC and NC. *C1Q*+ and *HMOX1*+ cells largely colocalise; however, neither spatially overlapped with *TREM2*hi cells. In a subset of plaques *C1Q*+ and *HMOX1*+ reside close to the lumen, while *TREM2*hi macrophages border and harbour the NC. Interestingly, in a control sample, all three substates appear subluminally, without *TREM2*hi cells separating. We hypothesise a gradual transdifferentiation from *HMOX1*+ macrophages via an intermediate phenotype (Mac_TREM2hi_CCL18hi) that reduces *CCL18* expression while increasing *SPP1, MMP9, MMP12*. Finally, these cells become *TREM2*hi, probably when reaching the crucial shoulder region of the FC. This sequence of cells and associated DEGs seem highly orchestrated, based on the repetitive layering and spatial correlation patterns of *HMOX1*+ macrophages with *TREM2*hi. Xenium RNA velocity analysis corroborated this by revealing a transcriptional trajectory over time from *C1Q*+ subtype towards *TREM2*hi via *HMOX1* + , with the *HMOX1* + → *TREM2*hi transition also supported by our scRNA-seq data, as previously reported[34]. Although intriguing, these findings require further experimental validation.

In conclusion, our study advances understanding of the cellular and molecular heterogeneity of human atherosclerosis and carotid artery disease. Using paired early and advanced lesions from the same patient, we identified a high number of DEGs in bulk RNA-Seq, and consequently, applied the same design to single cell and spatial transcriptomics. This integrative approach uncovered key mediators and pathways involved in plaque vulnerability, despite the substantial heterogeneity found in human atherosclerosis. Specific cell subsets and their substates appear consistently linked to plaque progression and destabilization. Future studies on the induction and therapeutic targeting of cellular plasticity may yield novel markers and interventions for plaque vulnerability, together with digital pathology approaches and enhanced vascular imaging modalities.

## Methods

### Human sample collection

Human samples were obtained from patients diagnosed with carotid stenosis and sourced from the Munich Vascular Biobank. The collection process adhered to the ethical principles of the Declaration of Helsinki and was approved by the local ethics committee at the Technical University of Munich (approval numbers: 2799/10 and 2023-297). Informed written consent was obtained from all participants, and clinical data were retrieved from electronic patient records.

Plaque samples were collected from patients undergoing carotid endarterectomy (CEA), the surgical removal of plaque buildup in carotid arteries. Immediately after removal from the intraoperative situs, plaque samples were transferred to pre-chilled PBS (Merck, Germany) for transportation.

The carotid plaques are then cut into several pieces. Specifically, for this study, we selected the most advanced (i.e., most unstable) part of the plaque and whenever possible also the least diseased part of the plaque for analysis (Supplementary Fig. 3). This unique setup allows for statistically powerful paired analyses of two pieces of the same tissue from the same individual patient.

### Histological analysis

Tissue samples were fixed in 4% paraformaldehyde (PFA) for 24 h and subsequently decalcified for five days using Entkalker soft SOLVA-GREEN® (Carl ROTH, Karlsruhe, Germany). Following decalcification, the specimens were embedded in paraffin and stored as formalin-fixed paraffin-embedded (FFPE) blocks. Paraffin-embedded sections (2 µm thick) were mounted on glass slides (Menzel SuperFrost, 76 × 26 × 1 mm, Fisher Scientific, Schwerte, Germany).

Hematoxylin-eosin (HE) staining (ethanolic eosin Y solution, Mayer's acidic hemalum solution, Waldeck, Münster, Germany) was performed according to the manufacturer's protocols. Slides were mounted using Pertex (Histolab products, Askim, Sweden) as the mounting medium, with glass coverslips (24 × 50 mm, Engelbrecht, Edermünde, Germany). All histological sections were scanned using the Aperio AT2 (Leica, Wetzlar, Germany), and slide analysis was conducted using the open-source software[72]. The histological classification of carotid plaques was performed following the guidelines established by the American Heart Association (AHA[73–76]) and the Redgrave & Rothwell criteria (Oxford Plaque Study)[18] for plaque stability.

### Total RNA sequencing of atherosclerotic plaques

Tissue samples for Bulk RNA Sequencing were transferred to chilled RNALater (Thermo Scientific, Germany), and stored at -80 °C awaiting further processing. Total (bulk) RNA was extracted from human carotid artery specimens provided by the Munich Vascular Biobank and generated libraries that were sequenced (RNA-seq) via the Illumina NovaSeq platform (NovaSeq6000, Illumina, San Diego, USA). RNA was purified using poly-T-oligo-attached magnetic beads and the TruSeq stranded total RNA kit and TruSeq stranded mRNA kit (Illumina, San Diego, USA) were used to prepare the short-read sequencing libraries. RNA and library integrity was assessed via Qubit (Thermo Fisher, USA), as well as real-time PCR for quantification and tape station (Agilent, USA) for size distribution detection.

Stranded RNA-Seq was conducted using the Illumina NovaSeq 6000 platform, with 101 base pair (bp) paired-end reads. Sequencing

depth ranged from 20 million to 140 million reads per sample, depending on RNA quality. The RNA-Seq reads were demultiplexed and aligned to the hg19 genome assembly (UCSC Genome Browser build) using STAR (v2.7.0a). Read quantification was performed with HTSeq-count (v0.6.0) to determine the number of reads mapping to annotated genes. Genes with an FPKM (Fragments Per Kilobase of transcript per Million mapped reads) below the 95th percentile of 1 were filtered out as insufficiently expressed.

After filtering, 14,572 genes were detected across a total of 297 samples derived from 252 patients. The dataset comprises transcriptomes from 232 advanced plaques and 65 early lesion controls. Among the plaques, 97 were classified as stable and 126 as unstable, with 9 remaining unclassified based on our histological analysis. For 42 patients, paired samples of plaque and control lesions were available, allowing for paired differential expression analysis.

Batch effects correction was performed using comBat-seq, adjusting for the influence of the "batch" variable. DESeq2 (v1.34.0) was employed for normalization and differential gene expression (DEG) analysis. DESeq2 normalized read counts by estimating size factors to adjust for differences in sequencing depth across samples. A design formula was constructed to model the count data using negative binomial generalized linear models (GLMs). The formula incorporated age, gender, and disease status as covariates. For paired analysis, individual IDs and disease status were included to account for within-sample correlations. The Wald test was used to evaluate significant differences in gene expression between patients and controls, while controlling for the effects of covariates. DEGs were identified using the results function in DESeq2, with Benjamini-Hochberg-adjusted p-values (false discovery rate, FDR) to account for multiple testing. Genes with an FDR < 0.05 and |$\log_2$ fold change > 1| were considered statistically significant. For the smoker - non-smoker comparison, differential expression was performed with pyDESeq2 (DESeq2 framework) using a negative binomial GLM with design ~ Smoking_ever. The analysis was restricted to plaque samples only. Smokers and former smokers were collapsed into a single 'Smokers' group and contrasted against 'No smokers' (reference). No additional covariates (age, gender) were included. Wald statistics were used and two-sided p-values were adjusted using the Benjamini-Hochberg procedure. Statistical significance was defined as FDR < 0.05 and |$\log_2$ fold change| ≥ 0.5. The full results for all assessed genes are provided in Supplementary Data 4, while Supplementary Data 5 contains the subset of genes also included in our Xenium panels.

### Single cell RNA sequencing of human carotid plaques

For single cell RNA sequencing of human carotid arteries, samples were harvested during CEA in our Department for Vascular and Endovascular Surgery. In total 17 plaques were minced and digested using the Multi Tissue Dissociation Kit 2 (Miltenyi Biotech, 130-110-203) and the 37 C_Mulit_G program (GentleMACS Dissociator; Miltenyi Biotech, 130-093-235) according to the manufacturer's instructions. The cell suspension was cleaned up and finally resuspended in PBS + 0,04% BSA. Cells were loaded into a 10xGenomics microfluidics Chip G and encapsulated with barcoded oligo-dT-containing gel beads. Gel-Beads-in-emulsion (GEM) cleanup, cDNA amplification and 3´Gene-eExpression Library Construction was performed according to the manu-facturer´s instructions (CG000204 Rev D). Libraries from individual patient samples 1-10 (batch 1) were multiplexed into one lane before sequencing on an Illumina NovaSeq6000 instrument. One whole flow-cell was occupied and used for sequencing. This data has been deposited in NCBI's Gene Expression Omnibus[77] and is accessible through GEO Series accession number GSE247238. Libraries from individual patient samples 11-17 (batch 2, GSE294291) were loaded on an Illumina NovaSeq with 2 × 150 paired-end kits and sequenced with Novogene (Martinsried, Germany). An additional scRNA seq dataset of human plaques was downloaded from GEO repository (GSE159677)[51].

**Alignment.** For the alignment a custom reference genome was built using the Gencode GRCh38-p14-V44 primary assembly FASTA file and matching Gencode primary assembly annotation.gtf file. For this we used the *mkref* function of 10x's cellranger version 7.1.0, including the following biotypes: "protein_coding", "lncRNA" "antisense", "IG_LV_gene", "IG_V_gene", "IG_V_pseudogene", "IG_D_gene", "IG_J_gene", "IG_J_pseudogene", "IG_C_gene", "IG_C_pseudogene", "TR_V_gene",,"TR_V_pseudogene", "TR_D_gene", "TR_J_gene", "TR_J_pseudogene", "TR_C_gene". Alignment of the reads was performed with cellranger's *count* function with default parameters, which included intronic reads.

**Preprocessing steps.** Background RNA correction was performed with cellbender (version 0.3.0)[78], droplets with a cell probability larger than 0.5 predicted by cellbender were retained for analysis. Single-cell analysis was performed with scanpy, version 1.9.8[79]. Cells with high erythrocyte marker gene expression (*HBB* (Hemoglobin Subunit Beta), *HBA1* (Hemoglobin Subunit Alpha 1), *HBA2* (Hemoglobin Subunit Alpha 2)) were removed before running doublet detection with scDblFinder (version 1.12.0)[80]. Quality control entailed retaining cells with following parameters: gene count per cell between 200 and 55 000, number of genes per cell between 200 and 8000, scDblFinder doublet score lower or equal than 0.5, mitochondrial read ratio lower or equal than 0.2. Genes expressed in at least 5 cells were kept in the final dataset, log-normalisation was carried out by applying the "acosh" transformation to the cellbender corrected count values.

**Batch correction and cell type annotation.** We planned to annotate cell types manually in a similar fashion as previously described in the case of the Xenium data: detecting clusters after batch correction and annotating them based on their differentially expressed genes. For this both linear (Harmony, version 1.1.0)[81], and nonlinear autoencoder based (scVI[82] with scvi-tools 1.1.2; scANVI with scarches version 0.6.1[83]) batch correction methods were applied on our dataset, but none of them were able to completely remove all of them, eventually confounding the further steps of clustering and DGE analysis. Therefore we adopted a different approach: we decided to integrate our data with a larger plaque scRNA-seq dataset of a similar experimental setup to ours and transfer its cell type labels, hypothesising that cell types were also similar across the two datasets. We selected the plaque scRNA-seq dataset published by Alsaigh et al[51], which had a similar experimental setup (matched atherosclerotic core - proximal adjacent samples from same patients) to ours (matched diseased - control samples from same patients). However, as cell type annotations were not published alongside the dataset, we had to annotate it manually. The dataset is accessible under the accession name GSE159677 in the Gene Expression Omnibus (GEO) repository (https://www.ncbi.nlm.nih.gov/geo/query/acc.cgi?acc=GSE159677).

**Cell type annotation - GSE159677 dataset.** We downloaded the filtered count matrix from the GEO repository, ran scDblFinder for doublet detection and kept cells with following parameters: gene count per cell between 400 and 55 000, number of genes per cell between 350 and 5000, scDblFinder doublet score lower or equal than 0.5, mitochondrial read ratio lower or equal than 0.1. Log-normalisation again occurred by "acosh" transformation. Batch correction was carried out with scVI[82], using the top 4000 highly variable genes, correcting for "patient" as batch effect.

Manual annotation of the GSE159677 dataset was performed in a similar fashion to the Xenium dataset: coarse Leiden-clustering and differential gene expression (DGE) between the coarse clusters with Wilcoxon-method provided us with marker genes of the clusters, which were compared to marker genes collected from literature to determine high-level cell types. To detect cell subtypes, we performed additional Leiden clustering and differential gene expression (DGE)

analysis within the high-level cell types, using a pseudobulk approach with DESeq2 (version 1.38.3)[84], controlling for "patient" as a factor and creating pseudobulk counts for each cluster by aggregating cells coming from the same sample. DESeq2 was applied with default parameters, returning two-sided Wald test p-values, which were then corrected for multiple testing using the Benjamini-Hochberg method and reported as false discovery rates (FDR). Clusters were annotated based on their marker genes and cell subtype marker genes from literature.

**Cell type annotation - integration and cell type label transfer from GSE159677 with subsequent refinement.** For this we chose scPoli (scarches version 0.6.1)[85], an autoencoder based model designed to regress out batch effects across datasets while retaining biological information of cell types present. First a reference model was created by training scPoli on the GSE159677 data, using our cell subtype level annotations as cell type labels and controlling for factors: "patient","-batch","condition","sequencing method". Then, we further trained the reference model on our data controlling for the same factors, resulting in an integrated latent representation of the two datasets. Of note: based on the methods section of Alsaigh et. al's publication we deemed their samples with condition label "necrotic core" equal to our "plaque", but we were not sure if "proximal adjacent" meant a proximal healthy or less affected diseased vascular segment, therefore we used 3 condition labels for the integration: "healthy","plaque", and "adjacent". Cell type labels were transferred to our dataset by the fully trained scPoli model along with a prediction uncertainty score for each cell ranging from 0 to 1, showing groups of cell types transferred with both low and high uncertainty levels. Inspecting the integrated latent representation revealed cells with high prediction uncertainty scores to display lower integration levels with the GSE159677 dataset. This lack of integration for certain cell groups may stem from technical factors, such as unknown batch effects or scPoli's model architecture, or from limited biological variability in the cell states of the GSE159677 dataset, given that, despite the high cell count, the data is derived from only three patients. We performed the integration with two other models as well (scVI and scANVI), but the integrated latent representations showed worse integration compared to scPoli. However, the subset of the scPoli latent representation containing only our cells demonstrated a more effective batch effect removal than the original single-dataset attempts with Harmony, scVI and scANVI, therefore we derived the final cell types by performing the same manual annotation pipeline on our data as described for the GSE159677 data, using the scPoli representation as batch corrected latent representation. DESeq2 was run with default settings as before (two-sided Wald test with Benjamini-Hochberg correction). DGE results across high-level cell types and low-level cell types are contained in Supplementary Data 6 and Supplementary Data 7 respectively.

**Differential gene expression analysis.** For differential gene expression (DGE) analysis we again utilised the pseudobulk approach: we created pseudobulk samples by aggregating cells of the same sample within low-level cell types and subsequently carried out DGE analysis for multiple condition variables using DESeq2. We included cells from all sample conditions (plaque and control) in the DGE analysis. The effects of "patient", "batch" and "condition" were controlled for in DESeq2, except for comparing across conditions, where only "patient" and "batch" were included as covariates. DESeq2 was again run with default parameters, producing two-sided Wald test p-values that were subsequently corrected for multiple testing using the Benjamini-Hochberg method. DGE results can be found in Supplementary Data 8.

**RNA velocity analysis of macrophages.** As our spatial transcriptomic dataset revealed intriguing spatial and expressional patterns in macrophage subtypes, we set out to validate this finding in our scRNA-seq

data. To do this, we investigated the transcriptional trajectories of macrophage subtypes by performing RNA velocity analysis using the recently published TFvelo[49] method, which utilises transcriptional regulation to calculate RNA velocity trajectories. We first subsetted the data to only contain macrophages (2,805 cells) and ran the pre-processing step with parameters recommended by the authors, using the scPoli-UMAP representation as a low-dimensional representation and acosh normalised counts. For the calculation of velocity stream plots, we only included genes with a larger latent time - pseudotime Spearman correlation of 0.4. Pseudotime and velocity streamplots (Supplementary Fig. 40A,B) were created using the scPoli-UMAP representation. In addition to the cellular-level trajectory information provided by the streamplots, we also aimed to assess trajectories at the cell type level. To this end, we applied the PAGA (partition-based graph abstraction) algorithm[86] using the inferred TFvelo pseudotime as priors and the macrophage low-level cell types as node groups. The resulting graph, displaying velocity-directed edges between cell types, is shown in Supplementary Fig. 40C. We also plotted the TFvelo pseudotime distribution of the three macrophage subtypes (Mac_C1Q, Mac_HMOX1, and Mac_TREM2hi; Supplementary Fig. 40D), which showed ordered spatial distribution patterns in the Xenium data, to better visualize their relative ordering along the TFvelo pseudotime axis.

To complement TFvelo with an orthogonal RNA velocity approach, we also applied a splicing count-based method to infer cell trajectories. Spliced and unspliced expression counts were obtained using Velocyto (version 0.17)[50] on the.bam files and filtered barcodes generated during alignment with Cell Ranger's *count* function.

For the splicing count-based analysis, we used scVelo (version 0.3.2)[87]. We first created a subset of our data containing only macrophage cells (1,837 cells) and calculated first- and second-order moments with *scvelo.pp.moments()* using the scPoli representation, with the number of neighbours set to 30. Next, we estimated velocity vectors with scVelo's dynamical model. Inferred pseudotime and velocity stream plots were generated using the scPoli-UMAP representation (Supplementary Fig. 41A, B). A PAGA graph based on pseudotime inferred by scVelo is shown in Supplementary Fig. 41C, together with the scVelo pseudotime distribution across the three macrophage subtypes (Supplementary Fig. 41D).

## Spatial transcriptomics of human carotid plaques

**Xenium sample preparation and run.** Tissue sectioning and sample preprocessing were performed according to the manufacturer's protocol. Briefly, 5 μm FFPE sections were mounted on Xenium slides and dried to optimize adherence. Subsequently, the samples went through a 3-day protocol including multiple steps such as deparaffinization, decrosslinking, probe hybridization, ligation, and amplification. The pre-processed samples were analyzed using the Xenium Analyzer with the instrument software version 1.5.1.2 or 1.7.6.0 and analysis version 1.5.0.3 or 1.7.1.0. Subsequently, H&E staining of the tissue sections was performed according to the manufacturer's protocol for post-Xenium stains.

**Panel Information.** Targeted spatial transcriptomics (Xenium, 10X Genomics) was run with two fully custom-made panels: "9MHNNK - Fully custom hArtery panel for TUM" with 314 individual genes (panel 1) and "PXD8AD - Fully custom hArtery panel for TUM" with 345 individual genes (panel 2). Genes were chosen based on cellular and disease associated pathways (apoptosis, DNA damage, efferocytosis, CAD associated genes, cell cycle, inflammasome, macrophage biology, senescence, unfolded protein response) as well as DEGs, marker genes and marker proteins from plaque[27,58,88] and GWAS[89] datasets, including our own scSeq dataset (GSE247238, used in[26,55,90]) (Supplementary Data 1). The panels consist of a core list present in both panels (comprising 111 central genes for cell type annotations and key processes) as

well as individual genes for the panels, resulting in a cell-defining (panel 1) and disease associated (panel 2) panel. In total 548 unique genes have been sequenced with both panels.

Sequencing with panel 1 and panel 2 genes was performed on consecutive FFPE sections for each sample, thereby introducing a 5 μm shift in the morphology of the mounted Xenium slides across the panels.

**Cell segmentation and transcript assignment with Baysor.** Xenium Onboard Analyzer (XOA) outputs the 3-dimensional coordinates of the transcripts and border coordinates of nuclei by performing nuclear segmentation using the high-definition DAPI-stained tissue slides. Furthermore, a standard cell segmentation was also provided, this involved the expansion of the nuclear segmentation borders by a constant number and assigned within border transcripts to the respective cells. A benchmarking paper by Salas et al[38]. showed that this approach could lead to misassignment of transcripts from neighbouring cells, and recommended best practices for cell segmentation and transcript assignment of Xenium data. To ensure the best possible cell segmentation, we have followed their best practices pipeline and performed our own cell segmentation and transcript assignment with Baysor, version 0.6.2[37]. Baysor uses spatial transcriptomic data to infer cell borders, optionally taking image data of nuclear or cell staining into account in addition to the transcript coordinates.

We considered four data combinations as input to Baysor: spatial transcriptomic data alone, spatial transcriptomic data paired with 10x XOA nuclear segmentation masks, and spatial transcriptomic data with segmentation masks created from the DAPI (4',6-Diamidino-2-phenyl-lindol) stained slides with either the ''Cytoplasm' or the 'Nucleus' model of Cellpose (version 2.2.2)[91]. The most sensitive parameter of Baysor is "scale", which corresponds to the approximate radius of the cells in the sample, along with "scale-std", which sets the standard deviation of cell radius across cells. As plaques normally contain cell types with different shapes and sizes (i.e spindle-like vascular smooth muscle cells[92], or round macrophages[93,94]), we have set the "scale-std" parameter to 50% instead of the default 25%, and ran Baysor with 3 increasing "scale" parameters (5,10,15 μm) for each of the four input combinations, thereby performing cell segmentations with 12 different data input - scale combinations for each sample. This way we were able to choose the best performing input data - scale parameter combination for each sample later, as there were significant technical and biological variations between samples. Of note, Baysor provides the possibility to estimate the "scale" parameter for each sample, however as we saw large variation in the estimated "scale" parameter across the samples, we decided to use the 3 fix values (5,10,15μm) approach. As per 10x Genomics recommendation (https://www.10xgenomics.com/support/software/xenium-onboard-analysis/latest/analysis/xoa-output-archive-data), we filtered out low-quality transcript reads (Q-Score<20) before running cell segmentation.

Baysor output cell boundary and cell centre coordinates of the identified cells, as well as the transcripts assigned to these cells along with confidence metrics for the cell assignments. We kept transcripts with assignment confidence > 0.5, a metric that reflected Baysor's confidence in assigning the transcript to the correct cell. A count matrix for each sample was constructed by summarising the transcripts for each cell in the sample and saved as an AnnData object, with the rows corresponding to cells and columns to transcripts. For the subsequent processing steps we used scanpy[79].

**Selection of best cell segmentation per sample for each "scale" parameter.** We relied on Baysor's cell assignment confidence values to select the best cell segmentation for each sample. Within a sample, we calculated the average transcript assignment confidence of the cells for each data input combination and plotted its distribution via a

kernel density estimation (KDE) plot. The majority of cells had an average transcript assignment confidence in the range of 0.75 - 1, therefore we calculated the area under the curve of the KDE curves in that range ($AUC_{0.75}$) and selected the data input combination with the highest $AUC_{0.75}$ value as the best cell segmentation for that sample. However, as the "scale" parameter had a large effect on the number and size of the segmented cells, we decided to perform the selection for input combinations within each "scale" parameter, resulting in one best cell segmentation per "scale" parameter for each sample. The next steps (quality control and normalisation) were performed on the AnnData objects derived from the best cell segmentations, i.e., 3 AnnData objects per sample, one for each "scale" parameter.

**Quality control, normalisation and batch correction.** We filtered cells with: less than 10 non-zero-count genes per cell; cells with lower average assignment confidence than 0.75; cells with lower "max_cluster_frac" value than 0.9, as low "max_cluster_frac" values often belong to doublets; cells with lower "lifespan" value than 50 (lower "lifespan" values correspond to noise). Normalisation was performed by dividing the individual cell counts by their individual cell area, similar to Salas et al's benchmarking paper[38]. After filtering, we have merged the data of the samples per "scale" parameter, resulting in three composite AnnData objects. As the samples were screened with two different panels of genes, we obtained three AnnData objects per gene panel respectively, totalling in six composite AnnData objects.

Similar to single-cell RNA-sequencing data, the cell counts followed a negative binomial distribution, therefore we have deemed it appropriate to use scVI for batch correction, a method developed for scRNA-seq[82]. We corrected for the effect of "patient", as this seemed to cause the main batch effects across samples based on the UMAPs created with raw expression counts. Batch correction was performed for each composite AnnData object separately.

**Cell type and cell substate annotation.** Initially data integration and cell type label transferring from an annotated plaque scRNA-seq dataset (Tabula Sapiens, vascular dataset[95]) were attempted with various methods (scVI[82], scANVI[83], scPoli[85]). However, as generally the number of genes and transcript counts per cell were substantially lower in our data compared to the Tabula Sapiens dataset, none of the methods were successful at integrating them. Instead, we performed manual cell type annotation by clustering our data and annotating the clusters based on their differentially expressed genes.

First, to detect high-level cell types reflecting main cell identities, we performed coarse Leiden-clustering and performed differential gene expression (DGE) analysis in a "one *vs.* all" manner using the Wilcoxon-method as implemented in scanpy (two-sided Wilcoxon rank-sum test, with Benjamini-Hochberg correction). For the detection of low-level cell types, corresponding to previously known cell subtypes, further Leiden-clustering and DGE analysis within high-level cell types was conducted. To account for the biological and technical variability across samples and patients[96], we employed a pseudobulk approach with DESeq2[84] as the DGE method in the latter step. Within each high-level cell type subcluster, pseudobulk samples were created by aggregating cells per sample. On top of "patient" we also controlled for "condition", as some patients had samples from both conditions, "control" and "plaque". Low-level cell types were determined by comparing the cluster marker genes (two-sided Wald test, with Benjamini-Hochberg correction) and cell subtype marker genes, collected from literature[34,88,97,98]. Of note, despite not having *C1Q* or *HMOX1* in our gene panels, we assigned the labels Mac_C1Q and Mac_HMOX1 based on overlapping genes in our panels and the top 20 highly expressed genes identified for these cell types in the study by Dib et al[34]. Differential gene expression analysis was performed using both approaches, the two-sided Wilcoxon rank-sum test and the two-sided Wald test with DESeq2 on pseudobulk counts (controlling for

"patient" and "condition"), with Benjamini-Hochberg correction applied across final high-level cell types (Supplementary Data 9, Supplementary Data 10) and low-level cell types (Supplementary Data 11, Supplementary Data 12).

Interestingly, during the high-level cell type annotation we consistently detected a cell cluster, which was hallmarked by low transcript counts per cell (Supplementary Fig. 46D, Supplementary Fig. 47D), high expression of multiple immune cell-related genes, along with genes of other major cell types. Based on these features we decided to annotate the cell cluster as "Inflammatory mixed cells". We hypothesize that these cells could reflect cell fragments, whose cytoplasm was only partly contained in the plane of the histology sample, although they can also be a result of incorrect cell segmentation. Although it is possible that these cells represent doublets, we consider this unlikely. Potential doublets arising from cell segmentation were excluded using Baysor's "max_cluster_frac" parameter, as recommended by the authors, and the low transcript counts observed in these cells further argue against the doublet interpretation.

We further refined the low-level cell types into low-level cell substates by Leiden-clustering within low-level cell types and performing DGE analysis using the previously described pseudobulk approach with DESeq2, controlling for factors "patient" and "condition" again. Annotation of the low-level cell substates was carried out manually by inspecting the top differentially expressed genes within low-level cell types.

Cell type and cell substate annotation was performed separately for each composite AnnData object.

**Annotation of sample subregions.** The annotation of sample subregions is a histological analysis, using established Oxford Plaque Studies[18–20] and AHA criteria[73–76] and was performed by an experienced cardiovascular pathologist. The borders of the annotated subregions were initially determined on the HE stained slides by a pathology expert, then transferred to the DAPI-stained slides manually by drawing the determined subregion borders by hand in Xenium Explorer. Border coordinates were extracted from Xenium Explorer and converted to polygons, which then were used to assign individual cells to their respective subregions and to calculate the subregion areas. Normalised sample subregion areas were calculated as the percentage of the whole sample area. Some of the samples tore or partly disintegrated during the preparation process (control samples of patient 1 & 2, plaque samples of patient 3, 4, 7, 8), in these cases only an approximation of the damaged region's area was possible. All samples with regions annotated are displayed in Supplementary PDF 5.

**Selection of final dataset based on Baysor's "scale" parameter.** The previous steps resulted in 3×2 annotated datasets: one dataset per each of the three Baysor "scale" parameters, for both gene panels. In the following we describe the insights regarding the impact of the scale parameter on the cell segmentation. Comparisons in Supplementary Fig. 48A,B and Supplementary Fig. 49A,B were carried out by applying paired two-sided t-tests with Benjamini-Hochberg correction, whereas for the comparisons in Supplementary Fig. 48C, D and Supplementary Fig. 49C, D unpaired two-sided t-tests were used.

- "Scale" of 5μm: Segmentation with scale parameter of 5μm resulted in a higher number of cells (Supplementary Fig. 48E, Supplementary Fig. 49E) with lower transcript and count per cell (Supplementary Fig. 46, Supplementary Fig. 47), especially for vascular smooth muscle cells (VSMCs), but also a significantly higher percentage of transcripts were considered as noise compared to segmentations with the larger scale parameters (Supplementary Fig. 48A, Supplementary Fig. 49A. We failed to detect VSMC-modul in both panels (Supplementary Fig. 50B, Supplementary Fig. 51B), in the panel 2 dataset cell subtypes Mac_C1Q and Mac_S100A8+ clustered together in a mixed cluster, whereas Mac_HMOX1 could not be annotated at all

(Supplementary Fig. 51B). Plotting the segmented individual cells with the largest cell sizes showed mostly one cell nucleus per cell and no obvious duplicates (Supplementary Fig. 52, Supplementary Fig. 53).

- "Scale" of 15μm: The lowest percentage of transcripts were deemed as background noise compared to the other two scale parameters (Supplementary Fig. 48A, Supplementary Fig. 49A), along with transcript and gene counts per cell being reaching the highest range (Supplementary Fig. 46, Supplementary Fig. 47). The same low-level cell types were found across both panels, amongst cell subtype VSMC_modul, which was missing with a scale of 5 (Supplementary Fig. 50F, Supplementary Fig. 51F). However, the cell size distribution of some VSMCs (VSMC_fb-like, VSMC_contr, VSMC_inflamed) and Macrophage subtypes (Mac_TREM2hi, Mac_HMOX1, Mac_C1Q) segmented with scale parameter 15 showed unrealistically large cell sizes (>2000μm$^2$) for the quantile 75% and above (Supplementary Fig. 48F, Supplementary Fig. 49F), along with higher median values compared to smaller scale parameters. Individual assessment of these large cells clearly showed multiple cells segmented together as one, especially for VSMC and for macrophage subtypes as well (Supplementary Fig. 54, Supplementary Fig. 55).

- "Scale" of 10 μm: Noise percentage was in between the noise percentage of scales 5μm and 15μm (Supplementary Fig. 48A, Supplementary Fig. 49A). We were able to detect cell subtype VSMC_modul, and the annotated low-level cell types were consistent with the cell subtypes found with scale parameter 15, except for Mac_ITGAXhi, which was not detected in panel 2 (Supplementary Fig. 50D, Supplementary Fig. 51D). The VSMC and macrophage cell subtypes having extremely large cells with scale parameter 15 also showed higher cell size values in the quantile 95% and above, however most of these cells were smaller than 1500 μm$^2$ (Supplementary Fig. 48F, Supplementary Fig. 49F). Inspection of individual VSMC subtype cells with the highest 5% cell sizes revealed that they were often multiple spindle shaped cells, segmented together in one round cell. Among the largest macrophages however, along with duplicates we saw an increased number of large, round macrophages, that were correctly segmented by Baysor (Supplementary Fig. 56, Supplementary Fig. 57).

Although significant differences presented in the percentage of transcripts labelled as noise across the three parameters (Supplementary Fig. 48A, Supplementary Fig. 49A), we did not observe strong significant differences in the percentage of transcripts surviving filtering (Supplementary Fig. 48B, Supplementary Fig. 49B). Taken together, the scale parameter of 5μm was more successful in segmenting the spindle shaped VSMC cells but seemed to fail at capturing the larger macrophage subtypes, whereas scale 15μm tended to result in 2 or more cells segmented as one for specific macrophage and VSMC subtypes with larger sizes. Therefore, we decided to select the data segmented with the scale parameter of 10μm, which constituted a middle ground between segmenting larger round macrophages and elongated, spindle shaped VSMCs.

Since Baysor can also estimate the "scale" parameter for each sample, individual values can be used for segmenting multiple samples. We recommend checking for outliers in the estimated values and considering the trade-off between Macrophage and VSMC cells based on the research question.

HE-stained slides of individual samples, along with the spatial distribution of cells from the final dataset, coloured by low-level cell types and subregional assignments, are presented in high quality format in Supplementary PDF 1 & 2.

**Assessment of unassigned noise transcript levels across conditions and subregions.** We also investigated the difference we observed between plaques and control samples in the level of noise transcripts that were unassigned to cells by Baysor (Supplementary Fig. 48C, D, Supplementary Fig. 49C, D), utilising the subregional annotation polygons. We hypothesized that the presence of necrotic cores may

explain the higher percentage of noise in plaques, as they generally take up large areas of plaques (Supplementary Fig. 58A), and contain vast cell-poor, cell debris filled domains, which may represent a challenge for cell segmentation. NCs contained on average the most transcripts in plaques, both assigned to cells as well as classified as noise (Supplementary Fig. 58B, Supplementary Fig. 58C), also displaying the highest relative contribution to the overall noise detected among the plaque subregions (Supplementary Fig. 58D, blue dots). This effect emerges as a consequence of their large size, possibly contributing to the difference in noise percentages across plaques and control samples seen earlier. Seeing this large impact of the necrotic core on noise, we were wondering, if this is only caused by the NC hoarding most of the transcripts, or does Baysor assign relatively more transcripts as noise in the NC as compared to other subregions? To assess this, we calculated the ratio of noise transcripts within each subregion and found NC having significantly higher relative percentage of noise compared to fibrous cap and media subregions, although not to intima (Supplementary Fig. 58E, blue dots). This suggests that, in addition to its size, the composition of the NC may also contribute to noise. We also compared the relative noise of cell-rich subregions intima and media between conditions and found control samples displaying lower noise percentages in their media compared to plaques (Supplementary Fig. 58E). All comparisons presented in Supplementary Fig. 58E (within plaque and across plaque-control subregions) were performed by applying unpaired two-sided Mann-Whitney-U tests, and corrected for multiple testing with the Benjamini-Hochberg method, yielding false discovery rates (FDR). These observed differences in the relative noise ratios within subregions may be caused by unknown technical factors in the pipeline (i.e., Xenium reader, Baysor), or could also arise from plaques generally containing more cell debris, even in regions with no obvious necrosis, like the media. Analysis with a larger sample size could further confirm these findings.

**Differential gene expression analysis.** For differential gene expression (DGE) analysis we again utilised the pseudobulk approach: we created pseudobulk samples by aggregating cells of the same sample within low-level cell types and subsequently carried out DGE analysis for multiple condition variables using DESeq2, always controlling for "patient" as factor. Differential gene expression was performed with DESeq2's default parameters, yielding raw two-sided p-values from the Wald test, which were then corrected for multiple testing using the Benjamini-Hochberg method and reported as false discovery rates (FDR).

Cells originating from intraluminal thrombi were excluded from DGE analysis. To uncomplicate our analysis, we used subsets of data according to our scientific questions: for the comparison "*Plaque vs. Control*" only samples from Patients 1-4 were included as they had matching "Control" and "Plaque" samples, all other comparisons were performed across the 12 "Plaque" samples (Patient 1-12). For Patients 1-4 with available control and plaque samples, we also performed DGE analysis including their media-derived cells only, due to the media being present and proposed to be relatively similar in both control and plaque samples. DGE results can be found in Supplementary Data 2, Supplementary Data 3 and Supplementary Data 13 and Supplementary Data 14.

**Clustering of samples - morphological clusters.** First, we calculated the cell fraction percentage of the different cell type substates normalised within the sample. This enabled us to capture finer differences in cellular composition across all plaques. Next, we performed hierarchical clustering of the samples based on these normalized cell type fractions, using Euclidean distance and Ward's-linkage method. This is an unsupervised clustering method, and does not rely on a predefined statistical test for determining group assignments. Cells originating from intraluminal thrombi were excluded from both the cell fraction calculations and clustering steps.

The final number of clusters (4) was selected based on cluster robustness (Fig. 5) and further supported by visual inspection of the samples (Fig. 6). Based on the dendrogram structure shown at the top of the clustermap in Fig. 5, selecting four clusters provided a biologically meaningful and robust separation between groups of samples, while at the same time, samples within each cluster displayed consistent patterns in their spatial morphology (Fig. 6) as described in the "Morphological plaque clustering" subsection in Results. These considerations together guided our choice of four clusters.

We only included plaque samples in the clustering, as we saw more significant heterogeneity among these samples compared to the controls. We used the cell type substates derived from panel 1 genes, as this panel resulted in ~ 20% more cells than panel 2, possibly giving us a higher resolution look into the plaque compositions.

We also performed hierarchical clustering of samples containing intraluminal thrombi, based on the low-level substate cell fractions within cells detected in each thrombus (Supplementary Fig. 59). Compared with the overall dataset, thrombi contained fewer cell substates, showed less variation across samples, and were predominantly composed of neutrophils, inflammatory mixed cells, and CD8 effector T-cells. The endothelial cells detected within intraluminal thrombi most likely originated from intimal layer lacerations introduced during sample preparation.

**Correlation analyses.** Associations between region areas and patient metadata (Supplementary Fig. 12, Supplementary Fig. 14–17), cell fraction of low-level cell substate and metadata (Supplementary Fig. 22, Supplementary Fig. 23), as well as correlation between cell segmentation metrics (Supplementary Fig. 9) were assessed to derive possible biological insights and to identify potential confounding factors. For associations between numerical variables involving only plaque samples we fitted a linear regression and report the R-squared value, the raw two-sided p-value of the linear fit and after Benjamini-Hochberg correction for multiple testing, we also report the corrected p-value of the fit as the false discovery rate (FDR) (Supplementary Fig. 12, Supplementary Fig. 14–17, Supplementary Fig. 22, Supplementary Fig. 23). In the case of cell segmentation metric associations, we compared all samples including some plaques and controls coming from the same patients, therefore we fitted a linear mixed model with patients having random intercepts to have an unbiased estimate of the association (Supplementary Fig. 9). Here we report the raw two-sided p-values and the Benjamini-Hochberg corrected p-values as FDR. To investigate the associations between numerical and categorical variables we conducted non-parametric Wilcoxon-tests and report their raw two-sided p-values along with the Benjamini-Hochberg corrected two-sided p-values as FDR (Supplementary Fig. 12, Supplementary Fig. 14-17, Supplementary Fig. 22, Supplementary Fig. 23).

**Cell neighbourhood clustering.** We hypothesized that although the samples show substantial heterogeneity in their morphology, assessing the local neighbourhoods of cells could reveal similar spatial domains across samples. Such domains might reflect shared biological processes or functions, potentially leading to a better understanding of plaque formation. Unlike histological subregions, which represent well-established anatomical compartments of plaques (i.e., lumen, intima, media, fibrous cap, necrotic core), we set out to identify in an unbiased manner patterns in local cell neighbourhoods. We applied unsupervised clustering of these neighbourhoods to generate clusters representing spatial domains that are characterised by similar cellular composition. Because the approach relies on local neighbourhoods and unsupervised clustering, the resulting clusters are independent of predefined subregion boundaries. To do this, for each cell we defined its cell neighbourhood as a circle with a 100 μm radius around it using squidpy's (1.4.1)[99] *spatial_neighbors()* function, and counted the number of each cell type substate within the neighbourhood. This resulted

in a neighbourhood matrix where the rows correspond to all cells across the samples and the columns correspond to all cell type substates in the data. Cell neighbourhood matrices were devised for each panel of genes separately. We settled on 100 μm as our neighbourhood radius because lower radiuses tended to result in cells with 0 neighbours, while higher radius neighbourhoods started to capture the unique composition of the individual samples. In the final step we performed Leiden-clustering of the cell neighbourhood matrices using scanpy's *pp.neighbors()* and *tl.leiden()* functions, resulting in cell neighbourhood clusters (Supplementary Fig. 60, Supplementary Fig. 61). To provide statistical support for the clusters, we performed permutation testing using neighbourhood enrichment score, a metric defined as the total number of unique cell-cell adjacency edges between substate pairs within a cluster. We hypothesised that if our clusters indeed capture distinct spatial patterns, by calculating the neighbourhood enrichment score, we could identify which cell substate pairs are significantly enriched in each other's neighbourhoods within a cluster.

First, we created a background distribution of cell neighbourhood matrices by randomly shuffling cell substate labels 10,000 times. Because samples exhibited large differences in cell substate composition, we performed label shuffling within samples. This was necessary since shuffling across samples would have violated the exchangeability assumption of permutation testing[100] and could have led to inflated p-values. Next, for neighbourhood enrichment analysis, we constructed spatial neighbourhood graphs by connecting cells within a 100 μm radius, with each edge representing a pair of cells within this distance. For each neighbourhood cluster, we restricted the graph to the cells assigned to that cluster and calculated neighbourhood enrichment as the total number of unique undirected adjacency edges connecting substate pairs within the cluster-specific graph. This was again computed for both the observed and randomly shuffled substates, resulting in one enrichment score matrix per cluster for the observed case and 10,000 matrices per cluster for the permutation case. Empirical left and right p-values were calculated using the Monte-Carlo p-value method as described by Phipson et al[101]:

$$p_{left} = \frac{l+1}{N+1} \qquad (1)$$

$$p_{right} = \frac{r+1}{N+1} \qquad (2)$$

where $p_{left}$ and $p_{right}$ are the left and right empirical p-values, $l$ is the number of times enrichment score from the permutation was lower than the enrichment score of the observed case, $r$ is the number of times enrichment score from the permutation exceeded the enrichment score of the observed case, and $N$ is the number of permutations. In addition, Agresti-Coull confidence intervals (CI)[102] of the empirical left and right p-values were also calculated. Two-sided empirical p-values and their Agresti-Coull confidence intervals were calculated using the formula applied in *scipy.stats.permutation_test()* function (https://docs.scipy.org/doc/scipy/reference/generated/scipy.stats.permutation_test.html):

$$p_{two-sided} : min(2 \times p_{left}, 2 \times p_{right}) \qquad (3)$$

$$CI_{two-sided} : min(2 \times CI_{left}, 2 \times CI_{right}) \qquad (4)$$

Raw empirical two-sided p-values were then corrected for multiple testing by applying the Benjamini-Hochberg correction method, and reported as false discovery rate (FDR). Significance was defined as FDR < 0.05. Permutation test results for both panel 1 (Supplementary PDF 6, Supplementary Data 15) and panel 2 (Supplementary PDF 7,

Supplementary Data 15) neighbourhood clusters identified significantly enriched or depleted substate pairs in every cluster compared to the random background. Based on these results we believe our neighbourhood clustering approach was able to identify spatial domains with significantly different compositional patterns, even across multiple samples. High resolution spatial scatterplots of all samples with cells coloured by low-level cell types and neighbourhood clusters are contained in Supplementary PDF 8 for panel 1 genes, and in Supplementary PDF 9 for panel 2 genes. To showcase the macrophage neighbourhoods, we selected plaques from patients 4 and 8 for inclusion in the main figures, as these contained macrophage clusters with the most illustrative spatial patterns. In addition, these samples also contained EC_neovessel neighbourhoods, which are also highlighted in the main text. To illustrate VSMC neighbourhoods, we selected samples with a cell-dense and intact media (plaques from patients 7 and 10, and the control from patient 3) for the main figures.

**Gene expression differences of macrophage substates in neighbourhood clusters 2 and 4.** As the spatial distribution of the four macrophage substates Mac_C1Q, Mac_HMOX1, Mac_TREM2hi_CCL18hi, Mac_TREM2hi within the panel 1 neighbourhood clusters 2 and 4 showed a somewhat ordered pattern from lumen to the necrotic core in specific plaques (Patients 4,7,8,10), we hypothesized that a gradual expression transition may exist in the direction of Mac_HMOX1/ Mac_C1Q → Mac_TREM2hi_CCL18hi → Mac_TREM2hi, as they cell expressions are also similar (Supplementary Fig. 35B). Therefore, we set out to identify genes characterising these substates and also to assess the subtle differences in their expression levels, aiming to derive biological insights. We filtered the data to include cells from the specified plaques (Patients 4,7,8,10) that belong to the three macrophage substates and Panel 1 neighbourhood clusters 2 and 4. To account for the sequencing depth differences across samples, we additionally min-max normalised the area-normalised expression of cells within samples, resulting in expression levels between 0 and 1. Next we averaged the min-max normalised expressions of all genes within cells of each substate, and selected 20 genes with the highest variance in mean expressions. We clustered the genes by the Z-score standardised mean expressions within the substate (Supplementary Fig. 35A), to detect genes with eventual gradients in expression levels across substates, also plotting the gene expressions of individual cells (Fig. 8B). To test statistical significance of gene expression level differences we saw in the previous plots, we performed pairwise differential gene expression (DGE) analysis between the three hypothesized "endpoint" substates Mac_C1Q, Mac_HMOX1 and Mac_TREM2hi in our sub cohort (Supplementary Fig. 35C-E, Supplementary Data 16). DGE was performed in the same manner as described in the previous section "Differential gene expression analysis".

**Spatial correlation of macrophage substates.** To assess the statistical significance of the spatial distribution patterns seen in the three "endpoint" macrophage substates (Mac_C1Q, Mac_HMOX1, Mac_TREM2hi), we computed the spatial correlation between pairs of these substates, using the Local Bivariate Moran's I statistic[103]. For each pair of substates, we first binarized cell identities across all spatial coordinates (1 = presence, 0 = absence), we then constructed a K-nearest neighbours (KNN) spatial weights matrix using Euclidean distances between all cells (k = 30), with row-standardization applied. Local Bivariate Moran's I was calculated for each cell, measuring the association between the presence of one cell type and the spatial distribution of another in its neighbourhood. The raw one-sided empirical p-values of local association significance were corrected for multiple testing using the Benjamini-Hochberg method per substate pair per sample, resulting in false discovery rates (FDR). Cells with significant spatial correlation (FDR < 0.05) were visualized by mapping significant spatial clusters (e.g., both substates high, one substate high-one

substate low, or both low) onto spatial scatterplots of samples (Supplementary Fig. 30-34).

**Spatial autocorrelation of plasma-cells and plaque tertiary lymphoid organ (PTLO) marker gene expression.** Lai et al. identified aggregations of lymphoid cells in plaques, especially enriched in B and plasma cells, as plaque tertiary lymphoid organs (PTLOs) in a recent publication[12]. Motivated by these findings, we sought to assess the presence of such domains in our samples by calculating the spatial autocorrelation of lymphoid cells. For this, we used the Getis-Ord G statistic, suited for "hotspot" analysis in spatial datasets[104]. As our Xenium panels did not include sufficient marker genes for a more detailed identification of lymphoid cell subtypes (i.e., B-cells), we were only able to test the autocorrelation of plasma-cells.

First, a spatial weight matrix was calculated for each sample using libpysal's (version 4.7.0) *DistanceBand()* function with a radius of 2500 µm. This larger threshold was chosen to allow detection of broader-scale clustering patterns and to reduce spurious local pairings, which were apparent at lower thresholds due to the relatively low number of plasma cells. Local Getis-Ord G statistics were then calculated using the *G_Local()* function from the esda package (version 2.4.3), with binarized cell identities across all spatial coordinates (1 = plasma cell, 0 = non-plasma cell). To generate an empirical background distribution, the permutations parameter was set to 999. Statistical significance of autocorrelation was determined by applying Benjamini-Hochberg correction to the empirical two-sided *p*-values, yielding false discovery rate (FDR) values.

In addition, we assessed the expression of four PTLO marker genes present in our panels (*CD19*, *CD3D*, *MZB1*, *XBP1*). Within each sample, normalised expression of the marker genes were compared between plasma-cells and all other cell types, using the Wilcoxon rank-sum test. The resulting two-sided p-values were corrected with the Benjamini-Hochberg method. Our analysis identified two plaques (patients 6 and 7) with consistent and significant plasma cell autocorrelation across both panels. Results obtained with panel 1 genes are shown in Supplementary Fig. 43, and those with panel 2 genes in Supplementary Fig. 44.

**RNA velocity analysis of macrophages in Xenium.** To further explore the potential relationship between the spatial distribution and expression patterns of macrophage subtypes observed in the Xenium data, we assessed their trajectories using RNA velocity analysis. For this purpose, we applied TFvelo[49], a transcription factor (TF) based model, since spliced and unspliced RNA counts cannot be derived from Xenium due to its in situ hybridization methodology. We subsetted our dataset to include only macrophages (panel 1: 35,265 cells; panel 2: 28,747) and performed preprocessing with the parameters recommended by the authors, using the batch-corrected scVI-UMAP representation as the low-dimensional embedding and acosh-normalised counts. We chose acosh normalisation because some cells had erroneously small area estimates from Baysor, which led to extreme values in area-normalized counts. These outliers interfered with TFvelo and prevented reliable trajectory inference. For the calculation of velocity stream plots, we only included genes with a larger latent time-pseudotime Spearman correlation of 0.3. Pseudotime and velocity streamplots are shown in Supplementary Fig. 38A, B for panel 1, and in Supplementary Fig. 39A, B for panel 2. Similar to our trajectory analysis of our scRNA-seq data, we constructed PAGA graphs for both panels using TFvelo-inferred pseudotime as priors and macrophage low-level cell types as node groups (panel 1: Supplementary Fig. 38C; panel 2: Supplementary Fig. 39C). For panel 1, we included the cell substate Mac_TREM2hi_CCL18hi into the construction of the PAGA graph, as well as in the colouring of cells in the streamplot.

Leveraging available cell coordinates, we plotted spatial scatterplots of the individual samples with macrophages coloured by TFvelo

pseudotime and cell type respectively (Supplementary PDF 3, Supplementary PDF 4).

Of note, macrophage cells with panel 2 dataset expressed more TFs (total number of TFs: 12; mean number of TFs/gene: 5.3), than panel 1 (total number of TFs: 5; mean number of TFs/gene: 1).

### Statistics & reproducibility
No statistical method was used to predetermine sample size. No data were excluded from the analyses. The experiments were not randomized. The Investigators were not blinded to allocation during experiments and outcome assessment.

### Reporting summary
Further information on research design is available in the Nature Portfolio Reporting Summary linked to this article.

### Data availability
Data from our own scRNA-Seq and single cell spatial transcriptomics experiments have been deposited in NCBI's Gene Expression Omnibus. The previously generated datasets, that are already published, are accessible through GEO Series accession number GSE247238 (scRNA-Seq, patients 1-10)[26,55,90,105] or in this manuscript in Supplementary Data 17 (raw bulk-seq count matrix: https://zenodo.org/records/17454209) and Supplementary Data 18[26,90,106–108]. Newly acquired datasets for this study are: GSE294291 (scRNA-Seq, patients 11–17) and GSE294466 (Xenium spatial transcriptomics). An additional external scRNA-Seq dataset of human carotid plaques[51] using a similar set-up of plaque vs. adjacent control was downloaded from the GEO repository (GSE159677). A preprocessed and manually annotated version of the GSE159677 dataset, along with the batch-corrected bulk RNA-seq count matrix used in this study, can be downloaded from Zenodo: https://zenodo.org/records/17521879. We also provide the final processed versions of our scRNA-seq and Xenium datasets as resources to the community. scRNA-seq data are available at (https://zenodo.org/records/17520898); Xenium datasets, along with subregion annotation and sample areas are available at: (https://zenodo.org/records/17526248). Interactive Supplementary Fig. 7 and 8, along with their legends can be downloaded from Zenodo: (https://zenodo.org/records/17523511).

### Code availability
Code is available at: (https://github.com/MendenLab/Carotid_artery_plaque).

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

## Acknowledgements

We want to thank CEP (comparative experimental pathology) unit at TUM Klinikum for technical support, Nadiya Glukha, Julia Ritzer and Annalena Huber (Institute of Molecular Vascular Medicine, TUM Klinikum) for technical assistance with CEA specimens; as well as Phong BH. Nguyen (Computational Health Center, Helmholtz Center Munich) for his valuable advice on scRNA-Seq data analysis. This work was supported by funding from the Bavarian State Ministry of Health and Care through the research project DigiMed Bayern (to HP and LM), the CRC1123 and TRR267 of the German Research Council (DFG; both to LM); an European Research Council (ERC) Consolidator grant (to LM) with the acronym LongTx and grant number 101088370; the BMBF in the framework of the Cluster4Future program (Cluster for Nucleic Acid Therapeutics Munich, CNATM - Project ID: 03ZU1201BM); the National Institutes of Health (R01HL171595, R01HL169766, R01HL166916 to MPR; R01HL151611, R01HL168174, R35HL177389 to HZ; 2R01HL107653-14, 2R01HL155431-05 to AT; R35HL145228 to IT; P01HL172741-01 to AT, IT, and HZ).

## Author contributions

D.G., F.P., J.W., and C.H. analysed the data. J.P., D.G., F.P., H.P., M.M. and L.M. interpreted the data. N.S., J.P., L.M., and D.B. designed cohorts and selected patients. J.P., N.S. and L.M. prepared the scSeq libraries and bulkSeq RNA. F.P. and H.P. sequenced the scSeq libraries, prepared and sequenced the bulk seq data. N.S. and D.B. prepared human plaque tissue for spatial transcriptomics. J.W. and K.S. performed spatial transcriptomic runs. H.Z., I.T., A.T., M.L., and M.R. guided the conceptualization, were involved in panel design, and provided critical feedback on the manuscript. J.P. and D.G. conceptualized and prepared the figures. J.P., D.G., M.M. and L.M. wrote the manuscript. All authors gave feedback on experiments, data interpretation, and the submitted manuscript.

## Funding

## Competing interests

L.M. has received research funds from Novo Nordisk (Malov, Denmark), Roche Diagnostics (Rotkreuz, Switzerland), and Bitterroot Bio (Palo Alto, CA, USA), and serves as a scientific advisor to Novo Nordisk (Malov, Denmark), DrugFarm (Guilford, CT, USA), and Angiolutions (Hannover, Germany). M.P.M. is a former employee of AstraZeneca (Cambridge, UK). MPM collaborates and is financially supported by GlaxoSmithKline (London, UK), F. Hoffmann-La Roche (Basel, Switzerland), and AstraZeneca (Cambridge, UK). D.B. serves on the advisory board for Terumo Aortic (UK), Medtronic (Minneapolis, USA), and COOK Medical (Bloomington, USA) and has received research funds and speaking fees from Artivion (Kennesaw, USA), Becton, Dickinson and Company (New Jersey, USA), Getinge (Gothenburg, Sweden), and Endologix (Irvine, USA). None of the relationships conflict with the present work. The other authors declare no relevant competing interests.

## Additional information

Jessica Pauli [1,2,13], Daniel Garger [3,13], Fatemeh Peymani[4,5,13], Justus Wettich[1,2], Nadja Sachs [2,6], Johannes Wirth [7], Katja Steiger [7], Christina Hillig [3], Hanrui Zhang [8], Ira Tabas[8], Alan Tall [8], Mingyao Li [9], Muredach P. Reilly [10], Daniela Branzan[6], Holger Prokisch[4,5,13], Michael P. Menden [3,11,13] ✉ & Lars Maegdefessel [1,2,12,13] ✉

[1]Institute of Molecular Vascular Medicine, TUM Klinikum, Technical University Munich, Munich, Germany. [2]German Center for Cardiovascular Research, partner site Munich Heart Alliance, Berlin, Germany. [3]Computational Health Center, Helmholtz Center Munich, Munich, Germany. [4]Institute of Human Genetics, TUM Klinikum, Technical University of Munich, Munich, Germany. [5]Institute of Neurogenomics, Computational Health Center, Helmholtz Munich, Neuherberg, Germany. [6]Department for Vascular and Endovascular Surgery, TUM Klinikum, Technical University Munich, Munich, Germany. [7]Institute for Pathology, TUM Klinikum, Technical University Munich, Munich, Germany. [8]Department of Medicine, Columbia University Irving Medical Center, New York, USA. [9]Department of Biostatistics and Epidemiology, Perelman School of Medicine, University of Pennsylvania, Philadelphia, USA. [10]Irving Institute for Clinical and Translational Research, Columbia University Department of Medicine, New York, USA. [11]Department of Biochemistry and Pharmacology, Bio21 Institute, The University of Melbourne, Parkville, Australia. [12]Department of Medicine, Karolinska Institutet, Stockholm, Sweden. [13]These authors contributed equally: Jessica Pauli, Daniel Garger, Fatemeh Peymani, Holger Prokisch, Michael P. Menden, Lars Maegdefessel.
✉e-mail: michael.menden@unimelb.edu.au; lars.maegdefessel@tum.de

