## [Transparent Peer Review file · Nature Communications]

Single cell spatial transcriptomics integration deciphers the morphological heterogeneity of atherosclerotic carotid arteries

Corresponding Author: Professor Lars Maegdefessel

Version 0:

Reviewer comments:

Reviewer #1

(Remarks to the Author)

The authors integrated spatial, single cell and bulk transcriptomics of carotid plaque, identifying inflammatory smooth muscle cell subtypes and localizing regions of neovascularization and macrophage activity. The authors applied a clever deconvolution algorithm to correct for immune cell bias and estimate true cell type proportions in bulkRNA-Seq data using scRNA-Seq-derived labels. The focus is on VSMCs and macrophages. There are several new and interesting findings:

1. TREM2 macrophages are not intermingled with C1Q macrophages, which sit below the endothelium
2. The authors convincingly discovered that immune cells are overrepresented in scRNA-Seq experiments
3. The authors discovered that there was no difference between symptomatic and asymptomatic plaques. In many studies, this is the only distinction between plaques provided

Specific comments

1. Previous work has provided systematic pathological classifications (Pasterkamp, Virmanu). Please discuss how your findings relate to these.
2. Note that endarterectomies do not provide access to the adventitia, where many immune cells reside. This may explain why the immune cell content is so low.
3. Recently, TLOs were discovered in human plaque¹ using methods similar to yours. Did you see any indications of TLOs?
4. Define CEA (probably the surgery).
5. The samples were macroscopically assessed and dissected into advanced atherosclerotic (plaque) and adjacent healthy/early lesion (control) tissue. It is not clear that the "control" is really healthy.
6. Since some samples had thrombus: what cells were in the thrombus?
7. FC thickness set at 200 μm : This is not easy to measure, because the sections may be at different angles, and cap thickness is not uniform. How reliable is this measure?
8. PI3K has many isoforms. Which ones are DE?
9. CD79A is not an immune regulator, but a B cell marker
10. Page 14: It is not clear which datasets are new (unpublished). Clearly distinguish between new and reanalyzed data.
11. Figure 2H: The immune cells are invisible. Either make an extra panel with a different scale, or use a log scale.
12. Figure 5A: numbers too small (not legible)
13. Figure 6A: show directionality (which genes were up in smokers?)
14. Page 4: "unpaired (n=232, n=65) and paired (n=42) designs, as well as stable (n=97) and unstable (n=126) plaques". Some may be overlapping. What was the total number of samples?
15. The authors often talk about "carotid plaques", but much of the resected material is not plaque but media. Maybe call them endarterectomy specimens.
16. SMCs, ECs and fibroblasts show up much better in single nuclei sequencing. Many studies used this approach, mitigating the bias. Please mention this and cite some studies.
17. The authors write that "VSMC-rich media from our CEA specimens (Fig 1A), a structure that has likely not been analysed in previous studies". Was there a difference in gene expression in the media of adjacent normal vs atherosclerotic artery tissue? Paired analysis should be possible.
18. "Four patients provided paired samples". Were these bilateral endarterectomies, with one side worse than the other? If so, what was the time elapsed between the two surgeries?

19. Page 6: "Correlation analysis revealed a strong inverse relationship between NC and lumen" suggests that the sum of the two may be constant and, by inference, the fibrous cap area % must also be constant.
20. The definition of controls is unclear. A sample with 45% intima is not a control. It seems that the authors call samples without NC controls.
21. It is interesting that there was no difference between symptomatic and asymptomatic plaques. This should be emphasized. In many studies, this is the only distinction provided.
22. Some patients showed "two NCs on each side of the lumen". This is interesting, but not conclusive. They may be connected in another plane.
23. In several scRNA-Seq studies and in at least 2 reviews, about half of the foam cells are SMC-derived and half are macrophages. There are two sets of macrophage foam cells and at least 3 other macrophage sets that are not foamy. The information obtained in the present study should allow these assignments.
24. Some of the correlations with clinical and demographic parameters are not statistically analyzed. Also, mentioning a single SNP in the smoking section is not appropriate. SNPs were not tested in this study.
25. Please define the difference between neighborhoods and subregions.
26. Figure 2E: the choice of colors is unfortunate. Impossible to see the cluster calling.
27. Figure 3D: what are "inflammatory mixed cells"? Doublets?
28. Figure 3E: Age seems to be a driver for necrotic core. This should be emphasized.
29. Figure 5: What statistical test was used to assign the morphological clusters? How do we know there should be 5 clusters? Or is this arbitrary?
30. Figure 7C: How were patients 4 and 8 selected?

1 Lai, Z. et al. Single-cell spatial transcriptomics of tertiary lymphoid organ-like structures in human atherosclerotic plaques. *Nat Cardiovasc Res*, doi:10.1038/s44161-025-00639-9 (2025).

(Remarks on code availability)

Reviewer #2

(Remarks to the Author)

Overall Summary:

Pauli et al. present their investigation seeking to integrate multiple transcriptomic methodologies (bulk RNA seq, scRNAseq, and spatial transcriptomics) as a means to molecularly define carotid atherosclerotic disease. Using arterial and atheroma samples collected during CEA, the investigators attempt to elucidate a spatially-resolved molecular architecture underlying plaque development. The authors highlight unique plaque morphologies linked to different cell type compositions, disease heterogeneity (i.e. symptomatology), and clinical demographics (i.e. smoking, etc) with inference towards plaque development and stability. Notably, they uncover spatial associations between HMOX1+ macrophages as they potentially transition to foam cells (TREM2+).

While the manuscript is well written and demonstrates technical rigor, the most important limitation is a lack of methodologic novelty or conceptual impact to our molecular understanding of atherosclerosis. Notably, HMOX1 and TREM2 macrophages are well established in the progression of atherosclerosis (PMID: 39731912 – HMOX1, 38974464 – TREM2) and modulation of macrophage polarity (PMID: 30282830 – HMOX1, 38646596 – TREM2). Furthermore, other spatial transcriptomic investigations of arterial atherosclerosis have recently been published (PMID: 39234691 and 39715784). Considering these limitations, the current manuscript offers incremental advancement.

Other Criticisms include:

1. Association with symptomatology / stability is uncertain. One area of potential significance was the assertion that the current findings molecularly define symptomatic (vs. asymptomatic) carotid artery stenosis. As presented, however, the data do not clearly support these claims. Figure 5 provides an example of this missed opportunity. A clearer presentation of the results would have been based upon symptom status as opposed to various morphologic clusters.
2. Surprising lack of DEG in bulk RNAseq. Figure 2c shows relatively few differences in DEG across stable and unstable plaques. This is in contrast to current literature (39234691), yet no explanation is given.
3. Underpowered study with significant demographic heterogeneity. It is difficult to interpret some of the results due to relatively few patients (N=4 Ctl) and variable clinical demographics.
4. Biased spatial transcriptomic approach based off Panels 1 and 2. Finally, the spatial transcriptomic analysis used a custom (pre-defined) gene set (~500) for cellular and disease status identification. While this approach is helpful in resolving cellular spatial location, it falls short of providing true spatially-resolved, single cell transcriptomic analysis. This undercuts the novelty.

(Remarks on code availability)

Reviewer #3

(Remarks to the Author)

Summary

This manuscript presents a comprehensive spatially resolved transcriptomic analysis of human carotid atherosclerotic plaques. The authors integrate bulk RNA-Seq, single-cell RNA-Seq (scRNA-Seq), and single-cell spatial transcriptomics (Xenium platform) on paired samples from patients undergoing carotid endarterectomy (CEA). Their multimodal approach reveals spatial microenvironments, including distinct VSMC and macrophage subtypes associated with plaque vulnerability, progression, and smoking status. They propose a model of subluminal HMOX1⁺ macrophage transdifferentiation into lipid-handling TREM2⁺ cells. A novel morphological clustering of plaques into “structured” and “chaotic” types is introduced, offering mechanistic insights into spatial immune-VSMC interplay and plaque destabilization.

Major Comments

1. Justification of Technology Choice: The authors employ the Xenium platform for spatial transcriptomics but do not justify its selection over other platforms like 10x Visium HD. A justification regarding resolution, sensitivity, and gene coverage tradeoffs is needed.
2. Immune Cell Overrepresentation Not Corrected: Despite deconvolution, the overrepresentation of immune cells in scRNA-Seq is not adequately addressed in the spatial analysis. This may bias subregion interpretations.
3. Smoking Association Requires Caution: The observed association with smoking is only partially supported by bulk and scRNA-Seq data. Acknowledgment of limited reproducibility is needed.
4. Trajectory Analysis of Macrophages: The proposed HMOX1⁺ to TREM2⁺ trajectory should be validated with orthogonal methods beyond TFvelo and visualized more explicitly.
5. Subregion Definition and Reproducibility: Manual annotation of regions may introduce bias. Providing inter-rater reliability or uncertainty metrics would enhance robustness.
6. Sample Size for Spatial Analysis: The n=12 for spatial transcriptomics limits generalizability. Interpretations of clustering and risk associations should be more cautiously framed.
7. Neighbourhood Cluster Analysis: Statistical support (e.g., permutation tests, spatial proximity significance) for proposed functional zones is missing and should be included.
8. Clinical Utility and Predictive Value: The translational relevance of findings should be discussed. Can DEGs or spatial patterns serve as prognostic markers for plaque rupture?

Minor Comments

1. Terminology Clarification: Terms like 'structured' vs. 'chaotic' could be more quantitatively or mechanistically defined.
2. Figure Readability: Dense color legends in Figures 6–8 make interpretation difficult. Consider enhancing clarity.
3. Figure 2 caption: “-log₁₀ value” => “-log₁₀ p-value”

(Remarks on code availability)

No code is provided.

Version 1:

Reviewer comments:

Reviewer #1

(Remarks to the Author)

The authors addressed all concerns. However, figure 2F-I is still unsatisfactory. Please show the immune cells using a different y axis scale. Since there are few immune cells, differences cannot be appreciated from the current figure

(Remarks on code availability)

Reviewer #2

(Remarks to the Author)

Pauli et al. represent their investigation seeking to integrate multiple transcriptomic methodologies (bulk RNA seq, scRNAseq, and spatial transcriptomics) as a means to molecularly define carotid atherosclerotic disease. Using arterial and atheroma samples collected during CEA, the investigators attempt to elucidate a spatially-resolved molecular architecture underlying plaque development. The authors highlight unique plaque morphologies linked to different cell type compositions, disease heterogeneity (i.e. symptomatology), and clinical demographics (i.e. smoking, etc) with inference towards plaque development and stability.

Previous critiques raised questions regarding 1) associations with disease symptomatology, 2) Comparison of the presented results to previously published data (coronary and carotid), 3) Novelty and clinical utility of the results, and 4) clarification of methodologies and sample size. To this end, the investigators have provided a robust response that is highly responsive to the reviewer concerns. This includes adjustments to the data presentation, new analysis, and additional commentary throughout the manuscript.

Regarding novelty, the team has provided sound arguments highlighting the relative impact their results may have on the molecular understanding of carotid artery disease and plaque progression. While the HMOX1 and TREM2 phenotypes are well established, the manuscript provides other novel findings including 1) spatial resolution of macrophages at the unstable plaque shoulder, 2) EC co-localization for neovascularization and immune cell entry into the necrotic core / plaque, 3) SMC transitions associated with plaque heterogeneity. While the clinical utility and implementation of many of these results

remains to be determined, the work nonetheless provides a thorough examination of carotid atherosclerotic disease.

Other Criticisms that have been adequately addressed:

1. Association with symptomatology is uncertain. Concerns have been well addressed and edited within the manuscript.
2. Surprising lack of DEG in bulk RNAseq. Concerns have been well addressed and edited within the manuscript.
3. Underpowered study with significant demographic heterogeneity. Concerns have been well addressed and edited within the manuscript.
4. Biased spatial transcriptomic approach based off Panels 1 and 2. Concerns have been well addressed and edited within the manuscript.

(Remarks on code availability)

Reviewer #3

(Remarks to the Author)

All my comments are adequately addressed.

(Remarks on code availability)

The code provided in code ocean are well organized. But the code ocean platform has some issues running the code with the 'Reproducible Run'. Also, only R code shows up properly. The ipython notebook must be downloaded and run locally, which is beyond my bandwidth.

Point-by-point response

We would like to thank the reviewers and editors for the assessment of our submitted manuscript. We carefully addressed all of the reviewers' concerns with new data analysis as well as clarifications, which we believe has enabled us to submit a substantially improved version of our study.

Reviewer #1 (Remarks to the Author):

The authors integrated spatial, single cell and bulk transcriptomics of carotid plaque, identifying inflammatory smooth muscle cell subtypes and localizing regions of neovascularization and macrophage activity. The authors applied a clever deconvolution algorithm to correct for immune cell bias and estimate true cell type proportions in bulkRNA-Seq data using scRNA-Seq-derived labels. The focus is on VSMCs and macrophages. There are several new and interesting findings:

1. TREM2 macrophages are not intermingled with C1Q macrophages, which sit below the endothelium

2. The authors convincingly discovered that immune cells are overrepresented in scRNA-Seq experiments

3. The authors discovered that there was no difference between symptomatic and asymptomatic plaques. In many studies, this is the only distinction between plaques provided

Specific comments

1. Previous work has provided systematic pathological classifications (Pasterkamp, Virmani). Please discuss how your findings relate to these.

We thank the reviewer for this important comment on variations of assessing plaque vulnerability statuses. This is obviously a rather debated question, and we believe that all current approaches have their advantages but also disadvantages. Pioneers in the field, like the mentioned groups led by Renu Virmani and Gerard Pasterkamp respectively, established their plaque phenotyping based on the available vascular specimens their labs and institutes had and partly still have access to. For the carotid plaque specimens in our study, we utilize the Rothwell/Redgrave criteria of the Oxford Plaque Study, being explained in more detail below. We made sure to more clearly state that characterization using this criterion for carotid artery disease in our revised manuscript.

Work by Virmani and colleagues was in large parts performed and influenced by postmortem autopsy specimens from different vascular beds, with a focus on coronary artery tissue specimens. According to these autopsy studies, the most common type of vulnerable plaques are so called thin-cap fibroatheroma (TCFA) with the following key features: fibrous cap thickness $<65\mu\text{m}$ (for coronary arteries and not carotids as used in our study) located over a lipid/cholesterol crystal-rich necrotic core,

high presence of T cell and macrophage infiltrations in close proximity to the fibrous cap (plaque shoulder), low numbers of smooth muscle cells, and areas of intraplaque bleedings (“Pathology of the Vulnerable Plaque,” 2006).

The work by Gerard Pasterkamp offers a broader, multifactorial definition of vulnerable atherosclerotic plaques, emphasizing not only histological features but also biological activity, mechanical stress, as well as systemic factors. In studies by his group, mostly tissue specimens from carotid endarterectomies (CEA) are utilized and investigated predominantly based on patient symptoms (asymptomatic or symptomatic for stroke, transient ischemic attack, and amaurosis fugax; (Langley et al., 2017; Mocci et al., 2024)), and more recently based on transcriptomic expression patterns (Mokry et al., 2022).

A third approach, which was used in our current study stems from Drs. Rothwell and Redgrave and the Oxford Plaque Studies. Here, a crucial fibrous cap thickness of $>/< 200\mu\text{m}$ is established exclusively for carotid artery disease, taking into account carotid imaging modalities and clinical outcomes from patients with stroke (Lovett et al., 2004; J. N. Redgrave et al., 2008; J. N. E. Redgrave et al., 2006). Based on our experience and work published by us in previous studies (Bashore et al., 2024; Fasolo et al., 2021), this carotid artery specific assessment of plaque vulnerability suits our approach of combining histomorphological plaque characterization, integrative multi-transcriptomic profiling (with bulk, single cell, and spatial transcriptomics) and correlation with clinical patient information (symptoms, risk factors, accompanying diseases) best.

We added the following statement in the **Results** section to clearly state our approach at the beginning of the manuscript:

“Human carotid plaques from CEA patients were selected based on conclusive morphology, including remnant media and macroscopically assessed and dissected into advanced atherosclerotic (plaque) and adjacent early lesion (control) tissue. Plaques are characterized based on the Oxford Plaque Studies (Lovett et al., 2004; J. N. Redgrave et al., 2008; J. N. E. Redgrave et al., 2006). Both pieces of this paired unique setup underwent bulk- and scRNA-Seq.”

We furthermore clarified the **Methods** section in this regard:

“The histological classification of carotid plaques was performed following the guidelines established by the American Heart Association (AHA(Stary, 2000; Stary et al., 1992, 1994, 1995)) and the Redgrave & Rothwell criteria (Oxford Plaque Study)(J. N. Redgrave et al., 2008) for plaque stability.”

2. Note that endarterectomies do not provide access to the adventitia, where many immune cells reside. This may explain why the immune cell content is so low.

The reviewer is correct that plaques from carotid endarterectomies (CEA) lack adventitia, as the surgical removal of the lesion must exclude the adventitia remaining in the patient. Thus, CEA plaques contain the media layer, which leads to a higher SMC content in general and therefore as suggested by the reviewer likely a lower abundance of immune cells. Of note, previous prominent studies have used FACS sorting of CD45+ cells to analyze human carotid lesions, which fueled the assumption that human plaques mainly consist of immune cells (Depuydt et al., 2023; Dib et al., 2023; Fernandez et al., 2019). This is the main reason why we performed single cell-bulk RNA-Seq deconvolution (utilizing BayesPrism) in our manuscript, enabling us to overcome the immune cell bias of applying single cell RNA-Seq in human plaques. Importantly, our finding is confirmed by our own spatial transcriptomics (Xenium) analysis, in concordance with very recent literature (Bleckwehl et al., 2025; Traeuble et al., 2024).

We clarified this matter in our **Introduction**:

“The intima-media layer, rich in VSMCs and fibroblasts, represents the pre-atherosclerotic arterial wall. The adventitia is not excised during the surgical procedure and is retained in the patient. All three substructures...”

3. Recently, TLOs were discovered in human plaque¹ using methods similar to yours. Did you see any indications of TLOs?

1 Lai, Z. et al. Single-cell spatial transcriptomics of tertiary lymphoid organ-like structures in human atherosclerotic plaques. *Nat Cardiovasc Res*, doi:10.1038/s44161-025-00639-9 (2025).

The TLOs described in the Lai *et al.* paper were aggregations of lymphoid cells, characterised by plasma and B cells in particular. As our current study focuses on integration of bulk, single cell and spatial transcriptomics in an unbiased fashion (not targeting specific structures), our Xenium probe panels did not necessarily include sufficient marker genes for a more detailed identification of lymphoid cell subtypes. In addition, plasma and B cell marker gene counts are comparatively low in relation to macrophage and SMC marker gene counts.

But to directly address the reviewer’s comment, we carefully examined whether TLOs could still be identified in our plaques. For this, we calculated the spatial autocorrelation of plasma cells (Gerdis-Ord G statistic) and assessed the expression of four genes overlapping between our panels and the PTLO marker genes reported by Lai et al. (CD19, CD3D, MZB1, and XBP1). Plaques from patients 6 and 7 consistently exhibited significant plasma cell autocorrelation across both the panel 1 (**Suppl. Fig. 43A,B**, see below) and panel 2 (**Suppl. Fig. 44A,B**, see below) datasets. In addition, upon revisiting our neighbourhood cluster analysis, we found cluster 13 in both panel 1 and panel 2 to be enriched in plasma cells (**Suppl. Fig. 28A,B**). These clusters were detected only in the plaque of patient 6 and showed strong spatial

overlap with the plasma cell aggregation in the necrotic core of this sample (**Suppl. Fig. 45**, see below).

However, among the four marker genes, only MZB1 and XBP1 showed significant expression within plasma cells in plaques from patient 6 and 7 (**Suppl. Fig. 43C,D; Suppl. Fig. 44C,D**).

Taken together, the spatial patterns of plasma cells may suggest the presence of PTLOs in the plaques of patients 6 and 7, however, these cells expressed only two of the four PTLO marker genes. This highlights that, given the limited ability of our current gene panels to resolve lymphoid cell subtypes, further clarification of these structures will require future analyses with more dedicated gene panels. To incorporate this analysis, we expanded our **Methods** section:

“Spatial autocorrelation of plasma-cells and plaque tertiary lymphoid organ (PTLO) marker gene expression

Lai et al. identified aggregations of lymphoid cells in plaques, especially enriched in B and plasma cells, as plaque tertiary lymphoid organs (PTLOs) in a recent publication (Lai et al., 2025). Motivated by these findings, we sought to assess the presence of such domains in our samples by calculating the spatial autocorrelation of lymphoid cells. For this, we used the Getis-Ord G statistic, suited for “hotspot” analysis in spatial datasets (Ord & Getis, 1995). As our Xenium panels did not include sufficient marker genes for a more detailed identification of lymphoid cell subtypes (i.e. B-cells), we were only able to test the autocorrelation of plasma-cells.

First, a spatial weight matrix was calculated for each sample using libpysal’s DistanceBand() function with a radius of 2500 μ m. This larger threshold was chosen to allow detection of broader-scale clustering patterns and to reduce spurious local pairings, which were apparent at lower thresholds due to the relatively low number of plasma cells. Local Getis-Ord G statistics were then calculated using the G Local() function from the esda package, with binarized cell identities across all spatial coordinates (1 = plasma cell, 0 = non-plasma cell). To generate an empirical background distribution, the permutations parameter was set to 999. Statistical significance of autocorrelation was determined by applying Benjamini–Hochberg correction to the empirical p-values, yielding false discovery rate (FDR) values.

In addition, we assessed the expression of four PTLO marker genes present in our panels (CD19, CD3D, MZB1, XBP1). Within each sample, normalised expression of the marker genes were compared between plasma-cells and all other cell types, using the Wilcoxon rank-sum test. The resulting p-values were corrected with the Benjamini–Hochberg method.

Our analysis identified two plaques (patients 6 and 7) with consistent and significant plasma cell autocorrelation across both panels. Results obtained with

panel 1 genes are shown in **Suppl. Fig. 43**, and those with panel 2 genes in **Suppl. Fig. 44**.

Along with the **Results** section:

“A recent publication identified plasma and B-cell rich areas, interspersed with VSMCs and macrophages as plaque tertiary lymphoid organs (PTLOs) (Lai et al., 2025). In our samples, spatial analysis revealed significant plasma cell enrichment in the plaques of patients 6 and 7 (**Suppl. Fig. 43A,B; Suppl. Fig. 44A,B**). In addition, the plasma cell aggregation domain in the NC of patient 6 was identified as cluster 13 (**Suppl. Fig. 45**), enriched in VSMCs, macrophages and plasma cells (**Suppl. Fig. 28A**). Although suggestive of PTLOs, only two (MZB1, XBP1) out of four PTLO marker genes (Lai et al., 2025) in our panels showed high expression in plasma cells (**Suppl. Fig. 43C,D; Suppl. Fig. 44C,D**), highlighting the need for further clarification with more specialised gene panels.

By performing neighbourhood clustering for panel 2, we detected similar neighbourhoods with similar features in the same plaques, with neighbourhood 0 and 1 being “VSMC neighbourhoods”, neighbourhoods 2, 5 and 8 being “macrophage neighbourhoods”, and neighbourhood 10 being the “EC_Neovessel neighbourhood”, and neighbourhood 13 the “plasma cell neighbourhood” (**Suppl. Fig. 28B,C**).“

We adapted the **Introduction** and mention the suggested publication, together with other spatial transcriptomics publications in human atherosclerotic plaques:

“Sequencing technologies, including bulkRNA-Seq, single cell (sc) RNA-Seq, and spatial transcriptomics, have helped tremendously to better determine molecular alterations and cellular composition in human diseases (Bleckwehl et al., 2025; Campos et al., 2025; Gastanadui et al., 2024; Lai et al., 2025; Longo et al., 2021; Sun et al., 2023)”

Suppl. Fig. 43: Spatial distribution of Plasma-cells and expression of plaque tertiary lymphoid organ (PTLO)-related marker genes in Xenium panel 1 dataset. (A-B) Spatial distribution of plasma cells identified in the Xenium panel 1 dataset. In the two plaques shown (panel A: patient 6; panel B: patient 7), plasma cells exhibited significant spatial autocorrelation. Plasma cells are coloured by the autocorrelation metric (local Getis-Ord G), while other cell types are shown in grey. (C-D) Normalised expression of PTLO-related marker genes (Lai et al.) present in panel 1 genes, compared between plasma cells and all other cell types. Panel (C) shows gene expression in plaque cells from patient 6, and panel (D) from patient 7. The x-axis depicts the marker genes present in panel 1, and dot colours indicate cell type (plasma cell vs. other cell types). Comparisons of normalized expression were performed using the Wilcoxon rank-sum test, and raw p-values were adjusted with the Benjamini-Hochberg method to obtain false discovery rate (FDR) values.

Suppl. Fig. 44: Spatial distribution of Plasma-cells and expression of plaque tertiary lymphoid organ (PTLO)-related marker genes in Xenium panel 2 dataset. (A-B) Spatial distribution of plasma cells identified in the Xenium panel 2 dataset. In the two plaques shown (panel A: patient 6; panel B: patient 7), plasma cells exhibited significant spatial autocorrelation. Plasma cells are coloured by the autocorrelation metric (local Getis-Ord G), while other cell types are shown in grey. (C-D) Normalised expression of PTLO-related marker genes (Lai et al.) present in panel 2 genes, compared between plasma cells and all other cell types. Panel (C) shows gene expression in plaque cells from patient 6, and panel (D) from patient 7. The x-axis depicts the marker genes present in panel 2, and dot colours indicate cell type (plasma cell vs. other cell types). Comparisons of normalized expression were performed using the Wilcoxon rank-sum test, and raw p-values were adjusted with the Benjamini-Hochberg method to obtain false discovery rate (FDR) values.

Suppl. Fig. 45: Spatial distribution of Plasma-cells and neighbourhood clusters 13. (A-B) Spatial distribution of cells in the plaque of patient 6, as identified in the panel 1 (A) and panel 2 (B) datasets. Each dot represents a cell, coloured by cell type. Plasma cells are shown with reduced transparency for improved visualization compared to other cell types. (C-D) Spatial distribution of cells in the plaque of patient 6, as identified in the panel 1 (C) and panel 2 (D) datasets. Each dot represents a cell, coloured by the respective neighbourhood cluster (C: panel 1; D: panel 2). Cells belonging to the plasma cell-rich cluster 13 are shown with reduced transparency for improved visualization compared to cells in other clusters.

4. Define CEA (probably the surgery).

We apologize for this mistake. Carotid endarterectomy (CEA) is defined as the surgical removal of plaque buildup in carotid arteries. We adapted the Introduction as well as the Methods section accordingly:

Introduction:

“Atherosclerotic plaques from patients undergoing carotid endarterectomy (CEA), the surgical removal of plaque buildup in carotid arteries, consist of three major subregions with distinct cellular and molecular features.”

Methods:

“Plaque samples were collected from patients undergoing carotid endarterectomy (CEA), the surgical removal of plaque buildup in carotid arteries.”

5. The samples were macroscopically assessed and dissected into advanced atherosclerotic (plaque) and adjacent healthy/early lesion (control) tissue. It is not clear that the “control” is really healthy.

We thank the reviewer for this comment. It is correct that the control pieces in this current study do not reflect a fully healthy artery, as this tissue still stems from elderly patients suffering from carotid artery disease. The ‘healthier’/early or lesser diseased lesions are located adjacent to the highly stenotic carotid artery tissue. Controls in our work are defined as the least diseased piece of tissue from the patient, with no signs of necrotic core formation (which is the defining hallmark of an advanced carotid artery plaque). Intimal thickening as well as early signs of fatty streaks - and in some cases immune cell infiltrates might still be present, as this initiation process occurs already very early in life and is highly present in all (in particular Western) populations (Finn et al., 2010; Masawa et al., 1994; Pasterkamp et al., 1998; Velican, 1969; Virmani et al., 2000).

We carefully screened our revised manuscript again and removed the word “healthy” from the Results section (1x), as well as from the Methods section (1x) when explaining and discussing sample selection and comparison thereof.

Results:

“Human carotid plaques from CEA patients were selected based on conclusive morphology, including remnant media and macroscopically assessed and dissected into advanced atherosclerotic (plaque) and adjacent tissue healthy/from the same patient (matched paired diseased-control samples). Plaques were characterized for stability based on the Oxford Plaque Studies. Both pieces of this unique setup (statistically powerful paired analysis possible) underwent bulk- and scRNA-Seq. In addition, formalin-fixed paraffin (FFPE)-

embedded samples of neighbouring pieces—maintaining the same paired plaque vs. control configuration—were processed for histomorphological and spatial transcriptomics analyses

Methods:

“We selected the plaque scRNA-seq dataset published by Alsaigh et al (Alsaigh et al., 2022), which had a similar experimental setup (matched atherosclerotic core - proximal adjacent samples from same patients) to ours (matched diseased - healthy control samples from same patients).”

Furthermore, we emphasized this unique setup of early/control and advanced lesion from the same individual patient that allows for statistical extremely powerful analyses in the same paragraph of the **Results** section:

*“Both pieces of this paired unique setup underwent bulk- and scRNA-Seq. Both pieces of this unique setup (statistically powerful paired analysis is possible) underwent bulk- and scRNA-Seq. In addition, formalin-fixed paraffin (FFPE)-embedded samples of neighbouring pieces—maintaining the same paired plaque vs. control configuration—were processed for histomorphological and spatial transcriptomics analyses (**Fig. 1**).”*

Additionally, we clarified our definition of the early lesion in the **Methods** section:

“Immediately after removal from the intraoperative situs, plaque samples were transferred to pre-chilled PBS (Merck, Germany) for transportation. The carotid plaques are then cut into several pieces. Specifically, for this study, we selected the most advanced (i.e., most unstable) part of the plaque and whenever possible also the least diseased part of the plaque for analysis (Suppl. Fig. 3). This unique setup allows for statistically powerful paired analyses of two pieces of the same tissue from the same individual patient.”

Specifically for this purpose, we also created a new supplemental figure - **Suppl. Fig. 3** - that illustrates the explanations from above:

Figure Legend: **“Suppl. Fig. 3: Carotid Artery Plaque tissue sectioning and characterisation routine. (A) shows the whole plaque upon surgical removal (CEA). (B) shows the sectioned plaque getting processed for molecular and histomorphological characterization. The arrows indicate the most advanced and earliest piece of tissue in this example. (C) Histological analysis is performed according to the American Heart Association (AHA) classification guidelines (Stary, 2000; Stary et al., 1992, 1994, 1995) using different histological stainings (HE and EvG). FC thickness is evaluated based on the Oxford Plaque Study ((J. N. Redgrave et al., 2008)) and measured using the**

AxioScan Software (Leica) or QuPath. Here, all four measured positions of the FC are below 200µm, therefore this carotid plaque was classified as unstable.”

Suppl. Fig. 3: Carotid artery plaque tissue sectioning and characterisation routine. (A) shows the whole plaque upon surgical removal (CEA). (B) shows the sectioned plaque getting processed for molecular and histomorphological characterisation. The arrows indicate the most advanced and earliest piece of tissue in this example. (C) Histological analysis is performed according to the American Heart Association (AHA) classification guidelines (Stary 2000; Stary et al. 1992; Stary et al. 1995; Stary et al. 1994) using different histological stainings (HE and EvG). FC thickness is evaluated based on the Oxford Plaque Study (Redgrave et al. 2008) and measured using the AxioScan Software (Leica) or QuPath. Here, all four measured positions of the FC are below 200 μm, therefore this carotid plaque was classified as unstable.

6. Since some samples had thrombus: what cells were in the thrombus?

This is an important and interesting comment by the reviewer. We excluded cells originating from intraluminal thrombi from our clustering and DGE analyses. This was however not mentioned in the previously submitted version of our manuscript. We now expanded the respective sections in our **Methods** part:

“For differential gene expression (DGE) analysis we again utilised the pseudobulk approach: we created pseudobulk samples by aggregating cells of the same sample within low-level cell types and subsequently carried out DGE analysis for multiple condition variables using DESeq2, always controlling for “patient” as factor. Cells originating from intraluminal thrombi were excluded from DGE analysis.”

“First, we calculated the cell fraction percentage of the different cell type substates normalised within the sample, then performed hierarchical clustering with Euclidean distance and Ward’s-linkage method. Cells originating from intraluminal thrombi were excluded from both the cell fraction calculations and clustering steps.”

Further, in the revised manuscript, we included the analysis of the five plaques and one control sample that contained cells detected in thrombi. We also expanded the Methods section accordingly and added a new supplementary figure (**Suppl. Fig. 59**):

*“We also performed hierarchical clustering of samples containing intraluminal thrombi, based on the low-level substate cell fractions within cells detected in each thrombus (**Suppl. Fig. 59**). Compared with the overall dataset, thrombi contained fewer cell substates, showed less variation across samples, and were predominantly composed of neutrophils, inflammatory mixed cells, and CD8 effector T-cells. The endothelial cells detected within intraluminal thrombi most likely originated from intimal layer lacerations introduced during sample preparation.”*

Suppl. Fig. 59: Clustering of samples based on thrombus-derived cells. Clustermap showing hierarchical clustering of samples containing thrombi. Clustering was performed on (A) panel 1 and (B) panel 2 low-level substate cell fractions within each sample's thrombus. "P" indicates "Patient" in the x-axis labels. Patient metadata is shown in the legend above the clustermap.

7. FC thickness set at 200 µm: This is not easy to measure, because the sections may be at different angles, and cap thickness is not uniform. How reliable is this measure?

We acknowledge the reviewer's concern regarding the challenges in measuring fibrous cap (FC) thickness due to sectioning angles and intrinsic variability in cap morphology. Our human samples originate from patients undergoing carotid endarterectomy, and all tissues are processed under a standardized biobank protocol that has been in place for >15 years. We made sure to add extra information into the revised manuscript to better explain how this is performed.

We added a detailed description of the FC measurement to **Suppl. Fig. 3 (Suppl. Fig. 3C)**, that can be found in your **Question #5**.

Please also see **Question #5** for additional details about how we addressed changes regarding our sample setup in the text for your clarification.

8. PI3K has many isoforms. Which ones are DE?

As we used short read sequencing to obtain bulk- and scRNA-Seq data, we cannot make distinctions on gene isoforms.

We have clarified this in the **Methods** section of our revised manuscript:

“RNA was purified using poly-T-oligo-attached magnetic beads and the TruSeq stranded total RNA kit and TruSeq stranded mRNA kit (Illumina, San Diego, USA) were used to prepare the short-read sequencing libraries.”

9. CD79A is not an immune regulator, but a B cell marker

We thank the reviewer for highlighting this mistake. We obviously agree and have changed the text accordingly:

“Key DEGs that were increased in advanced plaque vs. early lesion controls included immune regulators like FLT4 (Schmeisser et al., 2006) or B-cell markers like CD79A (Ma et al., 2021) and...”

10. Page 14: It is not clear which datasets are new (unpublished). Clearly distinguish between new and reanalyzed data.

We have clarified the text in our revised manuscript accordingly:

*“Data from our own scRNA-Seq and single cell spatial transcriptomics experiments have been deposited in NCBI’s Gene Expression Omnibus. The previously generated datasets, that are already published, are accessible through GEO Series accession number GSE247238 (scRNA-Seq, patients 1-10) (Fidler et al., 2024; Paloschi et al., 2024; Parma et al., 2025; Pourteymour et al., 2024) or in this manuscript in **Suppl. Table 6** and **Suppl. Table 7** (bulk seq data) (Bonfiglio et al., 2025; Fidler et al., 2024; Paloschi et al., 2024; Tan et al., 2025; Wang et al., 2024). Newly acquired datasets for this study are: GSE294291 (scRNA-Seq, patients 11-17) and GSE294466 (Xenium spatial transcriptomics).”*

11. Figure 2H: The immune cells are invisible. Either make an extra panel with a different scale, or use a log scale.

We have adapted **Figure 2H** to match the style of the previous figures, and split it into two subplots: **Figure 2H** and **Figure 2I**. This allows the reader to put the visualized results to determine differences of ‘Control vs. Plaque’ specimens.

Figure 2: Single-cell and bulk RNA sequencing results. (A-C) Differential gene expression results performed using the bulk RNA-seq data. Results of unpaired Plaque vs. Control sample comparison (A), results of paired Plaque vs. Control samples from the same patients (B), results of comparing unpaired Stable vs. Unstable samples (C). X-axis depicts the log₂ fold change of gene expression across the conditions, y-axis depicts the -log₁₀ p-value of Benjamini-Hochberg adjusted significance of the log₂ fold change (false discovery rate, FDR). Red colouring: significant overexpression, blue colouring: significant underexpression, significance thresholds: FDR<0.05, log₂ fold change: +/- 1. Highlighted genes have Xenium probes. (D-E) UMAP low-dimensional representation of gene expression in cells of human atherosclerotic plaques, measured with scRNA-seq method. Cells are coloured by high-level cell types (D), and by low-level cell types (E). (F-G) Cell fractions of high-level cell types per sample in scRNA-seq (F, n=31) and bulk RNA-seq (G, n=297). In the case of bulk RNA-seq, cell fractions are calculated by a deconvolution algorithm. Bars represent the mean of cell fractions per cell type across all samples. (H-I) Cell fractions of high-level cell types of control (H) and plaque (I) bulk RNA-seq samples, calculated by deconvolution. UMAP: Uniform Manifold Approximation and Projection, FB: fibroblast, NK-cell: natural-killer cell, VSMC: vascular smooth muscle cell.

12. Figure 5A: numbers too small (not legible)

We thank the reviewer for this suggestion. To improve readability, we have split the original Figure 5 into two separate figures. **Revised Figure 5** now contains only the enlarged cluster map with improved readability, while **revised Figure 6** presents the scatterplots of the morphological clusters. Figure numbering has been updated accordingly throughout the manuscript and figure captions.

Figure 5: Clustering of plaques into morphological clusters. Clustermap showing hierarchical clustering of plaques into 4 morphological clusters based on panel 1 low-level substate cell fractions within each sample. „P“ stands for „Patient“ in the x-axis labels, patient metadata legend is depicted above clustermap. „sympt“.: symptomatic; „asympt“.: asymptomatic.

Figure 6: Spatial scatterplots of morphological clusters. (A-D) Spatial scatterplots of plaques grouped by morphological clusters. Dot colouring refers to the top 5 most abundant cell subtypes per cluster, „L“ depicts lumen. FC: fibrous cap; NC: necrotic core

13. Figure 6A: show directionality (which genes were up in smokers?)

We thank the reviewer for the suggestion. We updated and revised the volcano plot with a legend clearly indicating directionality of DEGs in smokers.

14. Page 4: “unpaired (n=232, n=65) and paired (n=42) designs, as well as stable (n=97) and unstable (n=126) plaques”. Some may be overlapping. What was the total number of samples?

We revised the text in the **Method** section accordingly and added the following statement for clarification:

“After filtering, 14,572 genes were detected across a total of 297 samples derived from 252 patients. The dataset comprises transcriptomes from 232 advanced plaques and 65 early lesion controls. Among the plaques, 97 were classified as stable and 126 as unstable, with 9 remaining unclassified based on our histological analysis. For 42 patients, paired samples of plaque and control lesions were available, allowing for paired differential expression analysis.”

Furthermore, we clarified this in the **Results** section as well:

“...classified by FC thickness above or below 200 μ m. The advanced plaques were further classified into stable (n=97) and unstable (n=126) lesions based on FC thickness above or below 200 μ m¹⁸. Nine advanced plaque specimens (of n=232) were rendered non-classifiable, and not included into the stability-based comparison.”

15. The authors often talk about “carotid plaques”, but much of the resected material is not plaque but media. Maybe call them endarterectomy specimens.

We understand and partly agree with the reviewer that the word ‘plaque’ might be misleading - at least for some of the specimens being analyzed. However, the media is never the main component for any of the CEA specimens being assessed (see HE stains shown in **Figure 4, Suppl. Fig. 11** and **Suppl. Fig. 12** in the manuscript). We as well as others have termed these endarterectomy specimens carotid plaques, as the indication for surgical removal is the stenotic plaque and the occurrence of a lipid-rich lesion core, which is always present in all our advanced plaque specimens from CEA.

We added the term “endarterectomy specimens” to the **Discussion** to acknowledge the reviewer’s point:

“Understanding ~~plaque~~ the morphology of these endarterectomy specimens is critical to improve clinical decision making for patients at risk for stroke or myocardial infarction”

16. SMCs, ECs and fibroblasts show up much better in single nuclei sequencing. Many studies used this approach, mitigating the bias. Please mention this and cite some studies.

We thank the reviewer for this comment. It is indeed true that single nuclei sequencing usually overcomes the immune cell bias getting introduced with single cell RNA-Seq. This is due to the methodology (freezing down and/or fixation of tissue before nuclei isolation) as utilized in previous studies (Chou et al., 2022; Song et al., 2024). Also, nuclei have similar sizes in different cell types and usually do not lead to clogging of the microfluidics system used for single cell RNA-Seq. Therefore, single nuclei sequencing can indeed reveal more realistic proportionality of cell types.

In this current study, we however decided to overcome this bias by introducing sc-bulk RNA-Seq deconvolution to carotid plaques. In addition to deconvolution, spatial transcriptomics using Xenium is unbiased when it comes to cell type proportions, as the tissues are fixed and all RNA of the plaque *in situ* equally accessible. Of note, single nuclei sequencing has the disadvantage of losing all cytoplasmic RNAs, which are important regulators of cell function. In our belief, the integration of all three transcriptomic sequencing methods proved to be a valid approach to better determine realistic cell type proportions in human atherosclerotic plaques.

Based on the reviewer's comment, we added a sentence in the **Discussion** section of our revised manuscript:

“Alternatively, this bias could be overcome by performing single nuclei sequencing, with the downside of losing cytoplasmic RNAs (Chou et al., 2022; Song et al., 2024).”

17. The authors write that “VSMC-rich media from our CEA specimens (Fig 1A), a structure that has likely not been analysed in previous studies“. Was there a difference in gene expression in the media of adjacent normal vs atherosclerotic artery tissue? Paired analysis should be possible.

We thank the reviewer for this very relevant suggestion. We indeed had the same thought and addressed this issue in a spatial proteomics study that is available as a preprint (Sinha et al., 2025) and currently in revision elsewhere. In this study, laser microdissection of the different subregions (media, NC, FC) was performed, and the proteomes of these subregions were analyzed individually. We did not follow this approach with spatial transcriptomics, as in particular the necrotic core of advanced plaques is rather poorly conserved for RNA expression analysis.

However, we extended our work to include the proposed paired DGE analysis of media cells. In the comparison of “Plaque vs. Control”, we identified three DEGs, ITLN1, PECAM1, and CD34, all of which were downregulated in media EC_2 cells from plaques compared to controls. Importantly, no other cell type exhibited any significant changes in gene expression between “Plaque vs. Control”. The results of this analysis, along with comparisons of additional condition variables within medial cells, are provided in **Suppl. Table 15** and **Suppl. Table 16**. We have also added a description of this analysis to the corresponding section in the Methods:

“To uncomplicate our analysis, we used subsets of data according to our scientific questions: for the comparison “Plaque vs. Control” only samples from Patients 1-4 were included as they had matching “Control” and “Plaque” samples, all other comparisons were performed across the 12 “Plaque” samples (Patient 1-12). For Patients 1-4 with available control and plaque samples, we also performed DGE analysis including their media-derived cells only, due to the media being present and proposed to be relatively similar in both control and plaque samples. DGE results can be found in Suppl. Table 2, Suppl. Table 3 and Suppl. Table 15 and Suppl. Table 16.”

18. “Four patients provided paired samples”. Were these bilateral endarterectomies, with one side worse than the other? If so, what was the time elapsed between the two surgeries?

We thank the reviewer for a chance to clarify our sampling approach. Please also see **Questions #5 and #7** by the same reviewer for additional explanation about how we addressed changes regarding our sample setup in the text for your clarification.

CEA-derived plaque specimens are cut into several pieces. The most advanced and diseased plaque piece, as well as the healthiest/less diseased piece from the same carotid artery are taken for downstream analysis to capture the dynamic of athero-progression.

We changed the **Methods** section accordingly:

“Four patients provided paired samples, meaning that one CEA specimen from the same carotid artery was divided into early and advanced lesions (Suppl. Fig. 3), while eight provided advanced plaques only.”

19. Page 6: “Correlation analysis revealed a strong inverse relationship between NC and lumen” suggests that the sum of the two may be constant and, by inference, the fibrous cap area % must also be constant.

We appreciate the comment of the reviewer. While we observed a strong inverse correlation between normalised NC and lumen areas, we didn’t find the sum of the two subregion areas, whether absolute or normalised, to be constant as also shown previously (Schoenhagen et al. 2003; Glagov et al. 1987; Schoenhagen et al. 2000). We have expanded the manuscript with a new figure showing these values plotted against each other (**Suppl. Fig. 12**, see below), both of which display considerable variation, and clarified the corresponding section in the **Results**:

*“Correlation analysis revealed a strong inverse relationship between normalised NC and lumen areas (Suppl. Fig. 12B), which could imply a constant sum, however, this was not observed (Suppl. Fig. 12). In contrast, absolute area correlations showed no association between NC and FC or intima, suggesting NC expansion reduces the relative proportions of other regions (**Suppl. Fig. 12A**).”*

Suppl. Fig. 13: Correlation between absolute and normalised summed areas of necrotic core and lumen. The sum of necrotic core and lumen absolute areas correlated with the sum of necrotic core and lumen normalised areas of plaques. R-squared and p-values come from a linear fit. NC: necrotic core.

20. The definition of controls is unclear. A sample with 45% intima is not a control. It seems that the authors call samples without NC controls.

We thank the reviewer for a chance to clarify this. Please also see **Questions #5, #7 and #18** by the same reviewer for additional explanation about how we addressed changes regarding our sample setup in the text for more clarity.

Intimal thickening is a natural phenomenon that occurs in almost all populations (especially Western) early in life. Control pieces in our study are indeed samples with no necrotic core, but showing early signs of intimal thickening and little to none immune cell infiltration. We carefully checked our manuscript again to make sure that we are not using the term “healthy” control, as these tissue specimens are clearly not completely healthy or undiseased arteries (Finn et al., 2010; Masawa et al., 1994; Pasterkamp et al., 1998; Velican, 1969; Virmani et al., 2000).

21. It is interesting that there was no difference between symptomatic and asymptomatic plaques. This should be emphasized. In many studies, this is the only distinction provided.

The reviewer is absolutely correct that most studies solely base their characterization and analysis on comparing symptomatic vs. asymptomatic patients. These studies might lack the thorough histomorphological characterization provided in this study. Furthermore, studies that identify differences in symptomatic vs. asymptomatic patients either focus on much larger datasets (for example 632 bulk RNA-seq samples (Mokry et al., 2022)) or solely on differences in immune cell composition (Fernandez et al., 2019). Spatial transcriptomics using Xenium with limited gene probe sets are likely unsuited to provide enough statistical power to distinguish symptomatic from asymptomatic patients. However, one has to also consider in this context that linking patient symptoms to plaque phenotypes (stable or unstable) is extremely challenging, as the correlation is weak or even completely absent (Libby et al., 2019). This is due to many factors being capable of inducing stroke or stroke-like symptoms, including paroxysmal atrial fibrillation with embolization, microcirculatory disruption, microvascular deformations, as well as hemorrhagic insults (Lavallée et al., 2023). In addition, the heterogeneous timing in symptomatic patients from clinical event to CEA surgery - and in many cases the sudden initiation of pharmacological treatment regimen (e.g., high-dose statins) can rapidly alter plaque morphology and cellular composition right before lesion removal.

Importantly, among all spatial transcriptomics studies of human plaques, none reported differences in symptomatic vs. asymptomatic patients. All previous studies either do not report symptoms at all, or solely use plaques from symptomatic patients. For the reviewer's convenience, we have summarized all human plaque studies using spatial transcriptomics approaches in the table below:

	Sun et al, JACC, 2023	Bleckwehl et al, NCVR, 2024	Lai et al, NCVR, 2025	Gastanadui et al, ATVB, 2024	Campos et al, EMBO Mol. Med., 2025	Pauli et al, under review, 2025
tissue [artery]	Carotid	Carotid + Coronary (for spatial only coronary)	Carotid	Coronary	Coronary	Carotid
bulk seq [sample size]	n=163	-	n=149 PTLOs only (no full tissue)	- (assumptions from spatial)	-	n=297
scSeq [patients]	-	n=13 (previous studies; no own data)	n=9	- (assumptions from spatial)	-	n=17
spatial method	VISIUM (FFPE; probe-based whole transcriptome)	VISIUM (FFPE; probe-based whole transcriptome)	StereoSeq (fresh frozen tissue; ~ VISIUM HD, capture based)	GeoMX	GeoMX & CosMX (~ Xenium)	Xenium

spatial samples [samples; therein controls]	n=6 (no controls)	n=12 (2 controls)	n=10 (no controls)	n=10 (no controls)	n=9 (2 controls)	n=16 (4 controls)
symptomatic/asymptomatic status	no report in bulk seq; spatial all symptomatic	no report	report in PTLO bulk seq; spatial all symptomatic	no report	no report	report in spatial

We added this finding to our revised **Results** section:

“We could not detect any DEGs by comparing advanced plaques of symptomatic (n=67) vs. asymptomatic (n=162) patients (FDR threshold < 0.05; minimum FDR in our analysis 0.13).”

22. Some patients showed “two NCs on each side of the lumen”. This is interesting, but not conclusive. They may be connected in another plane.

This is correct, as it cannot be excluded that in another plaque section only one large NC is present. In all spatial transcriptomics approaches, only snapshots of plaque morphology are captured and described.

We changed the text in our revised manuscript accordingly:

“They are less structured, showing two NCs on each side of the lumen (potentially one NC in another plane of the plaque; patient 4 and 8) or one large surrounding NC (patient 9)...”

23. In several scRNA-Seq studies and in at least 2 reviews, about half of the foam cells are SMC-derived and half are macrophages. There are two sets of macrophage foam cells and at least 3 other macrophage sets that are not foamy. The information obtained in the present study should allow these assignments.

In human data, it is unfortunately impossible to make conclusive statements on cellular origin. All work that confidently confirms SMC origin of macrophages (or macrophage-like cells) stem from mouse lineage tracing studies (Allahverdian et al., 2018; Harman & Jørgensen, 2019; Y. Li et al., 2021; Shankman et al., 2015; Wang et al., 2019). Both macrophages and SMCs are highly plastic cells that acquire features of one another in a disease context like atherosclerosis. Thus, our ‘human-only’ study does not allow for such interpretation. Currently, others are working on a holistic integration of several lineage tracing studies (Sharma et al., 2024). Progress has also been made on translating (deconvolution and label transfer) murine fate mapping into human pathology (D. Y. Li et al., 2025).

24. Some of the correlations with clinical and demographic parameters are not statistically analyzed.

Thank you for your remark. We have plotted the associations between patient descriptors (plaque morphology, patient metadata, and medical conditions) in **Suppl. Fig. 11** and **12**; however, not all parameters were statistically analyzed. Peripheral artery disease (PAD) was included in the differential gene expression analysis of the spatial transcriptomic data (**Suppl. Tables 2** and **3**) but was erroneously omitted from these supplementary figures. In addition, the captions of the two plots incorrectly suggested that COPD was one of the assessed clinical parameters. A previous version of the figure did include COPD, but we removed it since only one patient had COPD, rendering statistical analysis unfeasible. We apologize for these errors and have updated both figures to include PAD as an assessed parameter and corrected their captions accordingly. In addition, we split them into four figures to improve readability (see below), and corrected their references in the main text:

Associations between plaque morphology, patient metadata, and medical conditions showed distinct trends (~~Suppl. Fig. 11, Suppl. Fig. 12~~ Suppl. Fig. 14-17). Patient age inversely correlated with FC absolute area, height correlated positively with total plaque area (~~Suppl. Fig. 11A, 11F~~ Suppl. Fig. 14A, 15C), and hypertensive patients presented with increased normalized media areas (~~Suppl. Fig. 12D~~ Suppl. Fig. 17A). Plaque stability and symptoms (e.g., stroke or transient ischemic attacks) showed no correlation with subregion morphology.

Suppl. Fig. 14: Associations between absolute subregion areas with patient metadata, medical conditions, and plaque features. Y-axis depicts absolute areas of respective subregions: (A) fibrous cap (FC), (B) intima, (C) lumen. For numerical variables (age, height, BMI), R^2 and p-values are from linear fits; for categorical variables, Wilcoxon tests were used. Raw p-values were corrected within each subregion using the Benjamini-Hochberg method, reported as false discovery rate (FDR). Asympt.: asymptomatic; symp.: symptomatic; CAD: coronary artery disease; PAD: peripheral artery disease.

Suppl. Fig. 15: Associations between absolute subregion and whole sample areas with patient metadata, medical conditions, and plaque features. Y-axis depicts absolute areas of respective subregions: (A) media, (B) necrotic core (NC), (C) whole sample. For numerical variables (age, height, BMI), R^2 and p-values are from linear fits; for categorical variables, Wilcoxon tests were used. Raw p-values were corrected within each subregion using the Benjamini-Hochberg method, reported as false discovery rate (FDR). Asympt.: asymptomatic; symp.: symptomatic; CAD: coronary artery disease; PAD: peripheral artery disease.

Suppl. Fig. 16: Associations between normalised subregion areas with patient metadata, medical conditions, and plaque features. Y-axis depicts normalised areas of respective subregions: (A) fibrous cap (FC), (B) intima, (C) lumen. For numerical variables (age, height, BMI), R^2 and p-values are from linear fits; for categorical variables, Wilcoxon tests were used. Raw p-values were corrected within each subregion using the Benjamini-Hochberg method, reported as false discovery rate (FDR). Asympt.: asymptomatic; symp.: symptomatic; CAD: coronary artery disease; PAD: peripheral artery disease.

Suppl. Fig. 17: Associations between normalised subregion areas with patient metadata, medical conditions, and plaque features. Y-axis depicts normalised areas of respective subregions: (A) media, (B) necrotic core (NC). For numerical variables (age, height, BMI), R^2 and p-values are from linear fits; for categorical variables, Wilcoxon tests were used. Raw p-values were corrected within each subregion using the Benjamini-Hochberg method, reported as false discovery rate (FDR). Asympt.: asymptomatic; symp.: symptomatic; CAD: coronary artery disease; PAD: peripheral artery disease.

Also, mentioning a single SNP in the smoking section is not appropriate. SNPs were not tested in this study.

Yes, the reviewer is correct: SNPs were not analyzed in this study. We changed the text in our revised manuscript accordingly:

“... ATM was concordantly upregulated in smokers. This gene has previously been linked to lung cancer development by ~~its SNP rs189037 (G>A)~~, increasing the risk in non-smokers compared to smokers. ...”

25. Please define the difference between neighborhoods and subregions.

Thank you for your comment and the chance for clarification. We realised that using the word “subregions” in the **Methods** section describing the neighbourhood cluster analysis was unfortunate and confusing. We apologise for the mistake, and expanded the corresponding section in our **Methods** to further clarify the goal of neighbourhood clustering, as well as the differences between the neighbourhood clusters and subregions:

“We hypothesized that although the samples show substantial heterogeneity in their morphology, assessing the local neighbourhoods of cells could reveal similar ~~subregions~~ spatial domains across samples. Such domains might reflect shared biological processes or functions, potentially leading to a better understanding of plaque formation. Unlike histological subregions, which represent well-established anatomical compartments of plaques (i.e., lumen, intima, media, fibrous cap, necrotic core), we set out to identify in an unbiased manner patterns in local cell neighbourhoods. We applied unsupervised clustering of these neighbourhoods to generate clusters representing spatial domains that are characterised by similar cellular composition. Because the approach relies on local neighbourhoods and unsupervised clustering, the resulting clusters are independent of predefined subregion boundaries.”

We clarified the fact that subregions are based on established histological characteristics in comparison to the otherwise computational heavy manuscript in the **Methods** section:

“The annotation of sample subregions is a histological analysis, using established Oxford Plaque Studies^{18–20} and AHA criteria^{78–81}, and was performed by an experienced cardiovascular pathologist.”

26. Figure 2E: the choice of colors is unfortunate. Impossible to see the cluster calling.

We thank the reviewer for this valuable comment. To improve visibility in **Figure 2E**, we increased the dot size in the UMAP plots and selected a color palette with higher contrast to better distinguish cell clusters.

Please see **Question #11** for the improved **Figure 2**.

27. Figure 3D: what are “inflammatory mixed cells”? Doublets?

We thank the reviewer for this comment and the chance to further clarify this aspect. A detailed rationale behind the annotation of “Inflammatory mixed cells” is included in the revised **Methods** section. To further clarify our hypothesis regarding the nature of these cells and their possible relationship to doublets, we have expanded this section of the revised manuscript. For your convenience, we include the revised section with the additional clarification highlighted below:

“Based on these features we decided to annotate the cell cluster as “Inflammatory mixed cells”. We hypothesize that these cells could reflect cell fragments, whose cytoplasm was only partly contained in the plane of the histology sample, although they can also be a result of incorrect cell segmentation. Although it is possible that these cells represent doublets, we consider this unlikely. Potential doublets arising from cell segmentation were excluded using Baysor’s “max cluster frac” parameter, as recommended by the authors, and the low transcript counts observed in these cells further argue against the doublet interpretation.”

28. Figure 3E: Age seems to be a driver for necrotic core. This should be emphasized.

We thank the reviewer for this observation. We assessed the relationship between normalised necrotic core area and patient age, which revealed a weak positive trend. However, as this association was not statistically significant (**Suppl. Fig. 12E**, please see **Question #24** for the figure), we did not include it in the main manuscript. Assessment in a larger patient cohort seems necessary to determine whether this trend represents a genuine biological association.

29. Figure 5: What statistical test was used to assign the morphological clusters?

We thank the reviewer for this comment, and have revised and expanded the **Methods** section to provide additional details on the clustering approach and its implementation:

~~“First, we calculated the cell fraction percentage of the different cell type substates normalised within the sample, then performed hierarchical clustering with Euclidean distance and Ward’s-linkage method.~~
First, we calculated the cell fraction percentage of the different cell type substates normalised within the sample. This enabled us to capture finer differences in cellular composition across all plaques. Next, we performed hierarchical clustering of the samples based on these normalized cell type fractions, using Euclidean distance and Ward’s-linkage method. This is an

unsupervised clustering method and does not rely on a predefined statistical test for determining group assignments.”

How do we know there should be 5 clusters? Or is this arbitrary?

We also revised the paragraph describing how the final number of clusters was chosen:

“We agreed on the final number of clusters (4) based on the clustermap and the individual inspection of the samples. ...The final number of clusters (4) was selected based on cluster robustness (Fig. 5) and further supported by visual inspection of the samples (Fig. 6). Based on the dendrogram structure shown at the top of the clustermap in Fig. 5, selecting four clusters provided a biologically meaningful and robust separation between groups of samples, while at the same time, samples within each cluster displayed consistent patterns in their spatial morphology (Fig. 6) as described in the “Morphological plaque clustering” subsection in Results. These considerations together guided our choice of four clusters.”

30. Figure 7C: How were patients 4 and 8 selected?

We thank the reviewer for the comment and clarified the rationale for our selection of samples included in the main figures. We have added a paragraph to the **Methods** section:

“...To showcase the macrophage neighbourhoods, we selected plaques from patients 4 and 8 for inclusion in the main figures, as these contained macrophage clusters with the most illustrative spatial patterns. In addition, these samples also contained EC neovessel neighbourhoods, which are also highlighted in the main text. To illustrate VSMC neighbourhoods, we selected samples with a cell-dense and intact media (plaques from patients 7 and 10, and the control from patient 3) for the main figures.”

Reviewer #2 (Remarks to the Author):

Overall Summary:

Pauli et al. present their investigation seeking to integrate multiple transcriptomic methodologies (bulk RNA seq, scRNAseq, and spatial transcriptomics) as a means to molecularly define carotid atherosclerotic disease. Using arterial and atheroma samples collected during CEA, the investigators attempt to elucidate a spatially-resolved molecular architecture underlying plaque development. The authors highlight unique plaque morphologies linked to different cell type compositions, disease heterogeneity (i.e. symptomatology), and clinical demographics (i.e. smoking, etc) with inference towards plaque development and stability. Notably, they uncover spatial associations between HMOX1+ macrophages as they potentially transition to foam cells (TREM2+).

While the manuscript is well written and demonstrates technical rigor, the most important limitation is a lack of methodologic novelty or conceptual impact to our molecular understanding of atherosclerosis. Notably, HMOX1 and TREM2 macrophages are well established in the progression of atherosclerosis (PMID: 39731912 – HMOX1, 38974464 – TREM2) and modulation of macrophage polarity (PMID: 30282830 – HMOX1, 38646596 – TREM2). Furthermore, other spatial transcriptomic investigations of arterial atherosclerosis have recently been published (PMID: 39234691 and 39715784). Considering these limitations, the current manuscript offers incremental advancement.

We appreciate the assessment by the reviewer of our current work. We have tried in the revision to clarify the intention, novelty, and relevance for atherosclerosis and carotid artery disease of our study. We fully agree with the reviewer on his comments regarding macrophage biology, their transition and modulation in human atherosclerosis. We still believe that our current study adds value to the existing macrophage knowledge by spatially locating (with single cell resolution) their subsets in early and advanced lesions. Transition from TREM2 high to CCL18 high for example can be specifically located to the highly relevant plaque shoulder region of advanced lesions. Unstable carotid artery plaques become prone to rupture in this specific area where this macrophage transition occurs.

Our study is not limited to macrophage biology, as it determines endothelial cell (EC) heterogeneity (angiogenic and EndoMT-like), spatial distribution of these EC subtypes within the plaque, as well as their highly specific expression patterns in early vs. advanced lesions. Angiogenic ECs colocalize with immune cells in areas of the necrotic core, likely providing an entry gate for immune cells that destabilize the plaque. Similar to ECs, various smooth muscle cell (SMC) subclusters surfaced in our study, suggesting specific spatial differentiation patterns between early and advanced carotid plaques.

Our study provides further novelty and disease relevance through the comparison of early vs. advanced lesions in a paired manner (meaning that the plaque

and control lesion stem from the exact same patient), which has not been analyzed with (multi-)transcriptomics before. We created a new Supplemental Figure 3 to explain our approach in sampling and our unique setup of plaque vs. control from the same patient better:

Suppl. Fig. 3: Carotid artery plaque tissue sectioning and characterisation routine. (A) shows the whole plaque upon surgical removal (CEA). (B) shows the sectioned plaque getting processed for molecular and histomorphological characterisation. The arrows indicate the most advanced and earliest piece of tissue in this example. (C) Histological analysis is performed according to the American Heart Association (AHA) classification guidelines (Stary 2000; Stary et al. 1992; Stary et al. 1995; Stary et al. 1994) using different histological stainings (HE and EvG). FC thickness is evaluated based on the Oxford Plaque Study (Redgrave et al. 2008) and measured using the AxioScan Software (Leica) or QuPath. Here, all four measured positions of the FC are below 200µm, therefore this carotid plaque was classified as unstable.

In addition to being the first study ever to perform Xenium spatial transcriptomics on human carotid artery plaques, the integration of all three transcriptomic sequencing modalities enables us to provide a comprehensive view on the dynamic of atherosclerotic lesion progression and destabilization (*via* comparing early to advanced lesions from the same patient).

The table provided below compares all performed studies to date (six in total, including our own study reviewed here) using varying spatial transcriptomic technologies. The reviewer mentioned two spatial transcriptomics studies (PMID: 39234691 and 39715784) that solely analyzed coronary arteries, and not plaques from carotids. Of note, none of the other four publications provide a full evaluation and integration of bulk, single cell and spatial transcriptomics. Our study offers by far the largest bulk RNA-Seq dataset with 297 samples, as well as the richest single-cell RNA-Seq dataset ($n=17$ patients) and the highest number of samples analyzed with spatial transcriptomics ($n=16$; including 4 controls/early lesions) of all five studies. We are furthermore the only study that can compare plaques from symptomatic vs. asymptomatic patients in spatial transcriptomics, as all other studies either do not report symptom status - or only use plaques from symptomatic individuals:

	Sun et al, JACC, 2023	Bleckwehl et al, NCVR, 2024	Lai et al, NCVR, 2025	Gastanadui et al, ATVB, 2024	Campos et al, EMBO Mol. Med., 2025	Pauli et al, under review, 2025
tissue [artery]	Carotid	Carotid + Coronary (for spatial only coronary)	Carotid	Coronary	Coronary	Carotid
bulk seq [sample size]	n=163	-	n=149 PTLOs only (no full tissue)	- (assumptions from spatial)	-	n=297
scSeq [patients]	-	n=13 (previous studies; no own data)	n=9	- (assumptions from spatial)	-	n=17
spatial method	VISIUM (FFPE; probe-based whole transcriptome)	VISIUM (FFPE; probe-based whole transcriptome)	StereoSeq (fresh frozen tissue; ~ VISIUM HD, capture based)	GeoMX	GeoMX & CosMX (~ Xenium)	Xenium
spatial samples [samples; therein controls]	n=6 (no controls)	n=12 (2 controls)	n=10 (no controls)	n=10 (no controls)	n=9 (2 controls)	n=16 (4 controls)

symptomatic/ asymptomatic status	no report in bulk seq; spatial all symptomatic	no report	report in PTLO bulk seq; spatial all symptomatic	no report	no report	report in spatial
---	--	-----------	--	-----------	-----------	-------------------

Other Criticisms include:

1. Association with symptomatology / stability is uncertain. One area of potential significance was the assertion that the current findings molecularly define symptomatic (vs. asymptomatic) carotid artery stenosis. As presented, however, the data do not clearly support these claims. Figure 5 provides an example of this missed opportunity. A clearer presentation of the results would have been based upon symptom status as opposed to various morphologic clusters.

We thank the reviewer for the chance to clarify our approach in carotid artery disease and human CEA specimens. We do not primarily characterize plaques based on patients' symptoms, despite symptoms of course being recorded and taken into account as a key clinical parameter. Most studies analyzing plaques solely based on patient symptoms have the limitation that they lack histomorphological stratification of carotid plaque biology and stability.

As aforementioned in response to Reviewer 1 (comment # 21), correlation of patient symptoms with plaque stability is weak and mostly absent in carotid artery disease (Libby et al., 2019). This well-established missing link between patient symptoms and plaque stability is a critical topic in stroke prevention research and clinical decision-making (Gupta et al., 2013; Saba et al., 2025; Underhill et al., 2010). In asymptomatic patients, the degree of carotid artery stenosis and plaque burden have traditionally guided treatment (e.g., surgery vs. medical therapy). But it has become increasingly clear that biological plaque characteristics - not just the severity of the narrowing caused by the plaque - play the most pivotal role in determining stroke risk (Libby et al., 2019; Paraskevas & AbuRahma, 2024). The main reason behind the weak linkage of symptoms with stability in carotid artery disease relates to the fact that a magnitude of factors can ultimately cause stroke and TIA (Lavallée et al., 2023). As aforementioned, additional factors to the here studied unstable carotid plaques include cardiac embolization due to accompanying heart diseases (dilative cardiomyopathies, atrial fibrillation), hemorrhagic events, vascular malformations, and microcirculatory disruption (Lavallée et al., 2023). In addition, the heterogeneous timing in symptomatic patients from clinical event to CEA surgery - and in many cases the sudden initiation of pharmacological treatment regimen (e.g., high-dose statins) can rapidly alter plaque morphology and cellular composition right before lesion removal potentially skewing results and interpretations thereof.

This missing link separates carotid artery disease from coronary artery disease, where the identification of the culprit lesion within the coronary system causing

myocardial ischemia can be easily distinguished (via CT, MRI, angiographical or intravascular imaging, electrocardiogram/ECG, and finally histopathology; (Kubo et al., 2007)).

In response to the reviewer's comment, we assessed the association between plaque stability and patients' symptoms. We pooled data across all methods for 252 patients in total (Xenium $n=12$ plaques, scRNA-seq $n=17$ plaques, bulk RNA-seq $n=223$ plaques). This was feasible because there were no overlapping patients across the different datasets. We then constructed a contingency table based on the combined data and performed Fisher's exact test, which indicated no significant association between plaque stability and symptomatic status (odds ratio = 1.144, p -value = 0.67). This confirms the above-described, well-established missing link between symptoms and plaque stability in carotid artery disease.

	asymptomatic	symptomatic
stable	78	30
unstable	100	44

Due to this, we and others (Bashore et al., 2024; Dib et al., 2023; Fidler et al., 2024; P. Singh et al., 2024; Sun et al., 2023) believe that it is highly relevant to characterize plaques histomorphologically, and to use plaque biology (e.g., fibrous cap thickness) as the main category to understand changes in lesion stability. Future developments into AI-driven digital pathology will hopefully provide enhanced tools to discriminate different plaque phenotypes from histopathology. But these approaches are still under development and have yet to be implemented into CEA biobank routines. Still, the comparison of early vs. advanced plaque in our study creates an additional useful resource for this previously lesser understood context of human atheroprogession.

2. Surprising lack of DEG in bulk RNAseq. Figure 2c shows relatively few differences in DEG across stable and unstable plaques. This is in contrast to current literature (39234691), yet no explanation is given.

We appreciate the comment by the reviewer and the chance to put our results into context with previous studies. The mentioned study (PMID: 39234691) was based on a small sample size (5 stable vs. 5 unstable plaques) and was potentially biased by selective regional sampling, limiting the generalizability of its findings and increasing the chance of false positive findings. In contrast, our analysis of transcriptomes from 126 unstable vs. 97 stable plaques provides a more robust assessment, with greater power and reduced susceptibility to sample-, technical-, and region-specific biases. The revised manuscript's **Result** section has been updated to follow the suggestion of

the reviewer to comment on the low count of DEGs when comparing stable and unstable plaque:

“While a previous study (Gastanadui et al., 2024) identified 107 DEGs between stable and unstable coronary plaques (n=10 in total), our bulk transcriptomes of 223 carotid plaques identified only four DEGs, including CXCL5 and PI3K (Fig. 2C). The lower DEG count in our study reflects lower susceptibility to sample-, technical-, and regional-specific biases. In addition, plaque stability, when defined solely by FC thickness as a key predictor of clinical outcomes, seemed to not be primarily driven by transcriptional differences at the whole-tissue level. We could further not detect any DEGs by comparing plaques from symptomatic (n=67) vs. asymptomatic (n=162) patients (FDR threshold < 0.05; minimum FDR in our analysis 0.13).”

3. Underpowered study with significant demographic heterogeneity. It is difficult to interpret some of the results due to relatively few patients (N=4 Ctl) and variable clinical demographics.

We understand the concern of the reviewer. It is indeed true that more samples, especially more controls would be needed to draw stronger conclusions. Still, our dataset is the largest spatial transcriptomics dataset of human plaques. Apart from the studies by (Bleckwehl et al., 2025) and (Campos et al., 2025), our study is the only one that utilizes controls, and therefore the only study that shows control samples for carotid arteries (as Bleckwehl *et al.* as well as Campos *et al.* solely studied coronary arteries). In comparison to all other spatial transcriptomics studies, we further provide the highest number of additional (and integrated) bulk and single cell RNA-Seq data from the same CEA patient cohort now getting released with this manuscript. Furthermore, we are the first ones to perform a targeted approach of spatial transcriptomics, including long non-coding RNAs, which are highly relevant markers for cell substate detection, while playing significant roles in plaque progression and vulnerability.

For details in study design and provided data of all current spatial transcriptomics studies, please refer to the **Table provided in response to the introductory comment** above.

4. Biased spatial transcriptomic approach based off Panels 1 and 2. Finally, the spatial transcriptomic analysis used a custom (pre-defined) gene set (~500) for cellular and disease status identification. While this approach is helpful in resolving cellular spatial location, it falls short of providing true spatially-resolved, single cell transcriptomic analysis. This undercuts the novelty.

In our current manuscript, spatial transcriptomics was not used as a discovery platform (for which the two Xenium panels would indeed be suboptimal given the other alternatives). Our aim was to spatially locate and integrate (with the highest available resolution) cell types, their substates, and genes that surfaced from our bulk and single cell RNA-Seq studies. Thus, the Xenium panels are based on markers from single cell RNA-Seq data, enabling us for the first time to locate cells and genes in the same tissues (comparing early vs. advanced lesions). Furthermore, Xenium allowed us to locate non-coding RNAs in FFPE tissue currently not included in Visium. As aforementioned, long non-coding RNAs play a significant role in determining cell substates based on their level of cell specificity. Visium (without HD; only available option at the time we initiated our study in August 2023) loses about 60-70% of the tissue area due to the spot anatomy. One spot has the size of 55 μ m, leading to more than 1 cell (rather 5-10 cells) being sequenced.

Of note, our manuscript introduces Baysor to achieve single cell spatial resolution during the analysis of human atherosclerotic plaques. Compared to Visium (HD and no HD), probe-based Xenium provides subcellular resolution (~0.5 μ m/pixel), whereas even Visium HD 'only' provides a resolution of ~5 μ m/pixel. For a complex disease like carotid artery disease, the highest possible resolution seems relevant for cell type detection and intercellular communication due to the complex and heterogeneous nature of these human tissue specimens.

Reviewer #3 (Remarks to the Author):

Summary

This manuscript presents a comprehensive spatially resolved transcriptomic analysis of human carotid atherosclerotic plaques. The authors integrate bulk RNA-Seq, single-cell RNA-Seq (scRNA-Seq), and single-cell spatial transcriptomics (Xenium platform) on paired samples from patients undergoing carotid endarterectomy (CEA). Their multimodal approach reveals spatial microenvironments, including distinct VSMC and macrophage subtypes associated with plaque vulnerability, progression, and smoking status. They propose a model of subluminal HMOX1⁺ macrophage transdifferentiation into lipid-handling TREM2⁺ cells. A novel morphological clustering of plaques into “structured” and “chaotic” types is introduced, offering mechanistic insights into spatial immune-VSMC interplay and plaque destabilization.

Major Comments

1. Justification of Technology Choice: The authors employ the Xenium platform for spatial transcriptomics but do not justify its selection over other platforms like 10x Visium HD. A justification regarding resolution, sensitivity, and gene coverage tradeoffs is needed.

Visium HD was released in March 2024, whereas our Xenium runs started in August 2023. And Visium (without HD; only available option back then) loses about 60-70% of the tissue area due to the spot anatomy. Also one spot has the size of 55µm, leading to 5-10 cells getting simultaneously sequenced. Another reason why we chose Xenium over Visium is the fact that Visium for FFPE is a probe based assay that can only establish data for protein coding genes. We also wanted to measure long-non coding RNAs, as they are important markers of cellular substates. Visium for fresh tissue (not fixed) did not work in our hands despite multiple trials using our plaque specimens due to tissue detachment from the slide.

Our goal was to utilize Xenium, as it reaches subcellular resolution (~0.5µm/pixel) compared to Visium's 55µm spot diameters. Of note, even the new Visium HD platform 'only' provides a resolution of about ~5µm/pixel, so 10x lower compared to Xenium.

The sensitivity (transcript detection/cell) is reported to be higher with Xenium (up to 5000 transcripts/cell) compared to Visium and Visium HD. Of note, our atherosclerotic plaques do not reach this sensitivity due to certain tissue specifics (matrix and lipid deposition, calcifications and scar tissue, multiple apoptotic cells and debris thereof).

Gene coverage is obviously higher in Visium compared to Xenium, as Visium spans 18,000 genes. This appears ideal for discovery-based approaches. However, the focus of our study was the integration of bulk, single cell and spatial transcriptomics from the same tissue, which is why we followed a more targeted spatial transcriptomics

approach using Xenium. Hence, the higher resolution of Xenium became our weapon of choice for the more integrative, and less discovery-driven approach.

2. Immune Cell Overrepresentation Not Corrected: Despite deconvolution, the overrepresentation of immune cells in scRNA-Seq is not adequately addressed in the spatial analysis. This may bias subregion interpretations.

Despite the same amount of marker genes per cell type (on average 5 genes/cell type) in our Xenium probe set, we were able to confirm the higher account of structural cells compared to immune cells with spatial transcriptomics. Of importance, probes for marker genes for Xenium were selected based on single cell RNA-Seq results (which provided the immune cell overrepresentation). Furthermore, Xenium confirmed single cell-bulk RNA-Seq deconvolution (depicted in **Fig. 1G** and **Fig. 2B**).

We further clarified this aspect in our revised manuscript (**Results**) and added the following sentence:

“In addition to this deconvolution approach, Xenium-based spatial transcriptomics (designed with an equal amount of probes for all cell types) similar to bulk RNA-Seq confirmed that the observed immune cell dominance (Depuydt et al., 2023; Dib et al., 2023; Fernandez et al., 2019) solely occurred in our scRNA-Seq study.”

3. Smoking Association Requires Caution: The observed association with smoking is only partially supported by bulk and scRNA-Seq data. Acknowledgment of limited reproducibility is needed.

We agree with the reviewer that more samples in Xenium would be needed to make more cautious statements on the association between smoking and plaque morphology. We therefore tried to validate our findings with bulk- and scRNA-Seq, but this data did not fully support our findings.

We further clarify this now and acknowledge the limited reproducibility.

“However, we did not observe reproducible differences in VSMC cell fractions across smoking status with either bulk- or scRNA-Seq, which reflects rigor in assessing reproducibility of our findings (Suppl. Fig. 23A,B).”

4. Trajectory Analysis of Macrophages: The proposed HMOX1⁺ to TREM2⁺ trajectory should be validated with orthogonal methods beyond TFvelo and visualized more explicitly.

Thank you for your valuable comment. Since TFvelo infers cell trajectories based on transcription factor activity, we considered an orthogonal approach using methods that rely on spliced and unspliced mRNA counts. Because our original alignment excluded intronic reads, calculation of spliced/unspliced counts was not feasible. We therefore repeated the alignment including intronic reads and subsequently obtained

spliced/unspliced counts using Velocyto. We chose to infer trajectories with scVelo, an established splicing count-based method. In addition to pseudotime and velocity streamplots, which display trajectories on a cellular level, we also aimed to visualise the trajectories on the cell type level. For this, we created PAGA-graphs, which show velocity-directed edges between cell types, and plotted pseudotime distribution within macrophage subtypes of interest (**Suppl. Fig. 41**). As the realignment altered our count matrix (details provided below), we also reran TFvelo on the dataset generated with intronic reads included (updated **Suppl. Fig. 40**). Furthermore, to provide an additional layer of analysis, TFvelo was also applied to our Xenium datasets, as spliced/unspliced counts cannot be derived with this methodology. The results are presented in **Suppl. Fig. 36-39** below, and due to space limitations, in **Suppl. PDF 3,4**. TFvelo analysis of the Xenium datasets revealed a clear ordering of the three macrophage subtypes of interest along pseudotime, with mean pseudotime values increasing in the direction Mac_C1Q → Mac_HMOX1 → Mac_TREM2hi in each sample (**Suppl. Fig. 36, 37**). This pattern was also evident in spatial scatterplots coloured by pseudotime and subtype (**Suppl. PDF 3,4**). PAGA-graphs showed strong edges with the same directionality (**Suppl. Fig. 38C, Suppl. Fig. 39C**), although the link between Mac_TREM2hi_CCL18hi and Mac_TREM2hi was weak with panel 1. While not fully concordant across all macrophage subtypes, both TFvelo and scVelo consistently inferred trajectories from HMOX1 towards TREM2hi cells in velocity stream plots and PAGA graphs, and both showed lower mean pseudotime values for *HMOX1+* cells compared to *TREM2hi* cells (**Suppl. Fig. 40; Suppl. Fig. 41**). These results could suggest the existence of a transdifferentiation route from *HMOX1+* toward *TREM2hi* subtypes, however further experimental proof is necessary.

We included the changes in the respective **Results** section of our revised manuscript:

“...Based on our spatial and DEG results, we hypothesize a gradual transdifferentiation of HMOX1+ into TREM2hi with Mac_TREM2hi_CCL18 as an intermediate, transient state. ~~This hypothesis was corroborated by RNA velocity analysis of our scRNA-Seq data, suggesting a transcriptional trajectory over time from C1Q+ subtype towards TREM2hi via HMOX1+ (Suppl. Fig. 31). We assessed this with velocity analysis using TFvelo (J. Li et al., 2024), which showed increasing mean pseudotime along subtypes Mac C1Q → Mac HMOX1 → Mac TREM2hi CCL18hi (panel 1) → Mac TREM2hi (Suppl. Fig. 36, 37), accompanied by a lumen-to-NC gradient in patients 4, 7, 8, and 10 (Suppl. PDF 3,4). Partition-based graph abstraction (PAGA) supported the same progression (Suppl. Fig. 38C, 39C), though the link between Mac TREM2hi CCL18hi and Mac TREM2hi was weak in panel 1. Despite not being fully concordant across all subtypes, complementary analyses of our scRNA-Seq data with two orthogonal methods (TFvelo, scVelo (La Manno et al., 2018)) yielded consistent trajectories from HMOX1+ to TREM2hi cells and lower pseudotime values for HMOX1+ relative to TREM2hi (Suppl. Fig. 40;~~

Suppl. Fig. 41), suggesting a possible transdifferentiation route that requires future experimental validation. ...”

We also revised the corresponding section in the **Discussion**:

“... In a subset of plaques C1Q⁺ and HMOX1⁺ reside close to the lumen, while TREM2^{hi} macrophages border and harbour the NC. Interestingly, in a control sample, all three substates appear subluminally, without TREM2^{hi} cells separating. RNA velocity analysis revealed a transcriptional trajectory over time from C1Q⁺ subtype towards TREM2^{hi} via HMOX1⁺. Of note, a similar HMOX1⁺ → TREM2^{hi} trajectory, along with HMOX1⁺ and C1Q⁺, though occurring in opposite direction along with pseudotime was described before (Dib et al., 2023). We report a gradual transdifferentiation from HMOX1⁺ macrophages via an intermediate phenotype (Mac_TREM2^{hi}_CCL18) that reduces CCL18 expression while increasing SPP1, MMP9, MMP12. Finally, these cells become TREM2^{hi}, probably when reaching the crucial shoulder region of the FC. This sequence of cells and associated DEGs seem highly orchestrated, based on the repetitive layering and positive correlation of HMOX1⁺ macrophages with TREM2^{hi}. We hypothesise a gradual transdifferentiation from HMOX1⁺ macrophages via an intermediate phenotype (Mac TREM2^{hi} CCL18^{hi}) that reduces CCL18 expression while increasing SPP1, MMP9, MMP12. Finally, these cells become TREM2^{hi}, probably when reaching the crucial shoulder region of the FC. This sequence of cells and associated DEGs seem highly orchestrated, based on the repetitive layering and spatial correlation patterns of HMOX1⁺ macrophages with TREM2^{hi}. Xenium RNA velocity analysis corroborated this by revealing a transcriptional trajectory over time from C1Q⁺ subtype towards TREM2^{hi} via HMOX1⁺, with the HMOX1⁺ → TREM2^{hi} transition also supported by our scRNA-seq data, as previously reported (Dib et al., 2023). Although intriguing, these findings require further experimental validation. ...”

We included the following changes in the respective **Methods** sections:

“... Alignment of the reads was performed with cellranger’s count function with default parameters, which included intronic reads, except for setting “include_introns” parameter to “false” to exclude intronic reads.”

“...For the calculation of velocity stream plots, we only included genes with a larger latent time - pseudotime Spearman correlation of 0.4. Pseudotime and velocity streamplots (**Suppl. Fig. 40A,B**) were created using the scPoli-UMAP representation.

In addition to the cellular-level trajectory information provided by the streamplots, we also aimed to assess trajectories at the cell type level. To this end, we applied the PAGA (partition-based graph abstraction) algorithm (Wolf

et al., 2019) using the inferred TFvelo pseudotime as priors and the macrophage low-level cell types as node groups. The resulting graph, displaying velocity-directed edges between cell types, is shown in **Suppl. Fig. 40C**. We also plotted the TFvelo pseudotime distribution of the three macrophage subtypes (Mac_C1Q, Mac_HMOX1, and Mac_TREM2hi; **Suppl. Fig. 40D**), which showed ordered spatial distribution patterns in the Xenium data, to better visualize their relative ordering along the TFvelo pseudotime axis.

To complement TFvelo with an orthogonal RNA velocity approach, we also applied a splicing count-based method to infer cell trajectories. Spliced and unspliced expression counts were obtained using Velocyto (La Manno et al., 2018) on the .bam files and filtered barcodes generated during alignment with Cell Ranger's count function.

For the splicing count-based analysis, we used scVelo (Bergen et al., 2020). We first created a subset of our data containing only macrophage cells (1,837 cells) and calculated first- and second-order moments with `scvelo.pp.moments()` using the scPoli representation, with the number of neighbours set to 30. Next, we estimated velocity vectors with scVelo's dynamical model. Inferred pseudotime and velocity stream plots were generated using the scPoli-UMAP representation (**Suppl. Fig. 41A,B**). A PAGA graph based on pseudotime inferred by scVelo is shown in **Suppl. Fig. 41C**, together with the scVelo pseudotime distribution across the three macrophage subtypes (**Suppl. Fig. 41D**)."

"RNA velocity analysis of macrophages in Xenium

To further explore the potential relationship between the spatial distribution and expression patterns of macrophage subtypes observed in the Xenium data, we assessed their trajectories using RNA velocity analysis. For this purpose, we applied TFvelo (J. Li et al., 2024), a transcription factor (TF) based model, since spliced and unspliced RNA counts cannot be derived from Xenium due to its in situ hybridization methodology.

We subsetted our dataset to include only macrophages (panel 1: 35,265 cells; panel 2: 28,747) and performed preprocessing with the parameters recommended by the authors, using the batch-corrected scVI-UMAP representation as the low-dimensional embedding and acosh-normalised counts. We chose acosh normalisation because some cells had erroneously small area estimates from Baysor, which led to extreme values in area-normalized counts. These outliers interfered with TFvelo and prevented reliable trajectory inference. For the calculation of velocity stream plots, we only included genes with a larger latent time-pseudotime Spearman correlation of 0.3. Pseudotime and velocity streamplots are shown in **Suppl. Fig. 38A,B** for panel 1, and in **Suppl. Fig. 39A,B** for panel 2. Similar to our trajectory analysis of our scRNA-seq data, we constructed PAGA graphs for both panels using TFvelo-inferred pseudotime as priors and macrophage low-level cell types as

node groups (panel 1: **Suppl. Fig. 38C**; panel 2: **Suppl. Fig. 39C**). For panel 1, we included the cell substate *Mac TREM2hi CCL18hi* into the construction of the PAGA graph, as well as in the colouring of cells in the streamplot. Leveraging available cell coordinates, we plotted spatial scatterplots of the individual samples with macrophages coloured by *TFvelo* pseudotime and cell type respectively (**Suppl. PDF 3,4**).

Of note, macrophage cells with panel 2 dataset expressed more TFs (total number of TFs: 12; mean number of TFs/gene: 5.3), than panel 1 (total number of TFs: 5; mean number of TFs/gene: 1).

For your convenience, we included the updated **Suppl. Fig. 40** and newly created **Suppl. Fig. 41, Suppl. Fig. 38, Suppl. Fig. 39, Suppl. Fig. 36, Suppl. Fig. 37** as shown below:

Suppl. Fig. 40: RNA velocity of macrophage subtypes in scRNA-seq data with TFvelo. (A) Macrophage cells coloured by pseudotime inferred by TFvelo. TFvelo analysis was performed using all macrophage cells (2805 cells) in dataset. Pseudotime reflects the progression of cell development, starting from 0 to 1. (B) RNA velocity stream plot. The arrows depict the transcriptional trajectory of the cells over pseudotime inferred by TFvelo based on cell expression data. (C) PAGA (partition-based graph abstraction) graph showing inferred connectivity between cell types. Nodes represent cell types, and edges indicate transcriptional transitions based on scvelo velocity. Solid edges reflect high-confidence transitions, while dashed edges indicate weaker or less certain connections. Node size corresponds to the number of cells per group. Dot colouring in (B-C) corresponds to macrophage subtypes. (D) Distribution of TFvelo pseudotime (y-axis) across three macrophage cell subtypes along the proposed transdifferentiation trajectory Mac_C1Q → Mac_HMOX1 → Mac_TREM2hi (x-axis). Comparisons of pseudotime across subtypes were performed using the Wilcoxon rank-sum test, and raw p-values were adjusted with the Benjamini-Hochberg method to obtain false discovery rate (FDR) values.

A

scvelo - Pseudotime

B scvelo - RNA velocity stream

C scvelo - PAGA graph

D

Suppl. Fig. 41: RNA velocity of macrophage subtypes in scRNA-seq data with scvelo. (A) Macrophage cells coloured by pseudotime inferred by scvelo. Scvelo analysis was performed using macrophage cells with available splicing count data (1834 cells). Pseudotime reflects the progression of cell development, starting from 0 to 1. (B) RNA velocity stream plot. The arrows depict the transcriptional trajectory of the cells over pseudotime inferred by scvelo based on cell expression data. (C) PAGA (partition-based graph abstraction) graph showing inferred connectivity between cell types. Nodes represent cell types, and edges indicate transcriptional transitions based on scvelo velocity. Solid edges reflect high-confidence transitions, while dashed edges indicate weaker or less certain connections. Node size corresponds to the number of cells per group. Dot colouring in (B-C) corresponds to macrophage subtypes. (D) Distribution of scvelo pseudotime (y-axis) across three macrophage cell subtypes along the proposed transdifferentiation trajectory Mac_C1Q → Mac_HMOX1 → Mac_TREM2hi (x-axis). Comparisons of pseudotime across subtypes were performed using the Wilcoxon rank-sum test, and raw p-values were adjusted with the Benjamini-Hochberg method to obtain false discovery rate (FDR) values.

Suppl. Fig. 38: RNA velocity of macrophage subtypes in Xenium panel 1 data with TFvelo. (A) Macrophage cells coloured by pseudotime inferred by TFvelo. TFvelo analysis was performed using all macrophage cells in panel 1 dataset (35265 cells). Pseudotime reflects the progression of cell development, starting from 0 to 1. (B) RNA velocity stream plot. The arrows depict the transcriptional trajectory of the cells over pseudotime inferred by TFvelo based on panel 1 gene expression data. (C) PAGA (partition-based graph abstraction) graph showing inferred connectivity between cell types in panel 1 data. Nodes represent cell types, and edges indicate transcriptional transitions based on TFvelo velocity. Solid edges reflect high-confidence transitions, while dashed edges indicate weaker or less certain connections. Node size corresponds to the number of cells per group. Dot colouring in (B-C) corresponds to macrophage subtypes.

A

TFvelo - Pseudotime

B TFvelo - RNA velocity stream

C TFvelo - PAGA graph

Suppl. Fig. 39: RNA velocity of macrophage subtypes in Xenium panel 2 data with TFvelo. (A) Macrophage cells coloured by pseudotime inferred by TFvelo. TFvelo analysis was performed using all macrophage cells in panel 2 dataset (28747 cells). Pseudotime reflects the progression of cell development, starting from 0 to 1. (B) RNA velocity stream plot. The arrows depict the transcriptional trajectory of the cells over pseudotime inferred by TFvelo based on panel 2 gene expression data. (C) PAGA (partition-based graph abstraction) graph showing inferred connectivity between cell types in panel 2 data. Nodes represent cell types, and edges indicate transcriptional transitions based on TFvelo velocity. Solid edges reflect high-confidence transitions, while dashed edges indicate weaker or less certain connections. Node size corresponds to the number of cells per group. Dot colouring in (B-C) corresponds to macrophage subtypes.

Suppl. Fig. 36: RNA velocity of macrophage subtypes in Xenium panel 1 data with TFvelo. Distribution of TFvelo pseudotime (y-axis) across four panel 1 macrophage cell subtypes along the proposed transdifferentiation trajectory Mac_C1Q/Mac_HMOX1 → Mac_TREM2hi_CCL18hi → Mac_TREM2hi (x-axis). Each subplot (A-L) shows pseudotime distributions for a plaque. Comparisons of pseudotime across subtypes were performed within each sample using the Wilcoxon rank-sum test, and raw p-values were adjusted with the Benjamini-Hochberg method to obtain false discovery rate (FDR) values. (n.s.: FDR >= 0.05, *: FDR < 0.05, **: FDR < 0.01, ***: FDR < 0.001, ****: FDR < 0.0001).

Suppl. Fig. 37: RNA velocity of macrophage subtypes in Xenium panel 2 data with TFvelo. Distribution of TFvelo pseudotime (y-axis) across four panel 2 macrophage cell subtypes along the proposed transdifferentiation trajectory Mac_C1Q/Mac_HMOX1 → Mac_TREM2hi (x-axis). Each subplot (A-L) shows pseudotime distributions for a plaque. Comparisons of pseudotime across subtypes were performed within each sample using the Wilcoxon rank-sum test, and raw p-values were adjusted with the Benjamini-Hochberg method to obtain false discovery rate (FDR) values. (n.s.: FDR ≥ 0.05, *: FDR < 0.05, **: FDR < 0.01, ***: FDR < 0.001, ****: FDR < 0.0001).

Including intronic reads in the new alignment required rerunning our complete scRNA-seq pipeline. This resulted in more cells (without introns: 12,149 cells; with introns: 14,641 cells) and more genes (without introns: 23,362 non-zero count genes; with introns: 28,816 non-zero count genes), however, the overall cell type landscape did not change significantly:

Cell fractions per sample also didn't show significant changes. We updated the corresponding section in the **Results**, as well as subplot Fig. 2F (see original, without and updated, with introns version below):

“...Microfluidics-based capture biases in combination with enzymatic and mechanical digestion is a potential issue with scRNA-Seq, leading to an overrepresentation of immune cells. Indeed, our scRNA-Seq revealed high mean macrophage (27%24%) and T cell fractions (23%29%), together exceeding VSMCs (24%22%) and ECs (15%; Fig. 2F)...”

Without intronic reads

With intronic reads

Including intronic reads, however, resulted in a greater number of differentially expressed genes across cell types. Consequently, we split the original Suppl. Fig. 1 into two figures (updated **Suppl. Fig. 1** and **Suppl. Fig. 2**, see below), and revised the **Results** section to reflect these changes:

*“Comparative analysis of scRNA-Seq data identified cell type-specific differentially expressed genes undetected by bulkRNA-Seq in plaque vs. adjacent control tissues (**Suppl. Fig. 1 G-H A-D**), including ALOX5 IFI27 and HIF1A CTSB in IL10high macrophages fibroblast-like VSMCs. Interestingly,*

stable vs. unstable lesion comparisons revealed an even greater number of DEGs (**Suppl. Fig. ~~1A-F~~ X4 A-K**), such as CXCR4 and HSPA5 in proliferating T cells, both markers of T cell infiltration in atherosclerosis (Dong et al., 2022; Kumar et al., 2006). Additionally, CXCL2 was enriched in ECs, while CCL18 ~~CXCL3~~ was altered in C1Q+ macrophages, both linked to atherosclerosis (A. Singh et al., 2024; Yan et al., 2021).”

Without intronic reads – original version

Suppl. Fig. 1: Differential gene expression analysis results in the single-cell RNA-seq data. Volcano plots depicting differentially expressed genes in the scRNA-seq dataset. Subplots (A - F) show the results of the condition comparison „stable vs. unstable“ within multiple low-level cell types respectively. (G - H): results of the condition comparison „plaque vs. control“ within multiple low-level cell types respectively. The x-axis refers to the \log_2 Fold change of the gene expression levels between the compared conditions, the y-axis shows the $-\log_{10}$ values of the Benjamini-Hochberg adjusted significance level (false discovery rate, FDR) of the gene expression level change. Vertical dashed line: \log_2 Fold change threshold at ± 0.5 , horizontal dashed line: FDR threshold of $-\log_{10}(0.05)$. Red coloured genes are significantly up-, blue coloured genes are significantly downregulated in cells of the first condition in the comparison. Genes with name labels are also present in our Xenium gene panels.

With intronic reads – Updated version

Suppl. Fig. 1: Differential gene expression analysis results in the single-cell RNA-seq data across plaque vs. control conditions. Volcano plots depicting differentially expressed genes in the scRNA-seq dataset. Subplots (A-D) show the results of the condition comparison „plaque vs. control“ within multiple low-level cell types respectively. The x-axis refers to the \log_2 Fold change of the gene expression levels between the compared conditions, the y-axis shows the $-\log_{10}$ values of the Benjamini-Hochberg adjusted significance level (false discovery rate, FDR) of the gene expression level change. Vertical dashed line: \log_2 Fold change threshold at ± 0.5 , horizontal dashed line: FDR threshold of $-\log_{10}(0.05)$. Red coloured genes are significantly up-, blue coloured genes are significantly downregulated in cells of the first condition in the comparison. Genes with name labels are also present in our Xenium gene panels.

With intronic reads – Updated version

Suppl. Fig. 2: Differential gene expression analysis results in the single-cell RNA-seq data across stable vs. unstable conditions. Volcano plots depicting differentially expressed genes in the scRNA-seq dataset. Subplots (A - K) show the results of the condition comparison „stable vs. unstable“ within multiple low-level cell types respectively. The x-axis refers to the \log_2 Fold change of the gene expression levels between the compared conditions, the y-axis shows the $-\log_{10}$ values of the Benjamini-Hochberg adjusted significance level (false discovery rate, FDR) of the gene expression level change. Vertical dashed line: \log_2 Fold change threshold at ± 0.5 , horizontal dashed line: FDR threshold of $-\log_{10}(0.05)$. Red coloured genes are significantly up-, blue coloured genes are significantly downregulated in cells of the first condition in the comparison. Genes with name labels are also present in our Xenium gene panels.

5. Subregion Definition and Reproducibility: Manual annotation of regions may introduce bias. Providing inter-rater reliability or uncertainty metrics would enhance robustness.

Currently, there is no automated subregion annotation tool for human plaques available. Manual subregion annotation is a well established method in the field (J. N. Redgrave et al., 2008; Stary, 2000; Stary et al., 1992, 1994, 1995). In our Xenium approach, as we only had DAPI images available (histological stainings were not embedded in the XOA at that time) and had to manually draw the subregions based on a cardiovascular pathology expert’s annotation.

We edited our **Methods** section for clarification:

“The borders of the annotated subregions were initially determined on the HE stained slides by a pathology expert, then transferred to the DAPI-stained slides manually by manually drawing the determined subregion borders by hand in Xenium Explorer.”

Digital pathology, including automated subregion annotations, is the tool of the future, but unfortunately not available yet for this study.

6. Sample Size for Spatial Analysis: The n=12 for spatial transcriptomics limits generalizability. Interpretations of clustering and risk associations should be more cautiously framed.

We are very well aware that our sample size for spatial transcriptomics is limited but want to stress once more that the overall concept of our study was not to use spatial transcriptomics for discovery, but for integration and deconvolution with our in parallel performed single cell- and bulkRNASeq study. In response to the reviewer’s suggestion, we carefully assessed the text again for cautious framing and believe that we address this appropriately in the resubmitted manuscript. As aforementioned to reviewers 1 & 2, compared with the other existing spatial datasets on human plaques, our effort provides the largest dataset to date:

	Sun et al, JACC, 2023	Bleckwehl et al, NCVR, 2024	Lai et al, NCVR, 2025	Gastanadui et al, ATVB, 2024	Campos et al, EMBO Mol. Med., 2025	Pauli et al, under review, 2025
tissue [artery]	Carotid	Carotid + Coronary (for spatial only coronary)	Carotid	Coronary	Coronary	Carotid
bulk seq [sample size]	n=163	-	n=149 PTLOs only (no full tissue)	- (assumptions from spatial)	-	n=297

scSeq [patients]	-	n=13 (previous studies; no own data)	n=9	- (assumptions from spatial)	-	n=17
spatial method	VISIUM (FFPE; probe-based whole transcriptome)	VISIUM (FFPE; probe-based whole transcriptome)	StereoSeq (fresh frozen tissue; ~ VISIUM HD, capture based)	GeoMX	GeoMX & CosMX (~ Xenium)	Xenium
spatial samples [samples; therein controls]	n=6 (no controls)	n=12 (2 controls)	n=10 (no controls)	n=10 (no controls)	n=9 (2 controls)	n=16 (4 controls)
symptomatic/ asymptomatic status	no report in bulk seq; spatial all symptomatic	no report	report in PTLO bulk seq; spatial all symptomatic	no report	no report	report in spatial

7. Neighbourhood Cluster Analysis: Statistical support (e.g., permutation tests, spatial proximity significance) for proposed functional zones is missing and should be included.

We thank the reviewer for this valuable comment. To provide statistical support, we have performed permutation tests by shuffling cell substate labels within samples 10,000 times, to create a background distribution of cell neighbourhoods. We then calculated neighbourhood enrichment score, a test metric defined as the total number of unique cell-cell adjacency edges between substate pairs within a cluster, using the generated background local neighbourhoods. Enrichment scores were calculated within cells of the original clusters, and the significance of the observed clusters was assessed by comparing the background distributions to the values obtained from the observed neighbourhoods.

Permutation tests of neighbourhood enrichment score detected significantly enriched or depleted cell substates in all clusters, both for panel 1 and 2. Due to space limitations, the complete results of the enrichment permutation tests are provided in **Suppl. Tables K1** and **K2**, while enrichment heatmaps showing significantly enriched or depleted substates are included in the newly created **Suppl. PDFs D1** and **D2**. We have expanded the Methods section accordingly (see revised text below):

“...In the final step we performed Leiden-clustering of the cell neighbourhood matrices using scanpy’s pp.neighbors() and tl.leiden() functions, resulting in cell

neighbourhood clusters (**Suppl. Fig. 60, Suppl. Fig. 61**). To provide statistical support for the clusters, we performed permutation testing using neighbourhood enrichment score, a metric defined as the total number of unique cell-cell adjacency edges between substate pairs within a cluster. We hypothesised that if our clusters indeed capture distinct spatial patterns, by calculating the neighbourhood enrichment score, we could identify which cell substate pairs are significantly enriched in each other's neighbourhoods within a cluster.

First, we created a background distribution of cell neighbourhood matrices by randomly shuffling cell substate labels 10,000 times. Because samples exhibited large differences in cell substate composition, we performed label shuffling within samples. This was necessary since shuffling across samples would have violated the exchangeability assumption of permutation testing (Good, 2002) and could have led to inflated p-values. Next, for neighbourhood enrichment analysis, we constructed spatial neighbourhood graphs by connecting cells within a 100 µm radius, with each edge representing a pair of cells within this distance. For each neighbourhood cluster, we restricted the graph to the cells assigned to that cluster and calculated neighbourhood enrichment as the total number of unique undirected adjacency edges connecting substate pairs within the cluster-specific graph. This was again computed for both the observed and randomly shuffled substates, resulting in one enrichment score matrix per cluster for the observed case and 10,000 matrices per cluster for the permutation case. Empirical left and right p-values were calculated using the Monte-Carlo p-value method as described by Phipson et al (Phipson & Smyth, 2016):

$$p_{left} = \frac{l+1}{N+1}; p_{right} = \frac{r+1}{N+1};$$

where p_{left} and p_{right} are the left and right empirical p-values, l is the number of times enrichment score from the permutation was lower than the enrichment score of the observed case, r is the number of times enrichment score from the permutation exceeded the enrichment score of the observed case, and N is the number of permutations. In addition, Agresti-Coull confidence intervals (CI) (Bozhko et al., 2021) of the empirical left and right p-values were also calculated. Two-sided empirical p-values and their Agresti-Coull confidence intervals were calculated using the formula applied in `scipy.stats.permutation_test()` function (Scientific Python Forum, n.d.):

$$p_{two-sided} = \min(2 \times p_{left}, 2 \times p_{right});$$

$$CI_{two-sided} = \min(2 \times CI_{left}, 2 \times CI_{right}).$$

Raw empirical two-sided p-values were then corrected for multiple testing by applying the Benjamini-Hochberg correction method, and reported as false discovery rate (FDR).

Permutation test results for both panel 1 (**Suppl. PDF 6, Suppl. Table 17**) and panel 2 (**Suppl. PDF 7, Suppl. Table 17**) neighbourhood clusters identified significantly enriched or depleted substate pairs in every cluster compared to the random background. Based on these results we believe our neighbourhood

clustering approach was able to identify spatial domains with significantly different compositional patterns, even across multiple samples. High resolution spatial scatterplots of all samples with cells coloured by low-level cell types and neighbourhood clusters are contained in **Suppl. PDF 8** for panel 1 genes, and in **Suppl. PDF 9** for panel 2 genes."

8. Clinical Utility and Predictive Value: The translational relevance of findings should be discussed. Can DEGs or spatial patterns serve as prognostic markers for plaque rupture?

This is an interesting thought and question by the reviewer. We indeed believe that the combination of multiple transcriptomic profiling methods will help pave the way for novel therapies as well as biomarkers of carotid plaque stability. What is currently lacking is data from longitudinal carotid disease and stroke studies, in which these potential markers could be tested for their prognostic value. Integration of additional data information from histology (using digital pathology tools as mentioned multiple times in this point-by-point response) on plaque stability and even more promising correlation to imaging (in particular CTs from carotid arteries) could further enhance decision making in patients with asymptomatic carotid artery disease. Our approach of assessing plaque instability with integrated multi-transcriptomics enhances our understanding of advanced plaque biology. Novel therapeutic targets can be linked, based on our data to cell type and spatial location (and thus relevance), within advanced plaques. Here, our dataset provides the most complete transcriptomic view on early vs. advanced plaques to date.

We addressed this thought in our revised **Discussion** section:

"Future studies on the induction and therapeutic targeting of cellular plasticity may yield novel markers and interventions for plaque vulnerability, together with digital pathology approaches and enhanced vascular imaging modalities."

Minor Comments

1. Terminology Clarification: Terms like 'structured' vs. 'chaotic' could be more quantitatively or mechanistically defined.

In this work, we tried to use simplified and understandable terms. We discussed this internally quite a lot, as it is a difficult task and challenging to choose additional or other terms to enhance the readers' understanding of plaque morphology. We introduced these terms for this study to highlight the complexity of human plaques. We want to show that easy discriminations like "stable and unstable" or "symptomatic and asymptomatic" (terms that are used in all other studies that deal with plaques in a spatial context (Bleckwehl et al., 2025; Gastanadui et al., 2024; Lai et al., 2025; Sun et al., 2023) are not sufficient to capture their complexity. **Fig. 5A**, shows the number

of cells in each cluster which serves as a quantitative definition, furthermore we speculate about mechanisms based on spatial location.

Based on the reviewer's comment we adapted the **Results** section for clarification:

“For intelligibility, we decided to term the two main groups of our plaque clusters “structured” (morphological clusters 1 & 2) ~~or~~ and “chaotic” (morphological clusters 3 & 4).”

2. Figure Readability: Dense color legends in Figures 6–8 make interpretation difficult. Consider enhancing clarity.

We appreciate this comment. At the initial submission, the figures were uploaded at 96 dpi due to a misunderstanding, which reduced their readability. To improve clarity, we have increased the dot size in the spatial scatterplots of Figures 6–8 (renumbered as **Figures 7–9**) and re-uploaded them at 300 dpi.

3. Figure 2 caption: “-log₁₀ value” => “-log₁₀ p-value”

We corrected the figure caption accordingly.

Reviewer #3 (Remarks on code availability):

No code is provided.

We apologize for the delay in providing the code. Setting up the Docker environment took additional time due to conflicting package versions. In the meantime, we prepared code to reproduce the main figures, as requested by the editors, since running the full analysis pipeline on raw data is not feasible due to storage constraints and time limitations. All code will be made publicly available upon publication.

References

- Allahverdian, S., Chaabane, C., Boukais, K., Francis, G. A., & Bochaton-Piallat, M.-L. (2018). Smooth muscle cell fate and plasticity in atherosclerosis. *Cardiovascular Research*, 114(4), 540–550.
- Alsaigh, T., Evans, D., Frankel, D., & Torkamani, A. (2022). Decoding the transcriptome of calcified atherosclerotic plaque at single-cell resolution. *Communications Biology*, 5(1), 1084.
- Bashore, A. C., Yan, H., Xue, C., Zhu, L. Y., Kim, E., Mawson, T., Coronel, J., Chung, A., Sachs, N., Ho, S., Ross, L. S., Kissner, M., Passegué, E., Bauer, R. C., Maegdefessel, L., Li, M., & Reilly, M. P. (2024). High-dimensional single-cell multimodal landscape of human carotid atherosclerosis. *Arteriosclerosis, Thrombosis, and Vascular Biology*, 44(4), 930–945.
- Bergen, V., Lange, M., Peidli, S., Wolf, F. A., & Theis, F. J. (2020). Generalizing RNA velocity to transient cell states through dynamical modeling. *Nature Biotechnology*, 38(12), 1408–1414.
- Bleckwehl, T., Babler, A., Tebens, M., Maryam, S., Nyberg, M., Bosteen, M., Halder, M., Shaw, I., Fleig, S., Pyke, C., Hvid, H., Voetmann, L. M., van Buul, J. D., Sluimer, J. C., Das, V., Baumgart, S., Kramann, R., & Hayat, S. (2025). Encompassing view of spatial and single-cell RNA sequencing renews the role of the microvasculature in human atherosclerosis. *Nature Cardiovascular Research*, 4(1), 26–44.
- Bonfiglio, C. A., Lacy, M., Triantafyllidou, V., Farina, F. M., Janjic, A., Nitz, K., Wu, Y., Bazioti, V., Avçilar-Kücükgoze, I., Marques, Y. F. S., Joppich, M., Kumkum, M., Röß, K., Venkatasubramani, A. V., Imhof, A., Enard, W., Maegdefessel, L., de Winther, M., Weber, C., ... Atzler, D. (2025). Ezh2 shapes T cell plasticity to drive atherosclerosis. *Circulation*, 151(19), 1391–1408.
- Bozhko, S., Brügggen, G. V. D., & Brandenburg, B. B. (2021). Monte Carlo Response-Time Analysis. *IEEE Real-Time Systems Symposium*, 342–355.
- Campos, J., McMurray, J. L., Certo, M., Hardikar, K., Morse, C., Corfield, C., Weigand, B. M., Yang, K., Shoaran, M., Otto, T. D., Neil, D., Maffia, P., & Mauro, C. (2025). Spatial transcriptomics elucidates localized immune responses in atherosclerotic coronary artery. *EMBO Molecular Medicine*.
<https://doi.org/10.1038/s44321-025-00280-w>
- Chou, E. L., Lino Cardenas, C. L., Chaffin, M., Arduini, A. D., Juric, D., Stone, J. R., LaMuraglia, G. M., Eagleton, M. J., Conrad, M. F., Isselbacher, E. M., Ellinor, P. T., & Lindsay, M. E. (2022). Vascular smooth muscle cell phenotype switching in carotid atherosclerosis. *JVS-Vascular Science*, 3, 41–47.
- Depuydt, M. A. C., Schaftenaar, F. H., Prange, K. H. M., Boltjes, A., Hemme, E., Delfos, L., de Mol, J., de Jong, M. J. M., Bernabé Kleijn, M. N. A., Peeters, J. A. H. M., Goncalves, L., Wezel, A., Smeets, H. J., de Borst, G. J., Foks, A. C., Pasterkamp, G., de Winther, M. P. J., Kuiper, J., Bot, I., & Slütter, B. (2023). Single-cell T cell receptor sequencing of paired human atherosclerotic plaques and blood reveals autoimmune-like features of expanded effector T cells. *Nature Cardiovascular Research*, 2(2), 112–125.

- Dib, L., Koneva, L. A., Edsfeldt, A., Zurke, Y.-X., Sun, J., Nitulescu, M., Attar, M., Lutgens, E., Schmidt, S., Lindholm, M. W., Choudhury, R. P., Cassimjee, I., Lee, R., Handa, A., Goncalves, I., Sansom, S. N., & Monaco, C. (2023). Lipid-associated macrophages transition to an inflammatory state in human atherosclerosis increasing the risk of cerebrovascular complications. *Nature Cardiovascular Research*, 2(7), 656–672.
- Dong, W., Du, D., & Huang, H. (2022). HSPA5 is a prognostic biomarker correlated with immune infiltrates in thyroid carcinoma. *Endokrynologia Polska*, 73(4), 680–689.
- Fasolo, F., Jin, H., Winski, G., Chernogubova, E., Pauli, J., Winter, H., Li, D. Y., Glukha, N., Bauer, S., Metschl, S., Wu, Z., Koschinsky, M. L., Reilly, M., Pelisek, J., Kempf, W., Eckstein, H.-H., Soehnlein, O., Matic, L., Hedin, U., ... Maegdefessel, L. (2021). Long noncoding RNA MIAT controls advanced atherosclerotic lesion formation and plaque destabilization. *Circulation*, 144(19), 1567–1583.
- Fernandez, D. M., Rahman, A. H., Fernandez, N. F., Chudnovskiy, A., Amir, E.-A. D., Amadori, L., Khan, N. S., Wong, C. K., Shamailova, R., Hill, C. A., Wang, Z., Remark, R., Li, J. R., Pina, C., Faries, C., Awad, A. J., Moss, N., Bjorkegren, J. L. M., Kim-Schulze, S., ... Giannarelli, C. (2019). Single-cell immune landscape of human atherosclerotic plaques. *Nature Medicine*, 25(10), 1576–1588.
- Fidler, T. P., Dunbar, A., Kim, E., Hardaway, B., Pauli, J., Xue, C., Abramowicz, S., Xiao, T., O'Connor, K., Sachs, N., Wang, N., Maegdefessel, L., Levine, R., Reilly, M., & Tall, A. R. (2024). Suppression of IL-1 β promotes beneficial accumulation of fibroblast-like cells in atherosclerotic plaques in clonal hematopoiesis. *Nature Cardiovascular Research*, 3(1), 60–75.
- Finn, A. V., Kolodgie, F. D., & Virmani, R. (2010). Correlation between carotid intimal/medial thickness and atherosclerosis: a point of view from pathology. *Arteriosclerosis, Thrombosis, and Vascular Biology*, 30(2), 177–181.
- Gastanadui, M. G., Margaroli, C., Litovsky, S., Richter, R. P., Wang, D., Xing, D., Wells, J. M., Gaggari, A., Nanda, V., Patel, R. P., & Payne, G. A. (2024). Spatial transcriptomic approach to understanding coronary atherosclerotic plaque stability. *Arteriosclerosis, Thrombosis, and Vascular Biology*, 44(11), e264–e276.
- Good, P. I. (2002). Extensions of the concept of exchangeability and their applications. *Journal of Modern Applied Statistical Methods: JMASM*, 1(2), 243–247.
- Gupta, A., Baradaran, H., Schweitzer, A. D., Kamel, H., Pandya, A., Delgado, D., Dunning, A., Mushlin, A. I., & Sanelli, P. C. (2013). Carotid plaque MRI and stroke risk: a systematic review and meta-analysis. *Stroke; a Journal of Cerebral Circulation*, 44(11), 3071–3077.
- Harman, J. L., & Jørgensen, H. F. (2019). The role of smooth muscle cells in plaque stability: Therapeutic targeting potential. *British Journal of Pharmacology*, 176(19), 3741–3753.
- Kubo, T., Imanishi, T., Takarada, S., Kuroi, A., Ueno, S., Yamano, T., Tanimoto, T., Matsuo, Y., Masho, T., Kitabata, H., Tsuda, K., Tomobuchi, Y., & Akasaka, T. (2007). Assessment of culprit lesion morphology in acute myocardial infarction:

- ability of optical coherence tomography compared with intravascular ultrasound and coronary angiography. *Journal of the American College of Cardiology*, *50*(10), 933–939.
- Kumar, A., Humphreys, T. D., Kremer, K. N., Bramati, P. S., Bradfield, L., Edgar, C. E., & Hedin, K. E. (2006). CXCR4 physically associates with the T cell receptor to signal in T cells. *Immunity*, *25*(2), 213–224.
- Lai, Z., Kong, D., Li, Q., Wang, Y., Li, K., Duan, X., Shao, J., Xie, Y., Chen, J., Zhang, T., Feng, Y., Deng, H., Wang, J., Wang, C., Shu, K., Zhao, H., Du, H., Jia, C., Dai, H., ... Liu, X. (2025). Single-cell spatial transcriptomics of tertiary lymphoid organ-like structures in human atherosclerotic plaques. *Nature Cardiovascular Research*, *4*(5), 547–566.
- La Manno, G., Soldatov, R., Zeisel, A., Braun, E., Hochgerner, H., Petukhov, V., Lidschreiber, K., Kastrioti, M. E., Lönnerberg, P., Furlan, A., Fan, J., Borm, L. E., Liu, Z., van Bruggen, D., Guo, J., He, X., Barker, R., Sundström, E., Castelo-Branco, G., ... Kharchenko, P. V. (2018). RNA velocity of single cells. *Nature*, *560*(7719), 494–498.
- Langley, S. R., Willeit, K., Didangelos, A., Matic, L. P., Skroblin, P., Barallobre-Barreiro, J., Lengquist, M., Rungger, G., Kapustin, A., Kedenko, L., Molenaar, C., Lu, R., Barwari, T., Suna, G., Yin, X., Iglšeder, B., Paulweber, B., Willeit, P., Shalhoub, J., ... Mayr, M. (2017). Extracellular matrix proteomics identifies molecular signature of symptomatic carotid plaques. *The Journal of Clinical Investigation*, *127*(4), 1546–1560.
- Lavallée, P. C., Charles, H., Albers, G. W., Caplan, L. R., Donnan, G. A., Ferro, J. M., Hennerici, M. G., Labreuche, J., Molina, C., Rothwell, P. M., Steg, P. G., Touboul, P.-J., Uchiyama, S., Vicaut, É., Wong, L. K. S., Amarenco, P., & TIAregistry.org Investigators. (2023). Underlying causes of TIA and minor ischemic stroke and risk of major vascular events. *JAMA Neurology*, *80*(11), 1199–1208.
- Libby, P., Buring, J. E., Badimon, L., Hansson, G. K., Deanfield, J., Bittencourt, M. S., Tokgözoğlu, L., & Lewis, E. F. (2019). Atherosclerosis. *Nature Reviews. Disease Primers*, *5*(1), 56.
- Li, D. Y., Kundu, S., Cheng, P., Gu, W., Jackson, W., Zhao, Q., Nguyen, T., Worssam, M., Monteiro, J. P., Caceres, R. D., Dale, S., Palmisano, B., Weldy, C. S., Kundu, R., Kundaje, A., Wirka, R., & Quertermous, T. (2025). Vascular smooth muscle cell atherosclerosis trajectories characterized at single cell resolution identify causal transcriptomic and epigenomic mechanisms of disease risk. In *bioRxiv.org*. <https://doi.org/10.1101/2025.06.04.655863>
- Li, J., Pan, X., Yuan, Y., & Shen, H.-B. (2024). TFvelo: gene regulation inspired RNA velocity estimation. *Nature Communications*, *15*(1), 1387.
- Li, Y., Zhu, H., Zhang, Q., Han, X., Zhang, Z., Shen, L., Wang, L., Lui, K. O., He, B., & Zhou, B. (2021). Smooth muscle-derived macrophage-like cells contribute to multiple cell lineages in the atherosclerotic plaque. *Cell Discovery*, *7*(1), 111.
- Longo, S. K., Guo, M. G., Ji, A. L., & Khavari, P. A. (2021). Integrating single-cell and spatial transcriptomics to elucidate intercellular tissue dynamics. *Nature Reviews. Genetics*, *22*(10), 627–644.

- Lovett, J. K., Coull, A. J., & Rothwell, P. M. (2004). Early risk of recurrence by subtype of ischemic stroke in population-based incidence studies. *Neurology*, *62*(4), 569–573.
- Masawa, N., Glagov, S., & Zarins, C. K. (1994). Quantitative morphologic study of intimal thickening at the human carotid bifurcation: II. The compensatory enlargement response and the role of the intima in tensile support. *Atherosclerosis*, *107*(2), 147–155.
- Ma, S. D., Mussbacher, M., & Galkina, E. V. (2021). Functional role of B cells in atherosclerosis. *Cells (Basel, Switzerland)*, *10*(2), 270.
- Mocci, G., Sukhvasi, K., Örd, T., Bankier, S., Singha, P., Arasu, U. T., Agbabiage, O. O., Mäkinen, P., Ma, L., Hodonsky, C. J., Aherrahrou, R., Muhl, L., Liu, J., Gustafsson, S., Byandelger, B., Wang, Y., Koplev, S., Lendahl, U., Owens, G. K., ... Björkegren, J. L. M. (2024). Single-cell gene-regulatory networks of advanced symptomatic atherosclerosis. *Circulation Research*, *134*(11), 1405–1423.
- Mokry, M., Boltjes, A., Slenders, L., Bel-Bordes, G., Cui, K., Brouwer, E., Mekke, J. M., Depuydt, M. A. C., Timmerman, N., Waissi, F., Verwer, M. C., Turner, A. W., Khan, M. D., Hodonsky, C. J., Benavente, E. D., Hartman, R. J. G., van den Dungen, N. A. M., Lansu, N., Nagyova, E., ... Pasterkamp, G. (2022). Transcriptomic-based clustering of human atherosclerotic plaques identifies subgroups with different underlying biology and clinical presentation. *Nature Cardiovascular Research*, *1*(12), 1140–1155.
- Ord, J. K., & Getis, A. (1995). Local spatial autocorrelation statistics: Distributional issues and an application. *Geographical Analysis*, *27*(4), 286–306.
- Paloschi, V., Pauli, J., Winski, G., Wu, Z., Li, Z., Botti, L., Meucci, S., Conti, P., Rogowitz, F., Glukha, N., Hummel, N., Busch, A., Chernogubova, E., Jin, H., Sachs, N., Eckstein, H.-H., Dueck, A., Boon, R. A., Bausch, A. R., & Maegdefessel, L. (2024). Utilization of an artery-on-a-chip to unravel novel regulators and therapeutic targets in vascular diseases. *Advanced Healthcare Materials*, *13*(6), e2302907.
- Paraskevas, K. I., & AbuRahma, A. F. (2024). A comparison of the 2022 Society for Vascular Surgery and the 2023 European Society for Vascular Surgery guidelines for the management of patients with asymptomatic and symptomatic carotid stenosis. *Journal of Vascular Surgery*, *79*(6), 1272–1275.
- Parma, L., Sachs, N., Sobczak, N., Li, Z., Merchant, K., Pauli, J., Depuydt, M. A. C., Wezel, A., Smeets, H. J., Bot, I., Slütter, B., Maegdefessel, L., Weber, C., Duchêne, J., & Megens, R. T. A. (2025). CXCL12 derived from ACKR1⁺ intraplaque neovessels mediates CD8⁺ T cell recruitment in human atherosclerosis. *Circulation*, *151*(8), 581–584.
- Pasterkamp, G., Schoneveld, A. H., Hillen, B., Banga, J. D., Haudenschild, C. C., & Borst, C. (1998). Is plaque formation in the common carotid artery representative for plaque formation and luminal stenosis in other atherosclerotic peripheral arteries? A post mortem study. *Atherosclerosis*, *137*(1), 205–210.
- Pathology of the Vulnerable Plaque. (2006). *Journal of the American College of Cardiology*, *47*(8), C13–C18.

- Phipson, B., & Smyth, G. K. (2016). Permutation p-values should never be zero: calculating exact p-values when permutations are randomly drawn. In *arXiv [stat.AP]*. arXiv. <http://arxiv.org/abs/1603.05766>
- Pourteymour, S., Fan, J., Majhi, R. K., Guo, S., Sun, X., Huang, Z., Liu, Y., Winter, H., Bäcklund, A., Skenteris, N.-T., Chernogubova, E., Werngren, O., Li, Z., Skogsberg, J., Li, Y., Matic, L., Hedin, U., Maegdefessel, L., Ehrenborg, E., ... Jin, H. (2024). PIEZO1 targeting in macrophages boosts phagocytic activity and foam cell apoptosis in atherosclerosis. *Cellular and Molecular Life Sciences: CMLS*, *81*(1), 331.
- Redgrave, J. N. E., Lovett, J. K., Gallagher, P. J., & Rothwell, P. M. (2006). Histological assessment of 526 symptomatic carotid plaques in relation to the nature and timing of ischemic symptoms: the Oxford plaque study. *Circulation*, *113*(19), 2320–2328.
- Redgrave, J. N., Gallagher, P., Lovett, J. K., & Rothwell, P. M. (2008). Critical cap thickness and rupture in symptomatic carotid plaques: the oxford plaque study. *Stroke; a Journal of Cerebral Circulation*, *39*(6), 1722–1729.
- Saba, L., Cau, R., Vergallo, R., Kooi, M. E., Staub, D., Faa, G., Congiu, T., Ntaios, G., Wasserman, B. A., Benson, J., Nardi, V., Kawakami, R., Lanzino, G., Virmani, R., & Libby, P. (2025). Carotid artery atherosclerosis: mechanisms of instability and clinical implications. *European Heart Journal*, *46*(10), 904–921.
- Schmeisser, A., Christoph, M., Augstein, A., Marquetant, R., Kasper, M., Braun-Dullaeus, R. C., & Strasser, R. H. (2006). Apoptosis of human macrophages by Flt-4 signaling: implications for atherosclerotic plaque pathology. *Cardiovascular Research*, *71*(4), 774–784.
- Scientific Python Forum. (n.d.). *permutation_test* — *SciPy v1.16.1 Manual*. Retrieved August 25, 2025, from https://docs.scipy.org/doc/scipy/reference/generated/scipy.stats.permutation_test.html
- Shankman, L. S., Gomez, D., Cherepanova, O. A., Salmon, M., Alencar, G. F., Haskins, R. M., Swiatlowska, P., Newman, A. A. C., Greene, E. S., Straub, A. C., Isakson, B., Randolph, G. J., & Owens, G. K. (2015). KLF4-dependent phenotypic modulation of smooth muscle cells has a key role in atherosclerotic plaque pathogenesis. *Nature Medicine*, *21*(6), 628–637.
- Sharma, D., Worssam, M. D., Pedroza, A. J., Dalal, A. R., Alemany, H., Kim, H.-J., Kundu, R., Fischbein, M. P., Cheng, P., Wirka, R., & Quertermous, T. (2024). Comprehensive Integration of Multiple Single-Cell Transcriptomic Data Sets Defines Distinct Cell Populations and Their Phenotypic Changes in Murine Atherosclerosis. *Arteriosclerosis, Thrombosis, and Vascular Biology*, *44*(2), 391–408.
- Singh, A., Kraaijeveld, A. O., Curaj, A., Wichapong, K., Hammerich, L., de Jager, S. C. A., Bot, I., Atamas, S. P., van Berkel, T. J. C., Jukema, J. W., Comerford, I., McColl, S. R., Mees, B., Heemskerk, J. W. M., Nicolaes, G. A. F., Hackeng, T., Liehn, E. A., Tacke, F., & Biessen, E. A. L. (2024). CCL18 aggravates atherosclerosis by inducing CCR6-dependent T-cell influx and polarization. *Frontiers in Immunology*, *15*, 1327051.

- Singh, P., Sun, J., Cavalera, M., Al-Sharify, D., Matthes, F., Barghouth, M., Tengryd, C., Dunér, P., Persson, A., Sundius, L., Nitulescu, M., Bengtsson, E., Rattik, S., Engelbertsen, D., Orho-Melander, M., Nilsson, J., Monaco, C., Goncalves, I., & Edsfeldt, A. (2024). Dysregulation of MMP2-dependent TGF-β2 activation impairs fibrous cap formation in type 2 diabetes-associated atherosclerosis. *Nature Communications*, 15(1), 10464.
- Sinha, A., Sachs, N., Kratz, E., Pauli, J., Steigerwald, S., Albrecht, V., Nordmann, T., Ugur, E., Rodriguez, E. H., Engl, M.-L., Skowronek, P., von Scheidt, M., Winter, H., Branzan, D., Schunkert, H., Maegdefessel, L., & Mann, M. (2025). Spatially resolved proteomic signatures of atherosclerotic carotid artery disease. In *medRxiv*. <https://doi.org/10.1101/2025.02.09.25321955>
- Song, Y., Zhang, Q., Ban, R., Zhao, X., Sun, H., Lin, J., Guo, T., Wang, T., Xia, K., Xin, Z., Zhang, G., Jia, X., & Xia, Z. (2024). Single-nucleus RNA sequencing reveals that macrophages and smooth muscle cells promote carotid atherosclerosis progression through mitochondrial autophagy. *Medicine*, 103(7), e37171.
- Sary, H. C. (2000). Natural history and histological classification of atherosclerotic lesions: an update: An update. *Arteriosclerosis, Thrombosis, and Vascular Biology*, 20(5), 1177–1178.
- Sary, H. C., Blankenhorn, D. H., Chandler, A. B., Glagov, S., Insull, W., Jr, Richardson, M., Rosenfeld, M. E., Schaffer, S. A., Schwartz, C. J., & Wagner, W. D. (1992). A definition of the intima of human arteries and of its atherosclerosis-prone regions. A report from the Committee on Vascular Lesions of the Council on Arteriosclerosis, American Heart Association. *Arteriosclerosis and Thrombosis: A Journal of Vascular Biology*, 12(1), 120–134.
- Sary, H. C., Chandler, A. B., Dinsmore, R. E., Fuster, V., Glagov, S., Insull, W., Jr, Rosenfeld, M. E., Schwartz, C. J., Wagner, W. D., & Wissler, R. W. (1995). A definition of advanced types of atherosclerotic lesions and a histological classification of atherosclerosis. A report from the Committee on Vascular Lesions of the Council on Arteriosclerosis, American Heart Association: A report from the Committee on Vascular Lesions of the council on arteriosclerosis, American heart association. *Circulation*, 92(5), 1355–1374.
- Sary, H. C., Chandler, A. B., Glagov, S., Guyton, J. R., Insull, W., Jr, Rosenfeld, M. E., Schaffer, S. A., Schwartz, C. J., Wagner, W. D., & Wissler, R. W. (1994). A definition of initial, fatty streak, and intermediate lesions of atherosclerosis. A report from the Committee on Vascular Lesions of the Council on Arteriosclerosis, American Heart Association. *Arteriosclerosis and Thrombosis: A Journal of Vascular Biology*, 14(5), 840–856.
- Sun, J., Singh, P., Shami, A., Kluza, E., Pan, M., Djordjevic, D., Michaelsen, N. B., Kennbäck, C., van der Wel, N. N., Orho-Melander, M., Nilsson, J., Formentini, I., Conde-Knape, K., Lutgens, E., Edsfeldt, A., & Gonçaves, I. (2023). Spatial transcriptional mapping reveals site-specific pathways underlying human atherosclerotic plaque rupture. *Journal of the American College of Cardiology*, 81(23), 2213–2227.
- Tan, J. M. E., Cheng, L., Calhoun, R. P., Weller, A. H., Drareni, K., Fong, S.,

- Barbara, E., Lim, H.-W., Xue, C., Winter, H., Auguste, G., Miller, C. L., Reilly, M. P., Maegdefessel, L., Lutgens, E., & Seale, P. (2025). PRDM16 controls smooth muscle cell fate in atherosclerosis. In *bioRxiv.org*.
<https://doi.org/10.1101/2025.02.19.639186>
- Traeuble, K., Munz, M., Pauli, J., Sachs, N., Vafadarnejad, E., Carrillo-Roa, T., Maegdefessel, L., Kastner, P., & Heinig, M. (2024). Integrated single-cell atlas of human atherosclerotic plaques. In *bioRxiv* (p. 2024.09.11.612431).
<https://doi.org/10.1101/2024.09.11.612431>
- Underhill, H. R., Hatsukami, T. S., Fayad, Z. A., Fuster, V., & Yuan, C. (2010). MRI of carotid atherosclerosis: clinical implications and future directions. *Nature Reviews. Cardiology*, 7(3), 165–173.
- Velican, C. (1969). Relationship between regional aortic susceptibility to atherosclerosis and macromolecular structural stability. *Journal of Atherosclerosis Research*, 9(2), 193–201.
- Virmani, R., Kolodgie, F. D., Burke, A. P., Farb, A., & Schwartz, S. M. (2000). Lessons From Sudden Coronary Death. *Arteriosclerosis, Thrombosis, and Vascular Biology*. <https://doi.org/10.1161/01.ATV.20.5.1262>
- Wang, Y., Dubland, J. A., Allahverdian, S., Asonye, E., Sahin, B., Jaw, J. E., Sin, D. D., Seidman, M. A., Leeper, N. J., & Francis, G. A. (2019). Smooth muscle cells contribute the majority of foam cells in ApoE (apolipoprotein E)-deficient mouse atherosclerosis. *Arteriosclerosis, Thrombosis, and Vascular Biology*, 39(5), 876–887.
- Wang, Y., Li, G., Chen, B., Shakir, G., Volz, M., van der Vorst, E. P. C., Maas, S. L., Geiger, M., Jethwa, C., Bartelt, A., Li, Z., Wettich, J., Sachs, N., Maegdefessel, L., Nazari Jahantigh, M., Hristov, M., Lacy, M., Lutz, B., Weber, C., ... Steffens, S. (2024). Myeloid cannabinoid CB1 receptor deletion confers atheroprotection in male mice by reducing macrophage proliferation in a sex-dependent manner. *Cardiovascular Research*, 120(12), 1411–1426.
- Wolf, F. A., Hamey, F. K., Plass, M., Solana, J., Dahlin, J. S., Göttgens, B., Rajewsky, N., Simon, L., & Theis, F. J. (2019). PAGA: graph abstraction reconciles clustering with trajectory inference through a topology preserving map of single cells. *Genome Biology*, 20(1), 59.
- Yan, Y., Thakur, M., van der Vorst, E. P. C., Weber, C., & Döring, Y. (2021). Targeting the chemokine network in atherosclerosis. *Atherosclerosis*, 330, 95–106.

Rebuttal, point-by-point response

Reviewer #1 (Remarks to the Author):

The authors addressed all concerns. However, figure 2F-I is still unsatisfactory. Please show the immune cells using a different y axis scale. Since there are few immune cells, differences cannot be appreciated from the current figure

Answer: We appreciate the positive comments by this reviewer. We have made adaptations to Figures 2F-I in our revised manuscript to better address and visualize the low amount of immune cells by choosing a varying scale for the y-axes.

Reviewer #2 (Remarks to the Author):

Pauli et al. represent their investigation seeking to integrate multiple transcriptomic methodologies (bulk RNA seq, scRNAseq, and spatial transcriptomics) as a means to molecularly define carotid atherosclerotic disease. Using arterial and atheroma samples collected during CEA, the investigators attempt to elucidate a spatially-resolved molecular architecture underlying plaque development. The authors highlight unique plaque morphologies linked to different cell type compositions, disease heterogeneity (i.e. symptomatology), and clinical demographics (i.e. smoking, etc) with inference towards plaque development and stability.

Previous critiques raised questions regarding 1) associations with disease symptomatology, 2) Comparison of the presented results to previously published data (coronary and carotid), 3) Novelty and clinical utility of the results, and 4) clarification of methodologies and sample size. To this end, the investigators have provided a robust response that is highly responsive to the reviewer concerns. This includes adjustments to the data presentation, new analysis, and additional commentary throughout the manuscript.

Regarding novelty, the team has provided sound arguments highlighting the relative impact their results may have on the molecular understanding of carotid artery disease and plaque progression. While the HMOX1 and TREM2 phenotypes are well established, the manuscript provides other novel findings including 1) spatial resolution of macrophages at the unstable plaque shoulder, 2) EC co-localization for neovascularization and immune cell entry into the necrotic core / plaque, 3) SMC transitions associated with plaque heterogeneity. While the clinical utility and implementation of many of these results remains to be determined, the work nonetheless provides a thorough examination of carotid atherosclerotic disease.

Other Criticisms that have been adequately addressed:

1. Association with symptomatology is uncertain. Concerns have been well addressed and edited within the manuscript.
2. Surprising lack of DEG in bulk RNAseq. Concerns have been well addressed and edited within the manuscript.
3. Underpowered study with significant demographic heterogeneity. Concerns have been well addressed and edited within the manuscript.
4. Biased spatial transcriptomic approach based off Panels 1 and 2. Concerns have

been well addressed and edited within the manuscript.

Answer: We appreciate that the reviewer agrees with us on the clarifications made during the revision process of our manuscript on novelty, clinical utility and selection of sample characterization utilized in our study.

Reviewer #3 (Remarks to the Author):

All my comments are adequately addressed.

Reviewer #3 (Remarks on code availability):

The code provided in code ocean are well organized. But the code ocean platform has some issues running the code with the 'Reproducible Run'. Also, only R code shows up properly. The ipython notebook must be downloaded and run locally, which is beyond my bandwidth.

Answer: We thank the reviewer for acknowledging that we have successfully addressed all of her/his previous concerns. We believe to have cleared all confusion regarding code availability in Code Ocean.